# Divergent evolutionary strategies pre-empt tissue collision in gastrulation

Bipasha Dey[1,4], Verena Kaul[2,3,4], Girish Kale[2,3,4], Maily Scorcelletti[2], Michiko Takeda[1], Yu-Chiun Wang[1✉] & Steffen Lemke[2,3✉]

Metazoan development proceeds through a series of morphogenetic events that sculpt body plans and organ structures[1,2]. In the early embryo, these processes occur concurrently such that forces generated in neighbouring tissues can impose mechanical stresses on each other[3–5], potentially disrupting development and consequently decreasing fitness. How organisms evolved mechanisms to mitigate inter-tissue mechanical conflicts remains unclear. Here, we combined phylogenetic survey, quantitative live imaging and functional mechanical perturbation to investigate mechanical stress management during gastrulation across the insect order of flies (Diptera). We identify two distinct cellular mechanisms that prevent tissue collision between the expanding head and trunk. In Cyclorrhapha, a monophyletic subgroup including *Drosophila melanogaster*, active out-of-plane deformation of a transient epithelial fold, called the cephalic furrow, acts as a mechanical sink to pre-empt head–trunk collision. Genetic and optogenetic ablation of the cephalic furrow leads to accumulation of compressive stress, tissue buckling at the head–trunk boundary and late-stage embryonic defects in the head and nervous system. By contrast, the non-cyclorrhaphan *Chironomus riparius* lacks cephalic furrow formation and instead undergoes widespread out-of-plane division that reduces the duration and spatial extent of head expansion. Re-orienting head mitosis from in-plane to out-of-plane in *Drosophila* partially suppresses tissue buckling, showing that it can function as an alternative mechanical sink. Our data suggest that mechanisms of mechanical stress management emerge and diverge in response to inter-tissue conflicts during early embryonic development.

The development of multicellular organisms proceeds through a series of morphogenetic events that sculpt tissue morphology. Morphogenesis in animals starts during gastrulation, when simple cell clusters or sheets are transformed into complex tissues with several layers and curved shapes. In the early embryo, several morphogenetic events typically occur simultaneously in a mechanical continuum that lacks clear compartmentalization, raising the question of how forces emanating from one process may influence another. Although forces exerted externally have been shown to be co-opted to facilitate local deformation[4–9], in cases where inter-tissue conflicts lead to the accumulation of mechanical stresses, it is unclear whether specific mechanisms evolved to mitigate potentially detrimental effects. Earlier studies are rare given the need for a well-resolved phylogeny, broad sampling and documentation from representative clades, and functional assessment through genetic and mechanical perturbation.

Gastrulation in the fruit fly *Drosophila melanogaster* (Drosophilidae) begins with concurrent morphogenetic movement of its three germ layers[10,11]. At gastrulation onset, mesoderm internalizes through ventral furrow formation, endoderm internalization begins with posterior midgut (PMG) invagination and at the boundary between the head and the trunk, ectoderm forms a deep epithelial fold called the cephalic furrow (CF)[12–14]. These morphogenetic movements are followed by germband extension (GBE) in which the trunk ectoderm undergoes convergent-extension to elongate its anterior–posterior axis[15]. Coincident with GBE, mitosis occurs in a spatially stereotypical and temporally ordered manner in locally semi-synchronous domains called the mitotic domains (MD), with four of five located in the head ectoderm[16]. The first 30 min of *D. melanogaster* gastrulation is thus characterized by temporally overlapping morphogenetic events. Among these, the CF stands out as a transient epithelial fold that forms and retracts back to the embryonic surface, giving rise to no internal cell type or tissue structure[13]. CF positioning is both precise and robust[12,17], and yet its function remains unknown. One proposal is that the CF functions as an anterior barrier to guide the long-range, posterior-directed flow of GBE, given that the CF is formed at the head–trunk boundary[18]. This would predict that the CF is unique to insects that undergo GBE. A contrasting hypothesis posits deep conservation between the CF and the vertebrate head–trunk boundary on the basis of homologous

[1]RIKEN Center for Biosystems Dynamics Research, Kobe, Japan. [2]Centre for Organismal Studies Heidelberg, Heidelberg University, Heidelberg, Germany. [3]Institute of Biology, Department of Zoology, University of Hohenheim, Stuttgart, Germany. [4]These authors contributed equally: Bipasha Dey, Verena Kaul, Girish Kale. ✉e-mail: yu-chiun.wang@riken.jp; steffen.lemke@uni-hohenheim.de

gene expression[19]. Neither suggestion has been put to rigorous tests, however.

## CF is a morphogenetic innovation

To study CF evolution, we combined sampling and imaging of phylogenetically informative species with a survey of published literature covering the entire dipteran phylogeny[20] (Fig. 1a–h and Extended Data Figs. 1 and 2). We found that CF is present only in cyclorrhaphan and not in non-cyclorrhaphan flies. This is in contrast to GBE, which is conserved across all dipteran species thus far examined. Therefore, the CF seems to be an evolutionary novelty and a synapomorphic trait of Cyclorrhapha.

CF formation in *D. melanogaster* requires overlapping expression of the transcription factors Buttonhead (Btd) and Even-skipped (Eve) (Fig. 1i), which combinatorially specify the CF initiating cells[12,13]. Similarly, *Megaselia abdita* (Phoridae) forms the CF (Fig. 1d) and expresses *btd* and *eve* in an overlapping pattern (Fig. 1j), suggesting the cyclorrhaphan stem group as the most recent possible origin of CF formation (Fig. 1a). By contrast, the non-cyclorrhaphan midge *Chironomus riparius* (Chironomidae), expresses its *btd* and *eve* orthologues (*Cri-btd* and *Cri-eve*) (Extended Data Fig. 3a) in non-overlapping regions (Fig. 1k,l and Supplementary Note 1), with a gap of one or two nuclei. The absence of an overlap between *btd* and *eve* expression has also been reported in the companion study[21] in two other non-cyclorrhaphan flies, *Clogmia albipunctata* (Psychodidae) and *Anopheles stephensi* (Culicidae). Together, these data suggest that non-cyclorrhaphan flies lack the positional code necessary for CF initiation.

A direct comparison of gastrulation shows unequivocally that *C. riparius* lacks CF formation. In *D. melanogaster*, the CF appears at ~33% embryo length, concurrent with PMG invagination[12], and persists for about 90 min before full retraction. Live imaging of *C. riparius* embryos shows no infoldings at the head–trunk boundary, from the blastoderm stage to the end of GBE (Extended Data Fig. 3b,c). At the cellular scale, CF onset is characterized by a decrease in apical cell surface area in one or two columns of initiating cells in *D. melanogaster*[12,17] (Fig. 1m; colour-coded in the region between ~28% and 39% embryo length in Extended Data Fig. 3d). In *C. riparius*, no such apical surface area decrease was observed between ~28% and 52% embryo length (Fig. 1n and Extended Data Fig. 3e), and there was no cell internalization, in contrast to *D. melanogaster* in which about five columns of cells from the flanking head and trunk ectoderm become incorporated into the CF (Extended Data Fig. 3b,c and Supplementary Video 1). In summary, the CF is a morphogenetic innovation originating in the cyclorrhaphan stem group, concomitant with a gain of overlapping expression between *btd* and the first stripe of *eve* (*eve1*). By contrast, genetic and cellular processes underlying the CF are absent in *C. riparius* and presumably in all other non-cyclorrhaphan flies.

## CF loss causes head–trunk buckling

To explain how the CF evolved, we analysed its role during development. Because both *btd* and *eve* are expressed in many non-CF cells and their mutants cause substantial developmental defects outside the head–trunk boundary (see FlyBase links in Methods for phenotypic data associated with *btd^{AX}* (ref. 22) and *eve^{R13}* (ref. 23)), we engineered flies that specifically lack *eve1* expression to block CF formation with minimal perturbation of genetic patterning elsewhere. We introduced a full-length *eve* genomic construct that lacks enhancer elements conferring *eve1* expression[24,25] into an *eve* null genetic background to yield the *eve1^{KO}* line and confirmed its lack of *eve1* expression (Extended Data Fig. 4a,b and Methods). At gastrulation onset, these embryos lack planar polarized non-muscle Myosin-II (MyoII) accumulation in cells predicted to initiate the CF[12,26] and do not form the CF (Fig. 2a,b and Supplementary Video 2; but see Extended Data Fig. 4c,d). Although lacking the CF,

the epithelium undergoes out-of-plane deformation at the head–trunk boundary at a later time point (Fig. 2c,d and Supplementary Video 2). We referred to such deformation as 'head–trunk buckling' because it differs from the CF in several ways that suggest it results from passive mechanical instabilities, not from genetically patterned active deformation. First, as mentioned above, the deformation is not associated with spatially confined, planar polarized Myosin enrichment. Second, unlike the CF initiating cells that shorten gradually, cells that initiate buckling move inwards abruptly, creating a broad indentation rather than a narrow cleft as seen in the CF (Fig. 2a,b and Supplementary Video 2). Third, head–trunk buckling occurs ~9.4 min after the onset of PMG invagination, in contrast to CF initiation that is concurrent with PMG invagination (Fig. 2g and Supplementary Video 2). Lastly, the initiating position of buckling varies along the dorso-ventral axis (Extended Data Fig. 4e), differing from the CF, which typically starts laterally and spreads dorsally and ventrally[12,26]. Of note, similar buckling also occurs in classic *eve* and *btd* mutants (Fig. 2e–g and Supplementary Video 3), as shown in previous and current reports[12,13,21], permitting the use of *eve* or *btd* mutants, or global RNA interference (RNAi) knockdown to induce head–trunk buckling. In addition to the head–trunk buckling, we observed buckling-like deformations elsewhere in the head region (Extended Data Fig. 5d), consistent with data reported in ref. 21. These deformations are more variable and occur at lower frequencies, contrasting with the fully penetrant head–trunk buckling. Although all observed bucklings are probably manifestations of mechanical instabilities related to the accumulation of compressive stress, we focused primarily on head–trunk buckling in the following analysis to show the functional role and evolutionary origin of the CF.

To fully rule out that the observed head–trunk buckling is related to genetic perturbation, we mechanically blocked CF formation using the optogenetic Opto-DNRho1 system that locally inhibits actomyosin contractility[12,27]. We precisely illuminated only the CF region (Methods) to avoid perturbing contractility elsewhere. This completely eliminated CF formation on one side of the embryo, and produced head–trunk buckling similar to that in the *eve1^{KO}* embryo (Fig. 2g–i and Supplementary Video 4). These data confirm that, in the absence of CF, head–trunk buckling occurs as a result of neither increased actomyosin contractility nor local genetic perturbation. Thus, both genetic and mechanical blockage of the CF results in passive buckling that arises from mechanical instabilities, suggesting that, without the CF, compressive stress accumulates at the head–trunk boundary.

How compressive stress might arise is suggested by the following two observations. First, buckling typically starts when the second MD cells (MD2) undergo mitotic rounding, suggestive of a link to mitosis (Extended Data Fig. 5a and Supplementary Video 4; note that in wild-type embryos MD2 cells round up well after CF initiation). Second, as buckling deepens, the anterior trunk ectoderm moves inwards (Supplementary Video 4), suggesting the involvement of GBE. We thus proposed that head–trunk buckling is related to compressive stress generated during head expansion through mitosis and trunk expansion through convergent-extension during GBE.

## Head and trunk tissue flows collide

To understand the build-up of compressive stress at the head–trunk boundary, we characterized tissue flows using particle image velocimetry (Extended Data Fig. 5b and Methods). In wild-type embryos, local, convergent flows occur at the head–trunk boundary where the CF is formed, suggesting that it behaves as a tissue sink with head and trunk tissues flowing into it (Fig. 2j, control at gastrulation onset), and indeed, flanking head and trunk cells become incorporated into the CF (Extended Data Fig. 3b and Supplementary Video 1, top panel). As gastrulation progresses, two persistent, regionally coherent flows can be observed flanking the CF: a posterior-wards flow in the head and a ventral-wards flow in the trunk that diverges along the

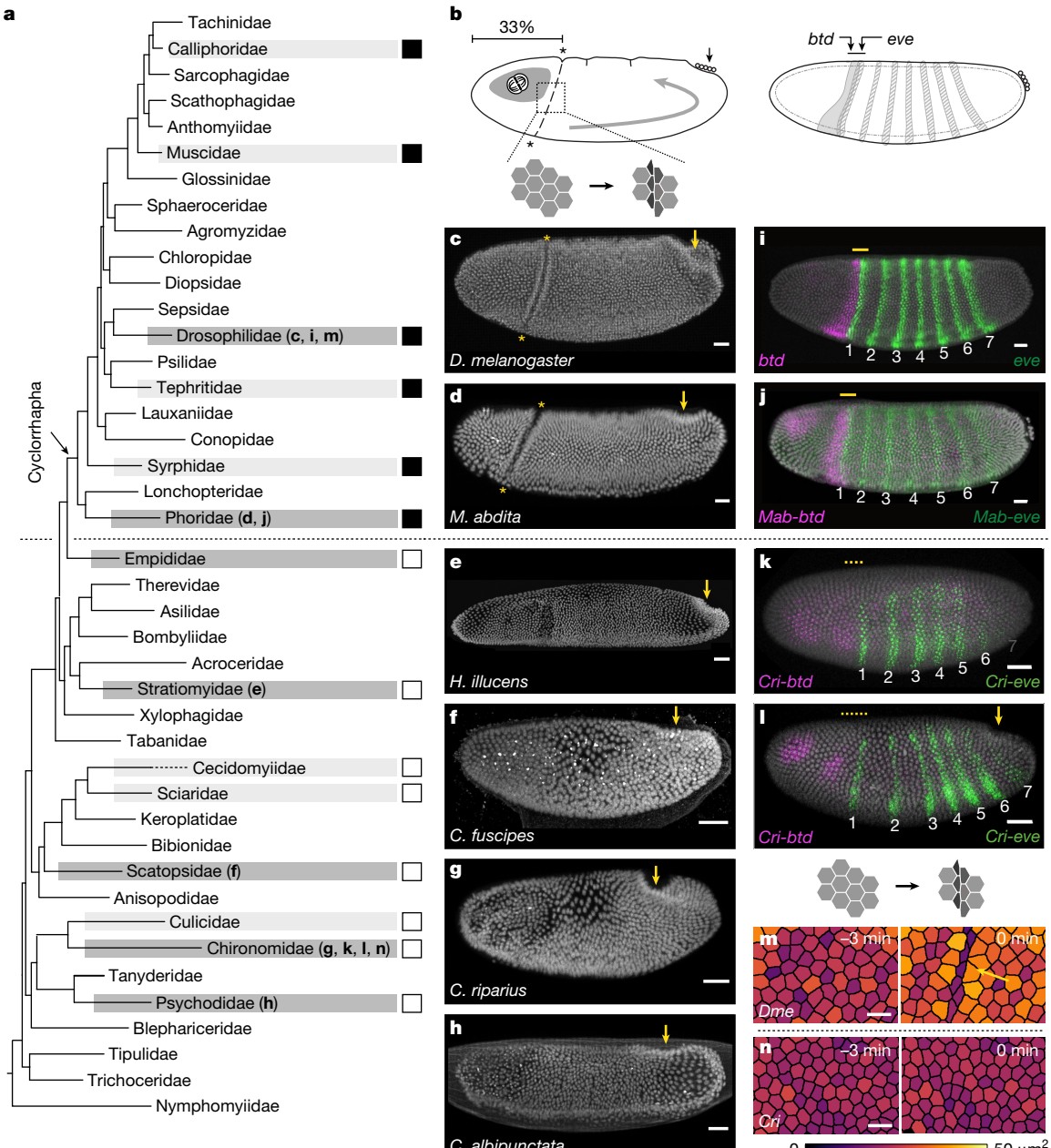

**Fig. 1 | The CF is an evolutionary innovation at the head–trunk interface of flies. a**, Dipteran phylogeny showing main fly families[20], marking CF presence or absence (filled or empty boxes), indicating CF emergence in the stem group of Cyclorrhapha (arrow). Families with evaluated (light shading) and studied (dark shading) species. Letters in parentheses correspond to the panels on the right (see also Extended Data Fig. 1). **b**, Left, schematics summarizing the spatio-temporal characteristics of CF formation relative to other morphogenetic events (dotted line, head–trunk boundary; grey patch in the head, mitotic activity; curved arrow, GBE; black arrow, PMG; inset, cellular or supracellular morphogenetic changes; or right, genetic patterning (*btd* and *eve* expression overlap). **c**–**h**, Representative embryos from selected species, fixed at comparable stages using nuclear staining (DRAQ5): *D. melanogaster* (*n* = 13) (**c**), *M. abdita* (*n* = 30) (**d**), *Hermetia illucens* (Stratiomyidae) (*n* = 57) (**e**), *Coboldia fuscipes* (Scatopsidae) (*n* = 24) (**f**),

*C. riparius* (*n* = 24) (**g**) and *C. albipunctata* (*n* = 13) (**h**). Arrows, PMG; asterisks, CF. **i**–**l**, Maximum projections of Btd (magenta) and Eve (green) expression patterns by means of immunofluorescence in *D. melanogaster* (*n* = 3) (**i**), fluorescent in situ hybridization in *M. abdita* (*n* = 3) (**j**) and HCR-based in situ hybridization in *C. riparius* in embryos before (*n* = 10) (**k**) or after (*n* = 10) (**l**) onset of gastrulation (Supplementary Note 1), with nucleus (grey) labelled using DAPI (**i**,**j**) or DRAQ5 (**k**,**l**). Partial overlap of expression patterns or lack thereof is shown by solid or dashed yellow lines, respectively. Numbers denote *eve* stripes. **m**,**n**, Mesoscopic view of head–trunk boundary region in *D. melanogaster* (*n* = 6) (**m**) and *C. riparius* (*n* = 5) (**n**) embryos before (*t* = −3 min, left) and at gastrulation onset (right). Arrow, apical cell area reduction; LUT bar, cell area the colour-code. Scale bars, 25 μm (**c**,**d**,**f**–**l**), 50 μm (**e**) and 10 μm (**m**,**n**).

anterior–posterior axis (Fig. 2j, control at MD1 telophase onset). This suggests that without a sink, the head and trunk tissues would have collided at the head–trunk interface. By contrast, the *eve1KO* embryo has a single, uninterrupted flow field at the onset of gastrulation, showing the absence of a sink and confirming a previous in silico prediction[18] (Fig. 2j, *eve1KO* at gastrulation onset). However, local convergent flows emerge

later at the head–trunk boundary during buckling, similar to CF onset in the wild type, suggesting that head–trunk buckling also behaves as a sink (Fig. 2j, *eve1KO* at MD1 telophase onset) and breaks the continuity of the flow field. Of note, similar flow fields were also observed in embryos that lack Btd expression (Extended Data Fig. 5c). In sum, we propose that without the CF as a genetically patterned mechanical

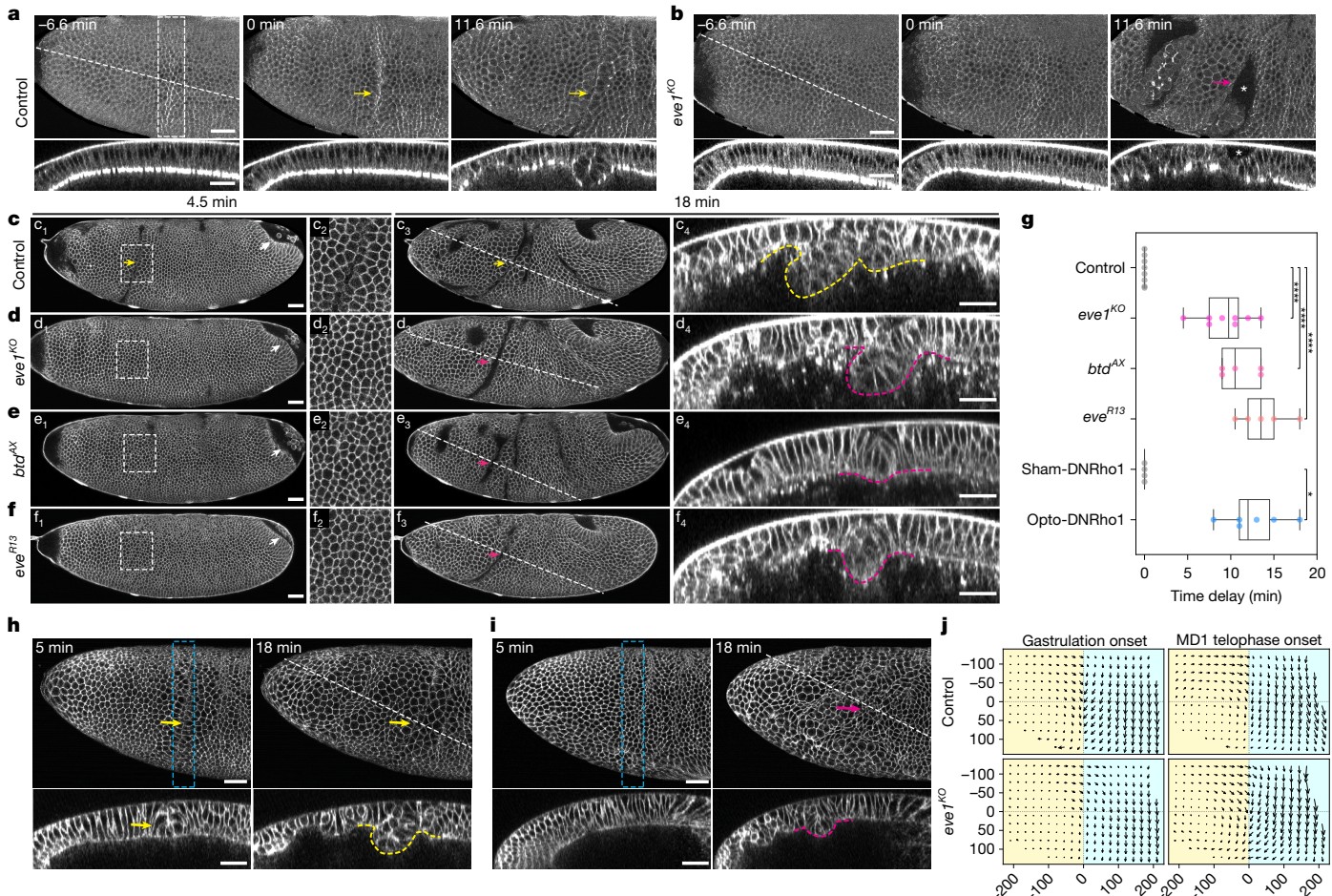

**Fig. 2 | Genetic or optogenetic blockage of CF results in tissue buckling.**
**a,b**, Time-lapse of control (*eve1*^KO/+^, *n* = 3) (**a**) or *eve1*^KO^ (*n* = 4) (**b**) embryos
expressing MyoII-mKate2. Lateral surface projections (top) of the anterior
region and corresponding z-reslices (bottom). Dashed rectangle, planar
polarized MyoII accumulation; asterisks, surface indentation of head–trunk
buckling. **c–f**, Time-lapse of control (*Gap43-mCherry*/+, *n* = 6) (**c**), *eve1*^KO^
(*n* = 3) (**d**), *btd*^AX^ (*n* = 5) (**e**) or *eve*^R13^ (*n* = 6) (**f**) embryos, showing lateral surface
projections ($c_1$–$f_1$ and $c_3$–$f_3$), enlarged CF region views ($c_2$–$f_2$, boxed areas in
$c_1$–$f_1$) and z-reslices ($c_4$–$f_4$). **g**, Timing of CF onset or head–trunk buckling relative
to gastrulation onset. *n* = 6 (control), *n* = 8 (*eve1*^KO^), *n* = 5 (*btd*^AX^), *n* = 5 (*eve*^R13^),
*n* = 4 (Sham-DNRho1) and *n* = 6 (Opto-DNRho1). One-way analysis of variance
(ANOVA) Tukey's post hoc test. ****$P$ < 0.0001 (*eve1*^KO^, $P$ = 0.00000017; *btd*^AX^,
$P$ = 0.00000009; *eve*^R13^, $P$ = 0.000000002 compared with control). Mann–
Whitney $U$-test (two-sided), *$P$ = 0.011. Each dot represents an embryo,

horizontal lines are the median, boxes show the interquartile range and
whiskers mark the 95% confidence interval. **h,i**, Time-lapse of sham
(Sham-DNRho1, *n* = 4) (**h**) and photo-activated (Opto-DNRho1, *n* = 7) (**i**) embryos
expressing the Opto-DNRho1 system, showing lateral surface projections
(top) of the anterior region and corresponding z-reslices (bottom). Blue
dashed rectangles, illumination ROIs. **j**, Tissue flow fields using particle
image velocimetry in control (*3xmScarlet-CaaX*/+, *n* = 3) and *eve1*^KO^ (*n* = 3) at
gastrulation onset and MD1 telophase. Yellow shading indicates the head
region and blue shading the trunk region; x-origin, head–trunk boundary;
y-origin, lateral midline. The units are pixels. Membranes were visualized using
Gap43-mCherry (**c–f**) or 3xmScarlet-CaaX (**h,i**). Dashed lines mark z-reslice
positions, cropped to highlight the CF. White arrows, PMG; yellow arrows and
dashed outlines, CF; magenta arrows and dashed outlines, head–trunk
buckling. Time is shown relative to gastrulation onset. Scale bars, 30 μm.

sink, converging flows of the expanding head and trunk 'tissue plates'
generate compressive stress and cause head–trunk buckling through
'tissue tectonic collision'. We note that 'tissue tectonics' was previously
introduced to describe in-plane epithelial deformation by means of
cell-shape changes and re-arrangements[28] and developmental tim-
ing regulation following the displacement of signalling and respond-
ing tissues[29]. We expand this term to describe epithelial out-of-plane
deformation resulting from tissue collision, drawing on the analogy to
Earth's tectonic plates, which collide at their convergent boundaries
to form mountain ranges and deep-sea trenches.

## Head and trunk collision causes buckling

To further test the hypothesis of tissue collision, we asked whether
reducing head and trunk expansion dampens buckling. To abrogate
head expansion, we blocked mitosis by removing the zygotic activity

of *string* (*stg*), which encodes the *Drosophila* homolog of Cdc25 and
drives mitosis in each MD[30,31]. In *stg* mutants, CF initiation occurs
normally, indicating that mitosis is not required for CF formation
(Extended Data Fig. 6a and Supplementary Video 5). We then knocked
down *btd* in *stg* mutants and observed either late-onset buckling with
reduced depth and persistence (class I, *stg*(I)), or a complete lack of
buckling (class II, *stg*(II)) (Fig. 3a,b,d,e, Extended Data Fig. 6b,e and
Supplementary Videos 5 and 6), which confirms the findings in the
companion study[21] (Supplementary Note 2). Thus, cell division fuels
head expansion, driving its posterior-wards flow to collide with the
trunk and cause buckling.

In the trunk, tissue flow stems from the active forces generated dur-
ing ventral furrow formation and GBE[18]. Ventral furrow formation drives
ventral-wards flow[18], whereas GBE generates anterior–posterior diver-
gent flow[5] (Fig. 2j). To examine whether reduced anterior–posterior
divergent flow dampens buckling, we used the quadruple mutants

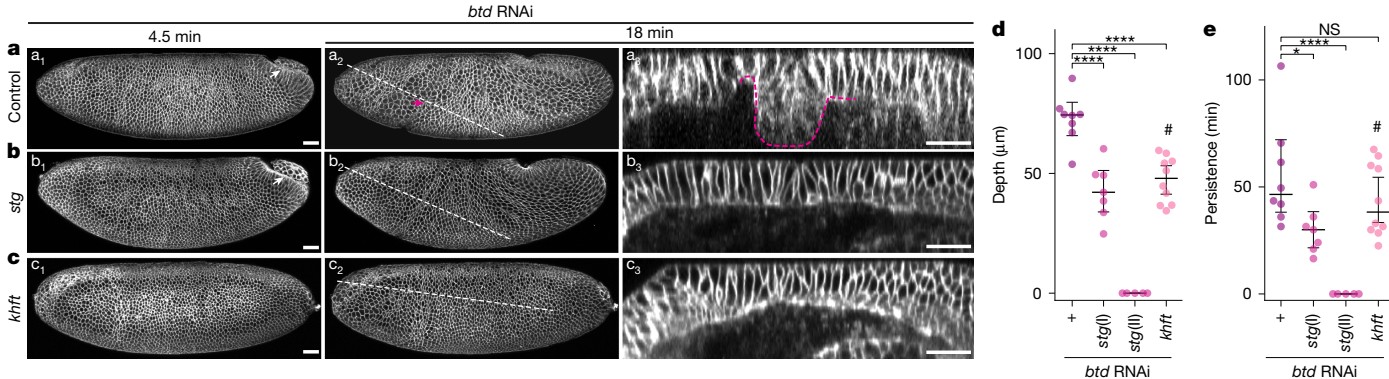

**Fig. 3 | The CF functions as a mechanical sink to prevent tissue collision.**
**a**–**c**, Time-lapse of *btd* RNAi embryos in control (**a**), *stg* (**b**) or *khft* (**c**) genetic background, visualized with 3xmScarlet-CaaX showing lateral surface projections ($a_1$–$c_1$ and $a_2$–$c_2$) and z-reslices ($a_3$–$c_3$). Sample size: control, $n = 8$; *stg*(I), $n = 7$; *stg*(II), $n = 5$; *khft*, $n = 10$. **d**,**e**, Maximum depths (**d**) and durations (**e**) of head–trunk buckling in *btd* RNAi embryos with extra genetic manipulations. Each dot represents an embryo, bold lines indicate the median and whiskers mark the 95% confidence interval. Two *stg* phenotypic classes (I and II) are shown. One-way ANOVA Tukey's post hoc test. ****$P < 0.0001$ (*stg*(I), $P = 0.00001$; *stg*(II), $P = 0.00000$; *khft*, $P = 0.00004$) (**d**); *$P < 0.05$ ($P = 0.04$); ****$P < 0.0001$ ($P = 0.00003$); NS, not significant, $P > 0.05$ ($P = 0.52$) (**e**). Sample size: control, $n = 8$; *stg*(I), $n = 7$; *stg*(II), $n = 5$; *khft*, $n = 10$. #, late buckling in *btd* RNAi-injected *khft* embryos (Extended Data Fig. 6); dashed lines, z-reslice positions cropped to highlight the CF; white arrows, PMG; yellow arrows and dashed outlines, CF; magenta arrows and dashed outlines, head–trunk buckling. Time relative to gastrulation onset. Scale bars, 30 μm.

*knirps hunchback forkhead* and *tailless* (*khft*)[32,33] to eliminate both local junctional exchanges and external drag by PMG invagination[5] to fully abrogate GBE (Extended Data Fig. 6a and Supplementary Video 5; note that CF initiation is normal, demonstrating that it does not require GBE). When CF initiation is inhibited using *btd* RNAi in *khft* mutants, no head–trunk buckling is observed (Fig. 3a,c–e (but note the existence of late buckling associated with mitoses), Extended Data Fig. 6b and Supplementary Videos 5 and 6). These data suggest that anterior–posterior divergent flow during trunk elongation collides with the head ectoderm to cause buckling.

Together, these data support our proposal that genetically programmed head and trunk expansion causes tissue collision and buckling when the CF is absent. In fact, despite the CF being initiated by local actomyosin contractility[12,26], its full invagination requires neighbouring tissue expansion, because both *stg* and *khft* mutants form shallower and less-persistent CF (Extended Data Fig. 6a,c,d). Thus, the CF can 'absorb' the expanding neighbouring tissues, acting as a bona fide, genetically patterned mechanical sink guiding tissue flows, thereby pre-empting tissue collision and buckling. In silico simulations in the companion study[21] further corroborate this interpretation.

## Abrogation of CF has deleterious effects

We next asked whether CF loss and head–trunk buckling have deleterious effects on embryonic development. We used Opto-DNRho1 to eliminate the CF such that the associated phenotypes could only be attributed to CF loss, and not altered genetic patterning. We precisely and bilaterally blocked CF initiation, and confirmed that gastrulation is otherwise normal, as evidenced by ventral furrow formation, which closes to form a straight ventral midline (Fig. 4a,b and Supplementary Video 7). Strikingly, following ventral furrow closure we observed an increased frequency of ventral midline distortion or rotation compared with the sham control, ~1.5 h after the onset of gastrulation (Fig. 4a–c and Supplementary Video 7). Ventral midline distortion is variable and often associated with asymmetric head–trunk buckling, suggesting that buckling releases compressive stress stochastically, whereas the programmed CF can reduce such stochasticity. Note also that we observed similar distortions in *btd* mutants and RNAi embryos (Extended Data Fig. 7a–e and Supplementary Video 7).

Although ventral midline distortion is a clear deviation from normal embryonic development in *D. melanogaster*, it is insufficient to indicate deleterious effects on embryonic development. To examine whether CF loss causes further, possibly more severe, developmental abnormalities detrimental to the function or viability of the organism, we monitored embryonic development for at least 18 h after a 1 h optogenetic inhibition of CF at gastrulation onset. We observed an increased frequency of head involution and ventral nerve cord (VNC) defects at later stages, with a substantial co-occurrence of head involution and VNC defects that are independent of ventral midline distortions, suggesting that they represent a separate phenotypic repertoire (Fig. 4d–f, Extended Data Fig. 7f,g and Supplementary Video 8). These late defects suggest that focusing on early gastrulation substantially underestimates the abnormalities associated with CF abrogation, which extend far beyond the gastrulation stage. Because head involution ensures proper internalization of embryonic head segments, including vital organs such as the mouth[34,35], and VNC condensation promotes proper assembly and wiring of the central nervous system[36], these data suggest that CF loss is deleterious to embryonic development (see Discussion and Supplementary Note 3).

## Head mitosis differs among fly species

Whereas cyclorrhaphan flies, like *D. melanogaster*, prevent tissue collision through a genetically programmed mechanical sink (the CF), non-cyclorrhaphan flies may use alternative mechanisms to dissipate compressive stress. Given that GBE is conserved (Extended Data Figs. 1 and 2), and the absence of CF does not correlate with low cell density (Extended Data Fig. 8a–g), it remains unclear how non-cyclorrhaphan flies compensate for their lack of a CF. Therefore, we asked whether early head morphogenesis differs between these groups. Using nuclei as a proxy for cells, we found that the non-cyclorrhaphan species *Hermetia illucens*, *Coboldia fuscipes*, *C. riparius* and *C. albipunctata* have a double layer of nuclei in the head epithelium during early gastrulation, in contrast to *D. melanogaster* and *M. abdita* that have a monolayer epithelium (Fig. 5a). Live imaging in a *C. riparius* embryo showed that this double-layering is due to out-of-plane division, as evidenced by the orientation of the cytokinetic rings (Fig. 5b–d and Supplementary Videos 9 and 10), rather than pseudo-stratification or cell extrusion (Extended Data Fig. 8h).

Head mitosis in the *C. riparius* embryo occurs soon after gastrulation onset and temporally overlaps with trunk expansion, comparable with *D. melanogaster* (Extended Data Fig. 9a). We categorized the MDs

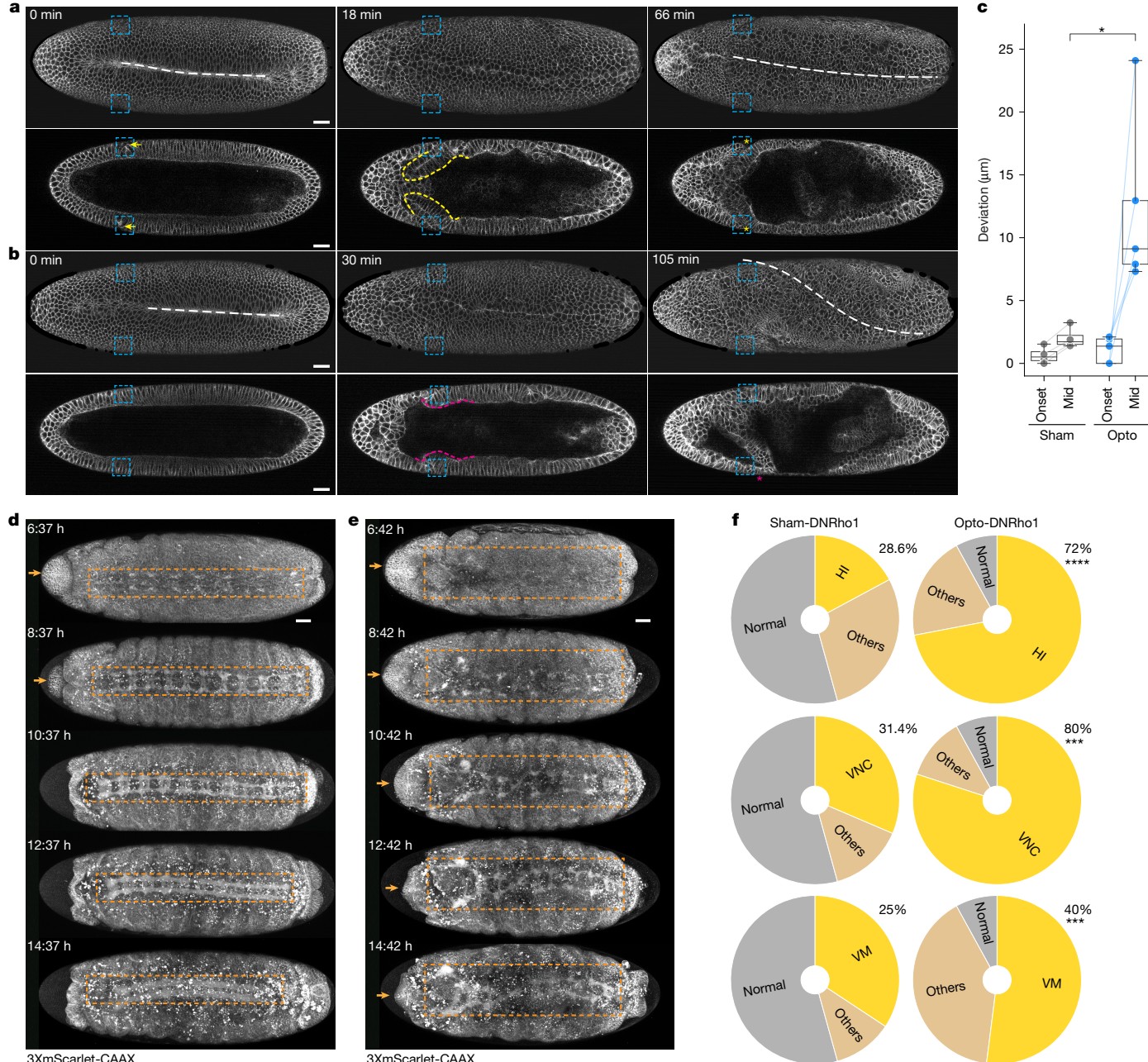

**Fig. 4 | CF loss causes midline distortion during gastrulation and late embryonic abnormalities. a,b**, Time-lapse series of a representative sham control (Sham-DNRho1, *n* = 4) (**a**) or photo-activated (Opto-DNRho1, *n* = 5) (**b**) embryo expressing the Opto-DNRho1 system, visualized with 3xmScarlet-CaaX showing a ventral surface projection (top) and a single coronal section (bottom). Blue dashed rectangles, illumination ROIs for the sham activation or photo-activation; white dashed lines, ventral midlines; yellow arrows and yellow dashed outlines, CF; yellow asterisks, bilaterally symmetric CFs; magenta dashed outlines, head–trunk bucklings; magenta asterisks, laterally asymmetric buckling. **c**, Box plot showing mean deviation of the manually marked ventral midline from the expected linear ventral midline position measured at the onset of gastrulation (onset) and a mid-gastrulation (mid)

stage when deviation reaches a maximum. Mann–Whitney *U*-test (two-sided), *$P$* < 0.05 ($P$ = 0.0159). Sample size: Sham *n* = 4, Opto *n* = 5. Bold lines indicate the median, boxes show the interquartile range and whiskers indicate the minimum–maximum range. **d,e**, Time-lapse series of a representative sham control (Sham-DNRho1, *n* = 35) (**d**) or photo-activated (Opto-DNRho1, *n* = 25) (**e**) embryo, visualized with 3xmScarlet-CaaX, showing a ventral view with maximum intensity projection. Orange arrows, head involution (HI); orange dashed rectangles, VNC condensation. **f**, Pie charts showing the percentage of embryos defective for HI, VNC or ventral midline (VM). Time is relative to the onset of gastrulation. Fisher's exact test with Bonferroni correction ($\alpha$ = 0.0167). Head involution, ****$P$ < 0.0001 ($P$ = 0.00002); VNC, ***$P$ < 0.001 ($P$ = 0.0002); ventral midline, ***$P$ < 0.001 ($P$ = 0.0003). Scale bars, 30 μm.

chronologically, similarly to the naming convention established in *D. melanogaster*[16] (Fig. 5b and Supplementary Note 4) and focused on division orientation in MD2, a large domain immediately anterior to the head–trunk boundary. Of the cells whose division orientation can be determined unambiguously, about 50% divide out-of-plane (Extended Data Fig. 9b), which we termed MD2o (for 'out-of-plane'). These cells

are located in a distinct anterior subdomain that has an ellipsoid shape (Fig. 5c). By contrast, MD2i (for 'in-plane') is posterior and ventral to MD2o and has a crescent shape. Divisions begin in the centre of MD2o and spread out as a concentric wave travelling across the remainder of MD2o and subsequently across MD2i (Fig. 5c, Extended Data Fig. 9c and Supplementary Video 9). Thus, head mitosis in *C. riparius* differs

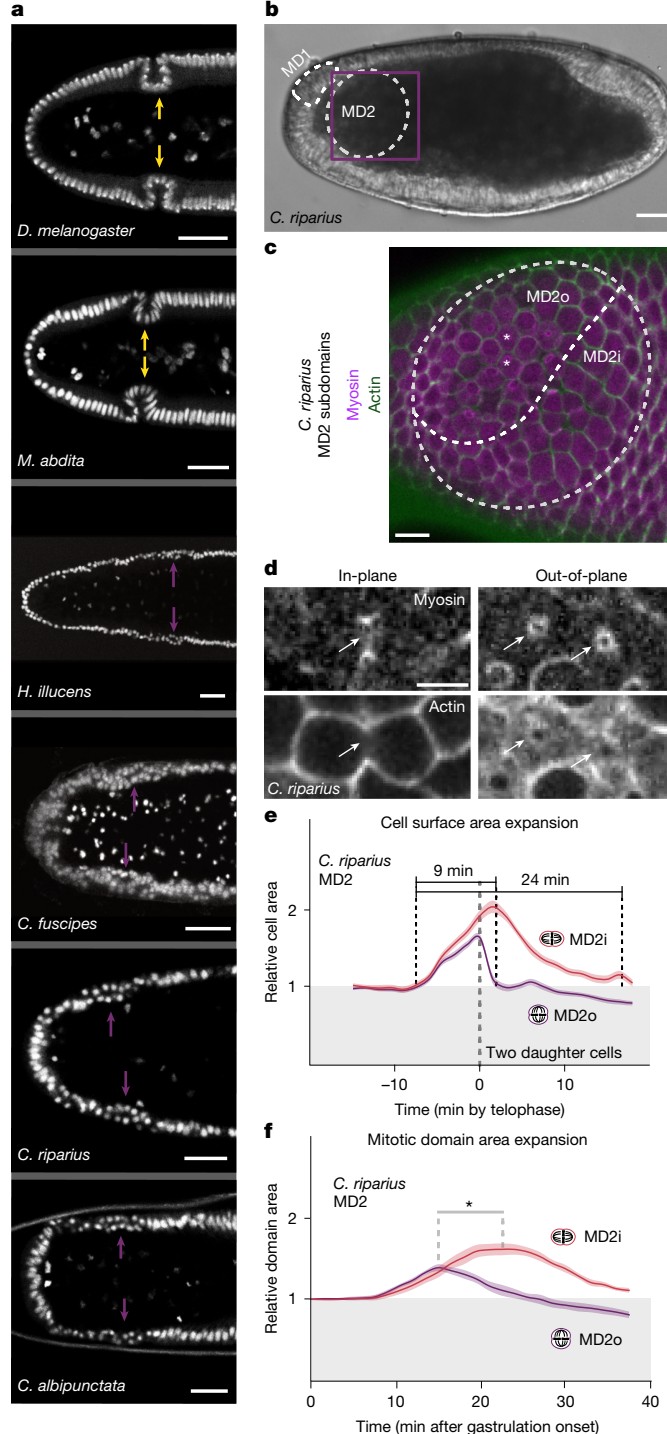

**a**

*D. melanogaster*

*M. abdita*

*H. illucens*

*C. fuscipes*

*C. riparius*

*C. albipunctata*

**b**

MD1

MD2

*C. riparius*

**c**

*C. riparius*
MD2 subdomains
Myosin
Actin

MD2o

MD2i

**d**

In-plane

Out-of-plane

Myosin

Actin

*C. riparius*

**e**

Cell surface area expansion

*C. riparius*
MD2

9 min

24 min

MD2i

MD2o

Two daughter cells

Relative cell area

Time (min by telophase)

**f**

Mitotic domain area expansion

*C. riparius*
MD2

*

MD2i

MD2o

Relative domain area

Time (min after gastrulation onset)

**Fig. 5 | MD cells in non-cyclorrhaphan flies divide out-of-plane to reduce head expansion. a**, Presence of the CF (yellow arrows) or out-of-plane division (magenta arrows) as shown with nuclear staining (DRAQ5) in the anterior half of *D. melanogaster* (*n* = 13), *M. abdita* (*n* = 30), *H. illucens* (*n* = 57), *C. fuscipes* (*n* = 24), *C. riparius* (*n* = 24) and *C. albipunctata* (*n* = 13). **b**, Schematic representation of MD1 and MD2 in the *C. riparius* embryo head domain. **c,d**, Sum projection of MD2 showing MD2o and MD2i at tissue scale (**c**, purple box in **b**) and cellular views during telophase (**d**). Asterisks indicate the first dividing cells. Cell outlines, LifeAct-mCherry; cytokinetic rings, Sqh–eGFP; arrows, cytokinesis with cytokinetic rings perpendicular (in-plane) or parallel (out-of-plane) to the apical surface. *n* = 6 embryos. **e,f**, Apical cell surface area (**e**, *n* = 18 and 24 cells from 3 embryos) or subdomain surface area (**f**, paired two-sided *t*-test on maxima; *P* = 0.0145; *n* = 5) comparing MD2o and MD2i as a function of time, normalized to surface area at the blastoderm stage. Bold lines indicate means and the shaded region indicates standard errors. Note in **e**, the summed areas of two daughter cells were plotted for in-plane divisions after completion of division, whereas only the apical daughter cell was plotted for out-of-plane divisions. Scale bars, 25 μm (*C. fuscipes*, *C. riparius* and *C. albipunctata*) and 50 μm (*D. melanogaster*, *M. abdita* and *H. illucens*) (**a,b**); 10 μm (**c**); and 5 μm (**d**).

substantially from *D. melanogaster*, where MD1, MD2 and MD5—the three MDs spatio-temporally equivalent to *C. riparius* MD2—divide in-plane, whereas out-of-plane division occurs only in the late MD9 (ref. 16).

## Out-of-plane division reduces expansion

We proposed that out-of-plane division in *C. riparius* attenuates head expansion and tested it by quantifying apical surface expansion during mitotic rounding in MD2. The in-plane dividing cells increase their apical area on average to ~1.9-fold before telophase onset, and the two daughter cells have a combined area of ~2-fold after cytokinesis, before shrinking back to occupy a surface area identical to that of the mother

cell. The total duration of expansion is ~24 min (Fig. 5e and Extended Data Fig. 9e). By contrast, the out-of-plane dividing cells expand only to ~1.6-fold for ~9 min, following which the occupied surface area decreases rapidly because only one daughter cell remains on the surface and ultimately shows a net decrease (~0.8-fold of the mother cell) (Fig. 5e and Extended Data Fig. 9f). Thus, out-of-plane divisions require less surface area and do so for a shorter time, suggesting that they exert less expansile stress on the neighbouring cells.

Given that cell divisions in each domain are not fully synchronous (Extended Data Fig. 9c), we also measured tissue-level surface expansion, comparing MD2o with MD2i (~20 cells each). MD2o expands for ~12 min to ~1.4-fold, whereas MD2i expands for ~25 min to ~1.6-fold (Fig. 5f), indicating that the attenuating effect of out-of-plane division can also be seen at the tissue level. Thus, division orientation modulates accrued compressive stress, suggesting that non-cyclorrhaphan flies use out-of-plane divisions as an alternative mechanism to mitigate tissue collision during ectodermal expansion.

## Re-orienting mitosis mitigates buckling

Out-of-plane division in *D. melanogaster* in MD9 requires expression of the mitotic spindle anchoring protein Inscuteable (Insc)[37,38]. We found that *C. riparius insc* is expressed throughout the entire head region (Extended Data Fig. 9d), suggesting that the *C. riparius* head is genetically conducive to out-of-plane division. However, we were not able to directly assay its functional role in division orientation (Supplementary Note 5).

Leveraging the fact that Insc is necessary and sufficient to instruct out-of-plane division in *D. melanogaster*[38], we overexpressed Insc (Insc[OE]) throughout the *D. melanogaster* head (Methods) to re-orient in-plane division to out-of-plane. We confirmed the re-oriented division plane with the cytokinetic ring morphology (Fig. 6a,b and Supplementary Video 11) and observed reduced MD surface area expansion (from 2-fold to 1.7-fold in MD1 and from 2.8-fold to 2-fold in MD5) (Fig. 6c,d), which mirrors the difference between *C. riparius* MD2o and MD2i. This suggests that Insc[OE] effectively converts the *D. melanogaster* head into a *C. riparius*-like state, allowing us to ask whether widespread out-of-plane divisions can function as an alternative mechanical sink to release compressive stress.

To address this, we examined the effect of Insc[OE] on head–trunk buckling in *btd* RNAi embryos. We found that 54% (class I: 7 of 13) of embryos showed a complete loss of head–trunk buckling, suggesting that out-of-plane division can function as a mechanical sink in place of the CF. The remaining 46% (class II: 6 of 13) of embryos undergo

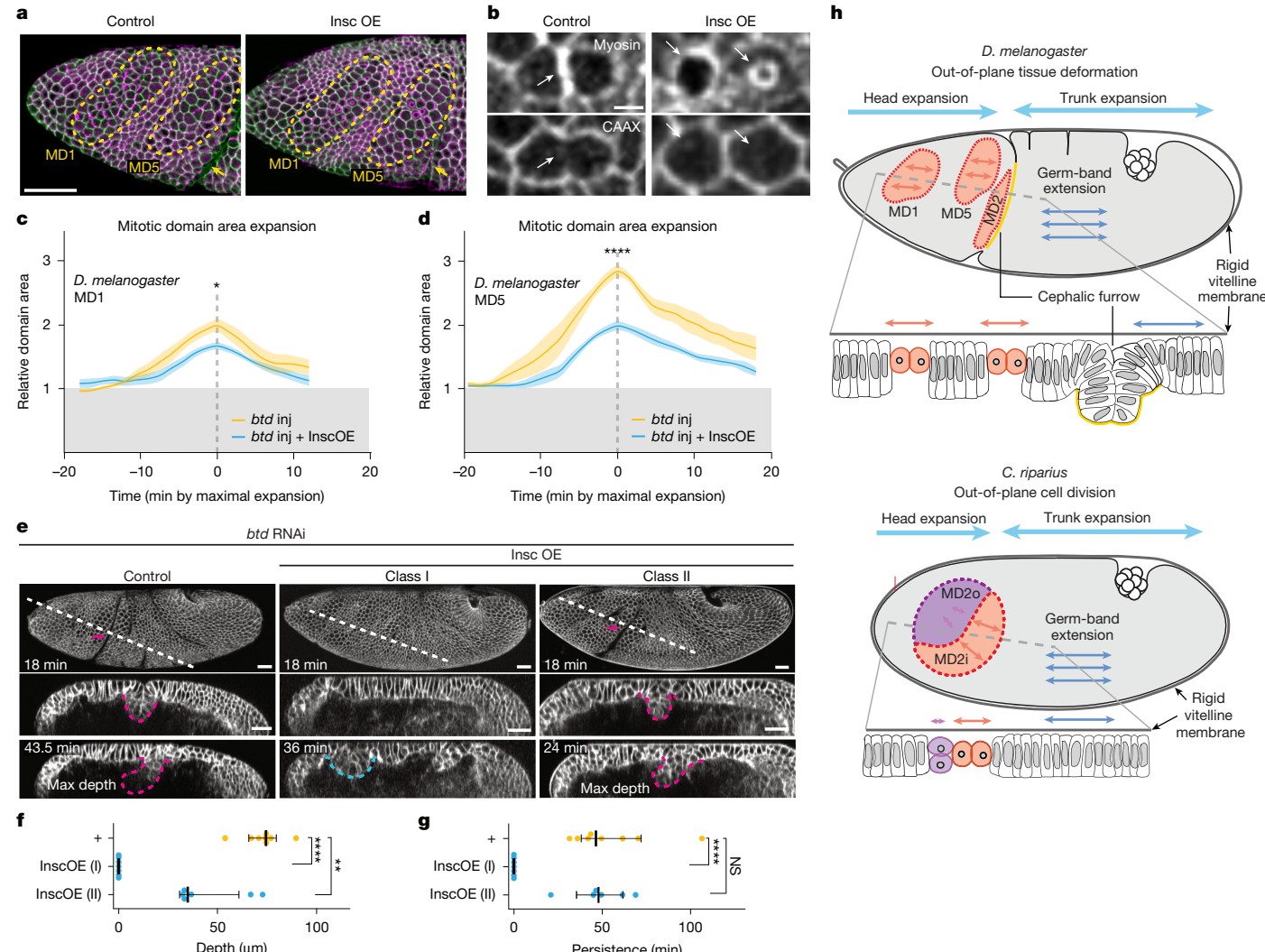

**Fig. 6 | Out-of-plane division acts as an alternative mechanical sink to mitigate tissue collision. a,b,** Lateral surface projection of the *D. melanogaster* head region showing MD1 and MD5 (**a**, yellow dashed outlines), and cellular view of telophase cells in MD5 (**b**) in control or InscOE embryo, visualized with membrane (3xmScarlet-CaaX, green) and MyoII (Sqh–eGFP, magenta) markers (*n* = 5). Yellow arrows, CF; white arrows, cytokinetic rings. **c,d,** Apical surface area of MD1 (**c**) and MD5 (**d**) in control or InscOE embryos injected with *btd* RNAi (*n* = 5 and 6, respectively) as a function of time, normalized to surface area at gastrulation onset. Bold lines indicate means and the shaded region indicates standard errors. Two-sided *t*-test on maxima. *P = 0.0166, ****P < 0.0001. **e,** Lateral surface projection (top) and *z*-axis reslice (bottom) of embryos with *btd* RNAi (*n* = 8) and *btd* RNAi with InscOE (xlass I, *n* = 7; class II, *n* = 6).

Membranes visualized with 3xmScarlet-CaaX. Magenta arrows and dashed outlines indicate head–trunk buckling and the blue dashed outline indicates buckling between MD1 and MD5. **f,g,** Maximum depths (**f**) or durations (**g**) of head–trunk buckling in *btd* RNAi embryos with and without InscOE. Each dot represents an embryo, bold lines indicate the median and whiskers mark the 95% confidence interval. One-way ANOVA Tukey post hoc test. ****P < 0.0001 (P = 0.000000001), **P < 0.01 (P = 0.0013) (**f**) and ****P < 0.0001 (P = 0.00003); NS, not significant, P > 0.05 (P = 0.782) (**g**). **h,** Schematic representation of our model illustrating the divergent strategies preventing head–trunk tissue collision in cyclorrhaphan and non-cyclorrhaphan flies, based on our findings in *D. melanogaster* (top) and *C. riparius* (bottom). See Supplementary Note 6 for more details. Scale bars, 50 μm (**a**), 5 μm (**b**) and 30 μm (**e**).

shallower head–trunk buckling than in *btd* RNAi alone, often with smaller bucklings between MD1 and MD5, or form a deeper anterior dorsal fold, or both (Fig. 6e–g and Supplementary Video 12). In material science, moderate compression of a thin elastic film on a soft compliant substrate produces short-wavelength wrinkles, whereas compression beyond a critical point leads to a 'wrinkle-to-fold transition', in which all wrinkles vanish and converge to a single deep fold[39,40]. The deep head–trunk buckling could potentially result from a wrinkle-to-fold transition, whereas small buckling seen with Insc[OE] is reminiscent of short-wavelength wrinkles, suggesting that Insc[OE] reduces compressive stress to below the critical point of such a transition. Indeed, Insc[OE] also decreases CF depth in wild-type embryos (Extended Data Fig. 10), supporting our proposal that out-of-plane divisions help dissipate compressive stress. Overall, our comparative and functional studies

show that two programmed morphogenetic strategies are used to prevent tissue collision and buckling in fly gastrulation: out-of-plane deformation (the CF) in cyclorrhaphan flies, and out-of-plane division in non-cyclorrhaphan flies.

## Discussion

Our phylogenetic survey suggests that the CF is an evolutionary novelty and a derived character in the monophyletic Cyclorrhapha. Together with the companion study[21], we show that this innovation performs a mechanical function in a physically confined embryo to pre-empt tissue collision and mechanical instabilities. Accordingly, the CF is unlikely to be homologous to the vertebrate midbrain–hindbrain boundary[19], or acting as an immobile fence that breaks the symmetry

of GBE tissue flow[18]. Instead, our results indicate that the CF behaves as a mechanical tissue sink that absorbs head and trunk tissue flows to release compressive stress.

Phylogenetic mapping suggests that both traits—the accumulation of compressive stress by widespread in-plane division in the head and its release through CF formation by active out-of-plane tissue deformation—emerged concurrently in the stem group of Cyclorrhapha (Fig. 6h and Supplementary Note 6), raising the question of which originated first.

If in-plane division emerged first, organisms that acquired this trait would have been prone to buckling and midline distortion given the increased mechanical instabilities, and might have survived only under conditions favourable to stress dissipation, such as slow development at low temperatures. The emergence of a genetically programmed out-of-plane deformation, like the CF, could then have increased developmental robustness so that the flies could adapt to a wider range of environmental conditions. Accordingly, increased compressive stress resulting from in-plane division might have imposed a mechanical constraint that facilitated positive selection for CF formation. One unresolved question related to this model, however, is how and why in-plane divisions emerged in the first place.

If the CF emerged before the shift to in-plane divisions, the CF may have arisen without the initial increase in compressive stress. Instead, active CF deformation would be expected to exert pulling forces on the neighbouring tissue, making in-plane division more energetically favourable. This could pave the way for an eventual transition to obligatory in-plane division, presumably through the loss of broad *insc* expression and the gain of extra mechanisms known to promote in-plane division, for example, high cortical tension in the mitotic cells and dedicated spindle anchorage cues[41–44]. This scenario posits that the CF is secondarily co-opted to function as a mechanical sink and would require that the CF evolved neutrally or for a different function in the first place. One such alternative function of the CF, which has been proposed previously, is that of a tissue reservoir that temporarily stores cells for subsequent head development[45]. Supporting this idea, lineage tracing experiments have previously shown that the MD2 cells in the CF contribute to the pharynx and its associated larval head cells[46]. This model could also account for the severe head involution defects we observed following CF ablation. Alternatively, the wide-ranging phenotypic consequences associated with CF ablation, including head involution and VNC defects, may result from improper stress release or tissue deformation related to mechanical instabilities, and could thus support the in-plane-division-first hypothesis. Irrespective of which originated first, both hypotheses highlight the plausibility that mechanical instabilities influence morphogenetic evolution.

Buckling transitions following CF ablation add to a growing list of epithelial morphogenesis events resulting from mechanical instabilities[47]. The variability in the number and positions of buckling observed across studies[21] may reflect intrinsic stochasticity and the sensitivity of such instabilities to small perturbations. However, spatially patterned mechanical cues can bias where passive buckling occurs[48,49]. Here, the fully penetrant buckling at the head–trunk boundary points to a possible contribution of spatially patterned 'tissue tectonic collision' between expanding tissues, rather than random stochastic effects. These findings are consistent with the hypothesis that the CF emerged under selection to mitigate the rise of mechanical instabilities at the head–trunk boundary during gastrulation (Fig. 6h). However, direct evidence for adaptation is lacking without explicit fitness measurements following CF ablation (Supplementary Note 3).

CF formation in *D. melanogaster* depends on genetic patterning and mechanical self-organization[12]. Our finding that the CF is an evolutionary novelty that co-emerged with in-plane mitotic divisions in the neighbouring tissue, raises the possibility that inter-tissue mechanical conflict, or mechanical constraint in general, constitutes a mechanism of positive feedback[50] that might drive swift evolutionary transitions and the emergence of new morphogenetic traits. The mechanical function of the CF implies that similar principles could apply to other transient epithelial folds, like the dorsal folds in *Drosophila*[51], raising the possibility that mechanical stress management represents a general functional feature of transient epithelial folding across organisms and developmental contexts.

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

## Methods

### Experimental animals and embryo collection

*D. melanogaster* embryos were collected on apple juice agar plates with yeast paste at 22 °C or at the temperatures indicated below. Laboratory cultures of *M. abdita* (Sander strain) were maintained as described[52]. *M. abdita* embryos were collected on apple juice agar plates with fish food paste at 25 °C. The laboratory cultures of *C. riparius* (Bergstrom strain) were maintained as described[52]. *C. riparius* embryos were collected as freshly deposited egg packages at ambient room temperature (23–26 °C). The laboratory cultures of *C. albipunctata* were maintained as described[53,54]. Embryos were obtained after dissecting-out adult female ovarioles, followed by experimental egg activation through a hypo-osmotic shock. The laboratory cultures of *H. illucens* were established from existing cultures in the Schmidt-Ott lab (University of Chicago). To collect *H. illucens* embryos, females were decapitated to trigger egg laying at the desired time.

A transient laboratory culture of *C. fuscipes* was established from wild-caught adults found near the municipal compost plant of the city of Heidelberg. *C. fuscipes* adults prefer to lay eggs in small cavities. To collect embryos, old culture plates of *C. albipunctata* were used as they conveniently have such cavities. Before using these for *C. fuscipes* egg collection, the plates were decontaminated by freezing them at −20 °C for at least 1 week, followed by thawing overnight at room temperature.

We could not establish a laboratory culture for any of the species from the family Empididae, because we could not optimise the culture conditions. During our excursion in Pula (Croatia) we managed to catch a few adults of an *Empis* species (most probably *Empis pennipes*, referred to as *Empis* sp. throughout the text). As a result, we resorted to repeatedly catching adults during that time from the wild, and then decapitated the females to trigger egg laying.

### *Drosophila* genetics and transgenic lines

*D. melanogaster* lines used for live imaging were *MyoII–eGFP* (also known as *Spaghetti-squash–eGFP*, or *Sqh–eGFP*)[55], *Gap43-mCherry*[56], *MyoII-mKate2* (ref. 57) and *mat-tub-3xmScarlet-CaaX* (this study). The membrane imaging line *mat-tub-3xmScarlet-CaaX* was made by cloning *3xmScarlet-CaaX* into the pBabr vector containing the mat-tub promoter (gift from D. St. Johnston, Gurdon Institute, UK)[58] and the *sqh* 3′ untranslated region (UTR), followed by ΨC31 site-directed integration into the attP2 or attP40 landing sites at WellGenetics. *D. melanogaster* mutant alleles used were *eve^R13* (FlyBase ID: FBal0003885), *btd^AX* (Fly-Base ID: FBal0030657), *stg^7M53* (FlyBase ID: FBal0016176) and the quadruple mutant[33] *knirps^IID48* (FlyBase ID: FBal0005780) *hunchback^7M48* (FlyBase ID: FBal0005395) *forkhead^E200* (FlyBase ID: FBal0004007) *tailless^L10* (FlyBase ID: FBal0016889). Descriptions of phenotypes associated with *eve^R13* (http://flybase.org/reports/FBal0003885.htm) and *btd^AX* (http://flybase.org/reports/FBal0030657.htm) were obtained from FlyBase (release FB2025_03)[59]. In live imaging experiments, the mutant embryos were identified on the basis of the absence of a balancer-linked reporter construct, hb0.7-Venus-NLS, inserted on the FM7h, CyO or TM3 balancer[12].

To generate the *eve1^KO* line, an *eve* genomic rescue construct, *eve^CH322-103K22*-*mNeonGreen*, was first created using *P[acman]^CH322-103K22* (BACPAC Resources Center), a BAC construct that encompasses the entire *eve* locus, from which the stop codon of *eve* was replaced with a standard protocol[60,61] with mNeonGreen following a linker (N-ter-GSAGSAAGSGEV-C-ter). To completely eliminate *eve* expression in the Eve1 region, the stripe1 (+6.6 to +7.4 kb relative to the transcriptional start site of *eve*)[25] and late element (−6.4 to −4.8 kb)[24] enhancers were deleted from *eve^CH322-103K22*-*mNeonGreen* through homologous recombination using the following homology arm sequences: stripe1 left, GCAAGTCCGAGACAAATCCACAAATATT GTCAACTCTTTGGCTCTAATCTG; right, CCAAGGCCGCAAAGT

CAACAAGTCGGCAGCAAATTTCCCTTTGTCCGGCGA; and late element left, TTGCGTTTGAGCTACGTTACTTACATTTTTCCCACAT GAGTCGGGCATACA; right, TCGATGGGTTGGTCACAATGTGGTGGC CTCTCAACATTGCAAGGCTCTTAC. The resultant BAC construct, *eve^CH322-103K22*-*mNeonGreenΔst1ΔLE* (Extended Data Fig. 4a) was integrated into PBac{y[+]-attP-3B}VK00033 at Rainbow Transgenics, and crossed into the *eve^R13* mutant line to generate the *eve1^KO* line. Identification of *eve1^KO* embryos in live imaging experiments was performed as above, on the basis of the absence of a balancer-linked reporter construct, hb0.7-Venus-NLS, inserted on the CyO[12].

For Insc overexpression, males of *UAS-insc* were crossed to females containing two copies of *nos-GAL4-GCN4-bcd3'UTR*, which directs targeted gene expression in the head region of resultant embryos[62]. These female flies also contained transgenes for the imaging markers Sqh–eGFP and 3xmScarlet-CaaX. The flies were incubated at 22 °C for embryo collection. For Opto-DNRho1 experiments, females of *UASp-CIBN-CaaX; UASp-CRY2-Rho1[N19, Y189]*[27] (a gift from B. He, Dartmouth College, USA) were crossed to males of *matαTub-Gal4VP16^67C; matαTub-Gal4VP16*[15] double driver line that also contains the transgene for *mat-tub-3xmScarlet-CaaX* imaging marker. The resultant F1 flies were used to set up egg deposition cages that were kept at 18 °C for collection of embryos used in the experiments.

### Protein tree

Predicted protein sequences of *eve* and *btd* were used as queries to identify closely related genes in *D. melanogaster* and putative orthologues in *M. abdita* and *C. riparius* using BLAST. Protein alignments were performed in Geneious by MUSCLE alignment with standard parameters. The protein tree was assembled using Jukes–Cantor as the genetic distance model and UPGMA (unweighted pair group method with arithmetic mean) for tree building, with a bootstrap of 1,000 replicates.

### Cloning, and messenger RNA and double-stranded RNA synthesis

*Cri-btd*, *Cri-eve*, *Cri-insc*, *Cri-sqh*, *Mab-btd* and *Mab-eve* were identified from published transcriptome sequences and cloned after polymerase chain reaction amplification from complementary DNA. In vivo labelling of cell outlines and MyoII in *C. riparius* used Gap43-linker–eGFP and Cri-Sqh-linker–eGFP, which were expressed in the embryo by the injection of in vitro synthesized messenger RNAs. The Gap43-linker–eGFP fusion construct for mRNA synthesis was generated by in-frame Gibson assembly of the Gap43 encoding sequence, a short linker (GSAGSAAGSGEV), and a previously published pSP35T expression vector (pSP-Mab-bsg–eGFP) that contained a 3′-terminal eGFP[63]. Analogously, the Cri-Sqh-linker–eGFP fusion construct was generated using a full-length fragment of *Cri-sqh* amplified by polymerase chain reaction from cDNA. Nascent mRNAs were generated using SP6 polymerase, followed by capping and poly(A)-tailing with dedicated capping and poly(A) kits (CELLSCRIPT). Synthesized mRNA was dissolved in $H_2O$.

For *btd* RNAi experiments in *D. melanogaster*, double-stranded RNA was synthesized on templates that contain the T7 promoter sequence (5′-TAATACGACTCACTATAGGGTACT-3′) at each end using a MEGAscript T7 kit (Ambion); templates were amplified from 0–4 h embryonic cDNA using specific primers (5′-AGCAGATGACGACGACAACA-3; 5′-TACTCGGACTTCATGTGGCA-3). For *insc* RNAi experiments in *C. riparius*, dsRNA was synthesized as previously described[63]. The dsRNAs comprised the following gene fragments (position 1 refers to first nucleotide in the open reading frame): *btd*, position 1,487 to 1,817; *Cri-insc* (GenBank PV919477), position 466 to 1,892.

### Injections

For dsRNA injections in *D. melanogaster*, 0–1 h-old (up to stage 2) embryos were collected, dechorionated with bleach and mounted

on an agar pad. The mounted embryos were then picked up using a coverslip painted with glue (prepared by immersing bits of Scotch tape in heptane), desiccated for 10–14 min using Drierite (W. A. Hammond Drierite Co.) and covered with a mixture of Halocarbon oil 700 and 27 (Sigma-Aldrich) at a ratio of 3:1. Needles for injection were prepared from micro-capillaries (Drummond Microcaps, outer diameter 0.97 mm, inner diameter 0.7 mm) pulled with a Sutter P-97/IVF and bevelled with a Narishige pipette beveller (EG-44). Injections were performed on a Zeiss Axio Observer D1 inverted microscope using a Narishige manipulator (MO-202U) and microinjector (IM300). A volume of ~144 pl of solution with a concentration of 1.1–1.6 µg µl$^{-1}$ or 8–12 µg µl$^{-1}$ dsRNA was injected into the embryo. Embryos were kept at 25 °C after injection in a moist chamber until early to mid-cellularization, followed by live imaging.

For injections in *C. riparius*, embryos were collected, prepared and injected essentially as described previously[52]. Embryos were injected before the start of cellularization (~4 h after egg deposition), and then kept in a moist chamber until the onset of gastrulation. Throughout all procedures, embryos were kept at 25 °C (±1 °C). Owing to their small size, *C. riparius* embryos (200 µm length) were always injected into the centre of the yolk (50% of anterior–posterior axis). Embryos were injected with dsRNA typically at concentrations of 300 to 700 ng ml$^{-1}$; mRNA was injected typically at concentrations of 1.5–2.5 µg µl$^{-1}$ (Cri-Gap43–eGFP and Cri-Sqh–eGFP). LifeAct-mCherry was injected as a recombinant protein as previously described at ~4.5 mg ml$^{-1}$ (ref. 63).

## Live imaging

Live imaging of *D. melanogaster* embryos was performed using two-photon scanning microscopy with a 25× water immersion objective (numerical aperture = 1.05) on an upright Olympus FVMPE-4GDRS system (InSight DeepSee pulsed IR Dual-Line laser, Spectra Physics) or an inverted Olympus FVRS-F2SJ system (Maitai and InSight DeepSee lasers), or a Plan-Apochromat 25× oil immersion objective (numerical aperture = 0.8) on a Zeiss LSM980 inverted microscope (Chameleon laser, Coherent Int). Excitation wavelengths were 920 nm for eGFP, 950 nm for Venus and 1,040 nm (upright) or 1,100 nm (inverted) for mKate2, mCherry or mScarlet. Three imaging settings were used with the following parameters (total z depth, xy dimension of the imaging region of interest (ROI), z-step size, time interval, imaging angle or view): (1) ~80 µm, 539.5 × 185.5 µm, 2 µm, 90 s, whole-embryo lateral or ventral views; (2) ~60 µm, 253.5 × 152 µm, 1.5 µm, 50 s, head domain; (3) ~40 µm, 208.3 × 152 µm, 1 µm, 45 s, cell division in head MDs. Embryos were collected, dechorionated and mounted on coverslips or glass-bottom dishes, and immersed in 1× phosphate-buffered saline for imaging.

Live imaging of *C. riparius* embryos was performed on a Leica SP8 confocal using a 63× glycerol immersion objective (numerical aperture = 1.30). z-stacks of ~25 µm depth were acquired at a z-step size of 1 µm and 90 s time interval. All recordings were performed at 25 °C.

Time-lapse imaging to visualize GBE was performed on Nikon Eclipse-Ti microscope in differential interference contrast mode, using a 20× objective (numerical aperture = 0.8) for *D. melanogaster*, *M. abdita*, *C. riparius* and *C. albipunctata*, with 1 frame every 1 min; on a Leica SP5 DMI6000CS inverted confocal microscope in transmission illumination mode, using a 40× objective (numerical aperture = 1.1) for *C. fuscipes*, with 1 frame every 2 min; and on Zeiss Colibri upright microscope in differential interference contrast mode, using a 10× objective (numerical aperture = 0.45) for *H. illucens* and a 20× objective (numerical aperture = 0.5) for *Empis* sp., with 1 frame every 3 min. All recordings were performed at 25 °C.

## Optogenetics

The Opto-DNRho1 system[27] was used as previously reported. To prevent unwanted photo-activation, Fly crosses and cages were kept in the dark and embryos were processed, staged and mounted in a dark room with a light source covered by a light red filter (no. 182, Lee Filters). Imaging was performed on an Olympus FVMPE-RS (InSight DeepSee pulsed IR Dual-Line laser system, Spectra Physics) with a 25× (numerical aperture = 1.05) water immersion objective and excitation wavelength of 1,040 nm for the membrane marker 3xmScarlet-CaaX. The efficacy of MyoII inhibition with the Opto-DNRho1 system was first benchmarked on ventral furrow formation to confirm that it resulted in a complete blockage of apical constriction[27].

Two photo-activation protocols were used: protocol no. 1 used a 405 nm diode laser at 0.1% power (5.48 µW) and protocol no. 2 used a 458 nm diode laser at 0.5% power (27.14 µW), both scanned at 2 µs per pixel. Sham controls were performed at 0% laser power. The photo-activation ROI was illuminated for 3 s in all experiments.

Three experimental designs were used: (1) lateral imaging with unilateral photo-activation (Fig. 2h,i, Extended Data Fig. 5a and Supplementary Video 4) used protocol no. 1 on a 28.15 × 197.05 µm ROI (50 × 350 pixels) centred on 'the pre-CF domain'[64] covering the entire region of CF initiation along the dorso-ventral circumference, beginning 16–33 min before gastrulation and repeated every 90 s; (2) ventral imaging with bilateral photo-activation (Fig. 4a,b and Supplementary Video 7) used protocol no. 1 on two 33.78 × 33.78 µm ROIs (60 × 60 pixels) each covering one side of the CF, beginning 18–30 min before gastrulation and repeated every 180 s; and (3) ventral imaging with bilateral photo-activation and long-term imaging (Fig. 4d,e and Supplementary Video 8) used protocol no. 2 on two 28.8 × 28.8 µm ROIs (40 × 40 pixels) each covering one side of the CF, beginning 15–30 min before gastrulation and repeated every 180 s for 1 h, followed by time-lapse imaging at 10 or 20 min per frame for 18–23 h.

## Immunofluorescence and fixed imaging

For antibody staining, embryos were fixed by a heat–methanol method[65] and immunostained with mouse monoclonal anti-Neurotactin (1:20, BP106, Developmental Studies Hybridoma Bank, USA), rabbit polyclonal anti-Eve (1:500, gift from M. Biggin, Lawrence Berkeley National Laboratory, USA), and rat polyclonal anti-Btd (1:500, gift from E. Wieschaus, Princeton University, USA), followed by DAPI staining to visualize nuclei. Imaging was performed on a Leica SP8 system using a 20× (numerical aperture = 0.75) multi-immersion objective with oil immersion (total z depth: 60–90 µm, z-step size: 1.04 µm).

For DNA staining, embryos were fixed by heat and devitellinized as described[66], followed by staining with DRAQ5 (1:1,000 for 1 h, Thermo Fisher Scientific, catalogue number 62251). Imaging was performed on a Leica SP8 system with a 20× glycerol objective (numerical aperture = 0.75) for *D. melanogaster*, *M. abdita*, *H. illucens* and *C. albipunctata*, and a 63× glycerol objective (numerical aperture = 1.3) for *C. fuscipes* and *C. riparius*, with a z-step size of 1 µm in a z-range that covers at least half of the embryo.

For the hybridization chain reaction (HCR), embryos were fixed by heat and devitellinized as described[54], probes for *Cri-btd* and *Cri-eve* were generated using previously published software[67] (https://github.com/rwnull/insitu_probe_generator) and ordered through Sigma-Aldrich. HCR amplifiers (B1-Alexa488 for *Cri-eve*; B2-Alexa594 for *Cri-btd*) were obtained from Molecular Instruments. Devitellinized embryos were re-hydrated in a series of 1× phosphate-buffered saline with Tween (PBT) and post-fixed for 40 min with 4% paraformaldehyde in PBT on a shaker. Following PBT washes, we followed the In situ HCR v.3.0 protocol[68] for whole-mount fruit fly embryos Revision 9 (13 February 2023) from Molecular Instruments. We then stained the embryos with DRAQ5 in 5× saline-sodium citrate with Tween (1:1,000 for 1 h) and mounted the embryos in 50% glycerol in 5× saline-sodium citrate with Tween. Imaging was performed as above for DNA staining in *Chironomus riparius*.

For in situ hybridization, embryos were fixed by a heat–formaldehyde method[63]. Transcripts were detected histochemically or fluorescently as described[69], using RNA probes for *Mab-btd* (comprising

1,473 nucleotides from +1 to 1,473, with position +1 referring to first nucleotide in the open reading frame), *Mab-eve* (comprising 984 nucleotides, from position 365 to 996 of the putative coding sequence and 351 nucleotides of the 3′ UTR), and *Cri-insc* (comprising 1,427 nucleotides from 466 to 1,892) labelled with either digoxigenin or fluorescein. *M. abdita* embryos were also stained with DAPI to visualize nuclei.

## Image processing and quantification

Images were processed, assembled into figures and converted into videos using FIJI, Affinity Designer, Adobe Illustrator and HandBrake. Quantitative data were analysed and processed using Excel, or custom-made ImageJ or FIJI macros and Python scripts using Numpy, Pandas and SciPy libraries. Plots were generated in GraphPad Prism or with Python scripts using Matplotlib and Seaborn graphic libraries. Detailed descriptions of image processing and analysis procedures are provided in Supplementary Methods.

## Statistical analyses

All of the statistical details of experiments, including the number of experiments ($n$), which represents the number of embryos used unless otherwise noted, are given in the figure legends. Python scripts using SciPy library were implemented to perform one-way ANOVA followed by Tukey's multiple comparison post hoc test for comparing means from more than two groups, and Mann–Whitney $U$-test was used as a non-parametric independent test for comparing two means. Graph-Pad Prism was used: (1) to perform statistical analyses to compare the blastoderm cell densities across species, including the calculation of medians and the 95% confidence intervals on the median, and one-way ANOVA with Kruskal–Wallis non-parametric test, without correcting for multiple comparisons (uncorrected Dunn's test); and (2) to perform Fisher's exact test with Bonferroni correction for pie chart distributions. For cell and domain area analysis, Microsoft Excel was used to perform paired and unpaired $t$-tests and to plot standard errors.

## Reporting summary

Further information on research design is available in the Nature Portfolio Reporting Summary linked to this article.

## Data availability

The data supporting the findings of this study are available in the Article and its Supplementary Information. Full datasets are available at Zenodo (https://doi.org/10.5281/zenodo.15870440)[70].

## Code availability

Custom macros (Fiji) and code (Python) based on publicly available libraries used to process and analyse data are available at Zenodo (https://doi.org/10.5281/zenodo.15870440)[70].

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

**Acknowledgements** We thank the BACPAC Resources, Center at Children's Hospital Oakland Research Institute, the Developmental Studies Hybridoma Bank, the Bloomington and Kyoto *Drosophila* Stock Centers, Y. Bellaïche, M. Biggins, S. De Renzis, N. Dostatni, B. He, S. Hayashi and E. Wieschaus for sharing reagents; Kobe BioImaging Facilities and Factory (KBiiF), S. Hayashi, L. Hufnagel, A. Guse and J. Wittbrodt for support; A. Eritano for initial characterization of the *eve1*^KO^ line; L. Schütz and A. Tok for initial conceptualization and observations of division orientation in *Chironomus riparius*; L. Popp and M. Fath for cloning and initial characterization of gene expressions in *C. riparius*; A. Remmel for establishing the lab culture for *H. illucens*, for creating a transient lab culture for *C. fuscipes*, for catching *Empis* sp. adults during the excursion in Pula, Croatia, and for sharing the recordings of embryo development for these three species; members of the Hayashi, Obata and Yoo laboratories for discussions; A. Erzberger and G. Deshpande for critical reading and comments on the manuscript; T.-Y. Huang for discussion on the functional generalizability of transient epithelial folds. This work was supported by the core funding at RIKEN BDR to Y.-C.W.; a Japan Society for the Promotion of Science (JSPS) Grant-in-Aid for Transformative Research Areas (A) grant to Y.-C.W. (grant no. 22H05167); a Human Frontier Scientific Program (HFSP) Young Investigators grant to S.L. and Y.-C.W. (grant no. RGY0082/2015); a Research Grant by the Deutsche Forschungsgemeinschaft to S.L. (grant no. LE 2787/3-1); G.K. was supported by HFSP Long-term postdoctoral fellowship (grant no. LT000597/2019-L).

**Author contributions** Y.-C.W. and S.L. conceived, designed and supervised the study. B.D., M.T. and Y.-C.W. generated reagents and performed experiments in *D. melanogaster*. V.K., G.K. and M.S. performed experiments in wild-type *D. melanogaster* and non-*Drosophila* species. B.D., V.K., G.K., M.S., M.T., Y.-C.W. and S.L. analysed the data and prepared the visualization. B.D., V.K., G.K, Y.-C.W. and S.L. wrote and edited the manuscript.

**Funding** Open access funding provided by Universität Hohenheim.

**Competing interests** The authors declare no competing interests.

**Additional information**
**Correspondence and requests for materials** should be addressed to Yu-Chiun Wang or Steffen Lemke.

| Group | a. Family | b. Species | c. Literature | d. Figure panels of interest | e. Image type | f. presence of GBE | CF |
|---|---|---|---|---|---|---|---|
| Cyclorrhapha | Calliphoridae | Calliphora erythrocephala | Davis et al, 1968 | Figure 1c-h | drawings * | ■ | ■ |
| | | Lucilia sericata | Mellenthin et al., 2006 | Figure 1c, d | transillumination | ■ | ■ |
| | | Lucilia sericata | Blechert et al., 2011 | Figure 3e, f | DIC in-situ | ■ | ■ |
| | | Calliphora vicina | Martin-Vega et al., 2016 | Figure 2e-i | transillumination | ■ | ■ |
| | Muscaidae | Musca domestica | Cantwell, 1976 | Figure 8, 9, 10 | stained sections | ■ | ■ |
| | | Stomoxys calcitrans | Ajidagba et al., 1983 | Figure 20,21,23 | SEM | ■ | ■ |
| | | Musca domestica | Sommer, 1991 | Figure 4e, 9a | DIC in-situ | ■ | ■ |
| | | Musca domestica | Wratten et al., 2006 | Figure 2f | DIC in-situ ** | ■ | ? |
| | Drosophilidae | Drosophila melanogaster | Bownes, 1975 | Figure 2, 3 | transillumination | ■ | ■ |
| | | Drosophila melanogaster | Turner and Mahowald, 1977 | Figure 1, 3, 6, 8-12 | SEM | ■ | ■ |
| | | Drosophila ananassae | Kuntz and Eisen, 2014 | Figure 4a | transillumination | ■ | ■ |
| | | Drosophila erecta | Kuntz and Eisen, 2014 | Figure 4a | transillumination | ■ | ■ |
| | | Drosophila melanogaster | Kuntz and Eisen, 2014 | Figure 4a | transillumination | ■ | ■ |
| | | Drosophila mojavensis | Kuntz and Eisen, 2014 | Figure 4a | transillumination | ■ | ■ |
| | | Drosophila persimillis | Kuntz and Eisen, 2014 | Figure 4a | transillumination | ■ | ■ |
| | | Drosophila pseudoobscura | Kuntz and Eisen, 2014 | Figure 4a | transillumination | ■ | ■ |
| | | Drosophila sechellia | Kuntz and Eisen, 2014 | Figure 4a | transillumination | ■ | ■ |
| | | Drosophila simulans | Kuntz and Eisen, 2014 | Figure 4a | transillumination | ■ | ■ |
| | | Drosophila virilis | Kuntz and Eisen, 2014 | Figure 4a | transillumination | ■ | ■ |
| | | Drosophila willistoni | Kuntz and Eisen, 2014 | Figure 4a | transillumination | ■ | ■ |
| | | Drosophila yakuba | Kuntz and Eisen, 2014 | Figure 4a | transillumination | ■ | ■ |
| | | Drosophila melanogaster | Eritano et al., 2020 | Figure 1a | fluorescence live imaging | ■ | ■ |
| | | Drosophila melanogaster | [This study] | Figure 1c | nuclear staining | ■ | ■ |
| | Tephritidae | Dacus tryoni | Anderson, 1962 | plate 2c-e | stained sections | ■ | ■ |
| | | Ceratitis capitata | Stefani et al., 2002 | Figure 2d | transillumination | ■ | ■ |
| | | Bactrocera oleae | Genc, 2014 | Figure 2b | transillumination | ■ | ■ |
| | | Bactrocera dorsalis | Suksuwan et al., 2017 | Figure 1e | transillumination | ■ | ■ |
| | | Ceratitis capitata | Strobl, 2022 | Figure 2II | fluorescence | ■ | ■ |
| | Syrphidae | Episyrphus balteatus | Lemke and Schmidt-Ott, 2009 | Figure 2d | DIC in-situ | ■ | ■ |
| | | Episyrphus balteatus | Lemke et al., 2010 | Figure 2e, e' | DIC in-situ | ■ | ■ |
| | Phoridae | Megaselia abdita | Wotton et al., 2014 | Figure 2 (stage 6-8) | DIC time lapse | ■ | ■ |
| | | Megaselia abdita | Caroti et al., 2015 | Figure 3c-d' | fluorescence | ■ | ■ |
| | | Megaselia abdita | [This study] | Figure 1d | nuclear staining | ■ | ■ |
| non-Cyclorrhapha | Empididae | Empis sp. | [This study] | Extended Data Figure 2 | transillumination | ■ | □ |
| | Stratiomyidae | Hermetia illucens | Tollenaar et al., 2021 | Figure 1e | transillumination *** | ■ | ? |
| | | Hermetia illucens | [This study] | Figure 1e | nuclear staining | ■ | □ |
| | Cecidomyiidae | Wachtliella persicariae | Wolf, 1969 | Figure 3l-n | transillumination | ■ | □ |
| | | Heteropeza pygmaea | Kaiser and Went, 1987 | Figure 1a | drawings * | ■ | □ |
| | | Aphidoletes aphidimyza | Havelka et al., 2007 | Figure 4a | transillumination | ■ | □ |
| | Sciaridae | Sciara coprophila | Bois, 1932 | Figure 6 | drawings * | ■ | □ |
| | | Sciara spec | Butt, 1934 | Figure F | drawings * | ■ | □ |
| | | Rhynchosciara americana | Vanario-Alonso et al., 1996 | Figure 1d-g | nuclear staining | ■ | □ |
| | | Rhynchosciara americana | Carvalho et al., 1999 | Figure 2b, c | nuclear staining | ■ | □ |
| | | Bradysia hygida | Uliana et al., 2018 | Figure 3D2-4 | nuclear staining | ■ | □ |
| | Scatopsidae | Coboldia fuscipes | [This study] | Figure 1f | nuclear staining | ■ | □ |
| | Culicidae | Aedes dorsalis | Telford, 1957 | Figure 4 | drawings * | ■ | □ |
| | | Culex tarsalis | Rosay, 1959 | Figure 3-5 | drawings * | ■ | ■ |
| | | Culex pipiens | Idris, 1960 | Figure 18b-g | drawings * | ■ | ■ |
| | | Aedes aegypti | Raminani and Cupp, 1975 | Figure 21 | stained sections **** | ■ | ? |
| | | Anopheles gambiae | Goltsev et al., 2004 | Figure 1e | DIC in-situ | ■ | □ |
| | | Culex quinquefasciatus | Juhn et al., 2008 | Figure 5e-h | DIC in-situ | ■ | □ |
| | | Culex quinquefasciatus | Yoon et al., 2019 | Figure 4e | DIC in-situ | ■ | □ |
| | Chironomidae | Chironomus riparius | Weismann, 1863 | Figure 5-10 | drawings * | ■ | □ |
| | | Chironomus riparius | Ritter, 1890 | Figure 12, 18, 19 | drawings * | ■ | □ |
| | | Chironomus riparius | Klomp et al., 2015 | Figure S5 | DIC in-situ | ■ | □ |
| | | Chironomus riparius | Urbansky et al., 2016 | Figure 1f | nuclear staining | ■ | □ |
| | | Chironomus riparius | [This study] | Figure 1g | nuclear staining | ■ | □ |
| | Simuliidae | Simulium pictipens | Gambrell and Jahn, 1933 | Figure 16, 20 | drawings * | ■ | □ |
| | Psychodidae | Phlebotomus papatasi | Abbassy et al., 1995 | Figure 7, 11, 12 | stained sections | ■ | □ |
| | | Clogmia albipunctata | Rohr et al., 1999 | Figure 4c | DIC in-situ | ■ | □ |
| | | Clogmia albipunctata | Garcia-Solache et al., 2010 | Figure 1c, d | DIC time lapse | ■ | □ |
| | | Clogmia albipunctata | Jimenez-Guri et al., 2014 | Figure 2 (stage 6-9) | DIC time lapse | ■ | □ |
| | | Clogmia albipunctata | Yoon et al., 2019 | Figure 2e | DIC in-situ | ■ | □ |
| | | Clogmia albipunctata | [This study] | Figure 1h | nuclear staining | ■ | □ |

**Extended Data Fig. 1** | See next page for caption.

**Extended Data Fig. 1 | Literature survey for the presence of CF and GBE in diptera.** Literature-based survey of CF and GBE presence across Diptera, complementing our species sampling. The CF is defined as a deep epithelial fold at the head-trunk boundary during early gastrulation. **a**, **b**, Fly families (**a**) and the species (**b**) in the order of Diptera for which references could be found. **c**, Literature references (see supplementary information for corresponding bibliography). **d**, Reference figure panels with images of embryos during early gastrulation that were used to interpret the presence or absence of GBE and CF. **e**, Description of image datatype in d. SEM, scanning electron micrographs; DIC, Differential Interference Contrast imaging; *in-situ*, whole mount micrographs of embryos from RNA *in-situ* hybridization experiments. **f**, Interpretation of the presence or absence of GBE and CF based on images in d. Black boxes, presence; hollow boxes, absence; question marks, unclear (see comments below). *, descriptive drawings are a classic resource for the depiction of embryonic development. However, given the potential for subjective interpretations[71], we avoid using these references as definitive evidence for the presence of GBE and/or CF in a given family. **, absence of images during early gastrulation, thus unclear evidence for the CF. Note a late furrow is visible in the image, though the stage is too late to fulfill the definition of the CF. ***, absence of images during early gastrulation, thus unclear evidence for the CF. Although no furrow is visible during late gastrulation, presence of an early transient furrow can not be excluded. ****, absence of images from the anterior region of the embryo, thus unclear evidence for the CF.

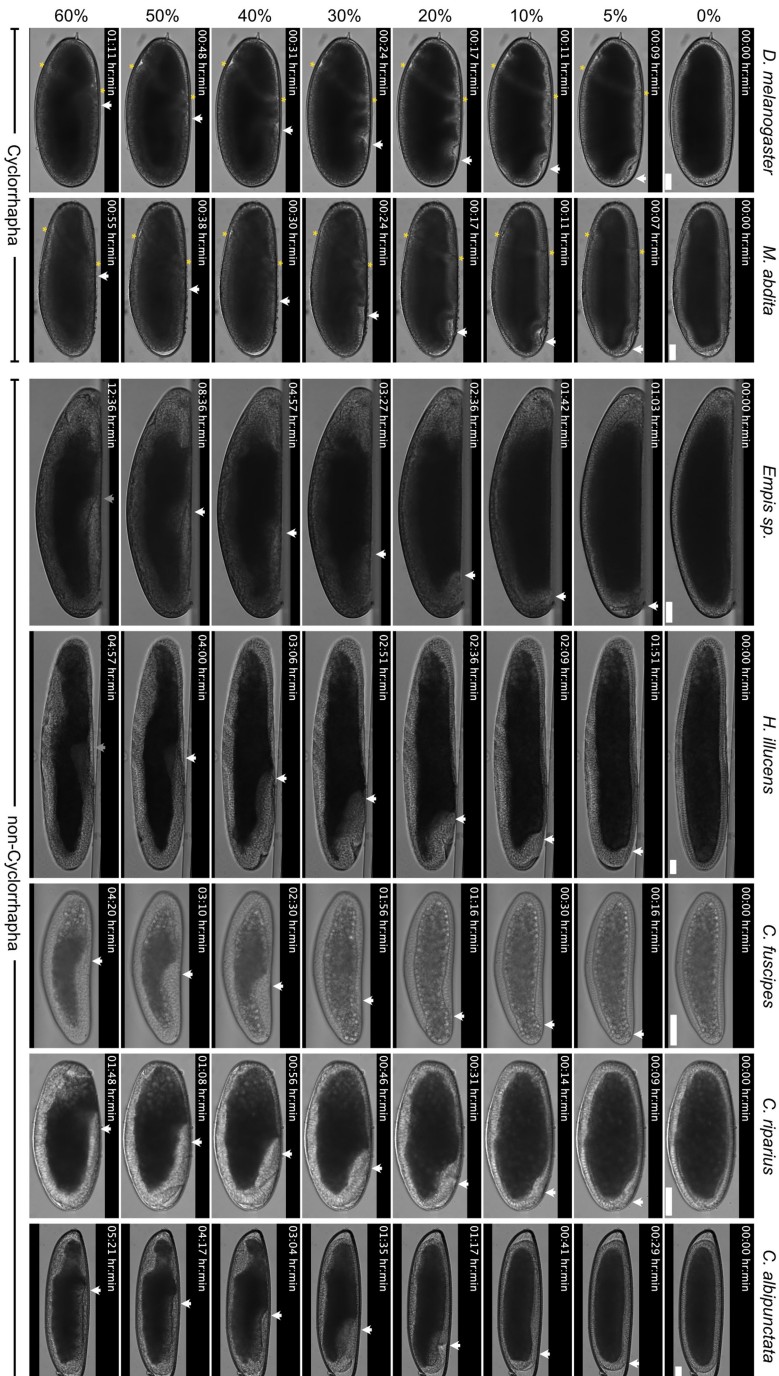

**Extended Data Fig. 2 | Progression of GBE is comparable in all species, while only _D. melanogaster_ and _M. abdita_ form the CF.** Figure shows montages of representative time lapse recordings of developing embryos from our species of interest. $T_0$ is defined as the initiation of gastrulation. Asterisks mark/track the CF in _D. melanogaster_ and _M. abdita_. In all species, the white arrows track the posterior end of the ectoderm at the indicated %EL on the left. In _Empis sp._ and _H. illucens_ GBE never reaches 60% EL; the gray arrows thus indicate the end point of GBE at ~55% EL. n = 35, 24, 3, 21, 6, 27, and 42 embryos respectively for _D. melanogaster_, _M. abdita_, _Empis sp._, _H. illucens_, _C. fuscipes_, _C. riparius_, and _C. albipunctata_. Of note, these numbers only include embryos with sagittal sections, as shown. We have also confirmed the presence (in Cyclorrhapha) and absence (in non-Cyclorrhapha) of CF throughout gastrulation, in embryos imaged in coronal views, since we can easily determine the initiation and end of GBE in these views. We do not present data with coronal views, however, as it is not possible to unequivocally determine the extent of GBE at a given time point, which is possible only in a sagittal view. Scale bars, 50 μm.

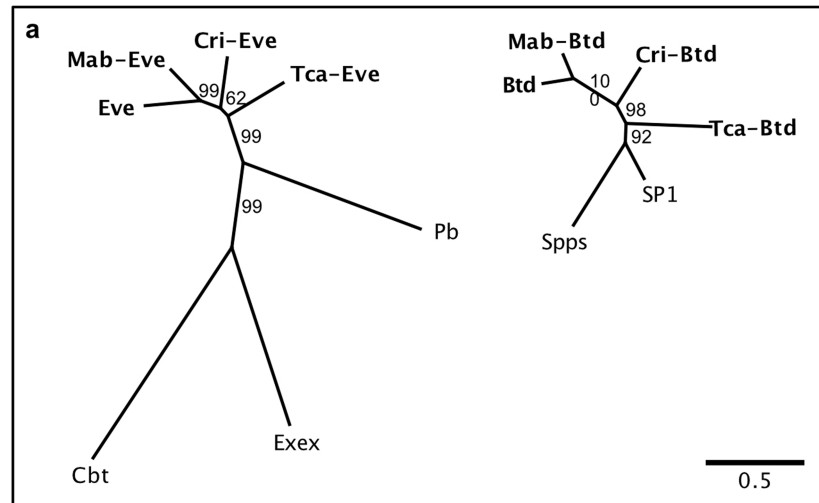

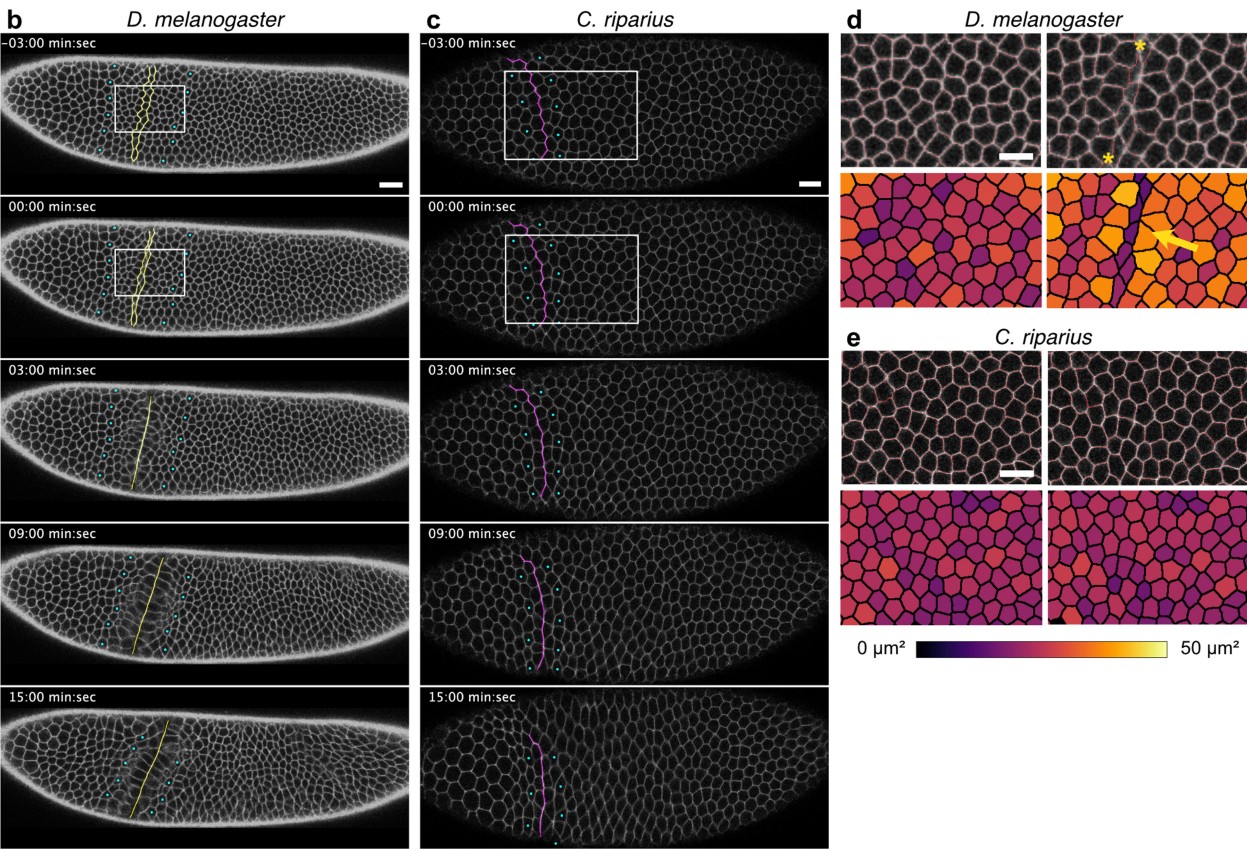

**Extended Data Fig. 3 | *Chironomus* does not form the CF at the head-trunk boundary; protein tree of predicted *eve* and *btd* orthologs in *M. abdita* and *C. riparius*. a**, Phylogenetic distances of Even-skipped (Eve) and Buttonhead (Btd) orthologs in *M. abdita* (*Mab*), *C. riparius* (*Cri*), *Tribolium castaneum* (*Tca*), and related *D. melanogaster* proteins were calculated in Geneious using the Jukes-Cantor model. Only reliability values above 60% are shown. Eve trees use full-length alignment; Btd trees are based on alignment to residues 151–433. Scale bar indicates changes per site. RefSeq protein sequences from NCBI are Btd (Buttonhead, NP_511100), Cbt (Cabut, NP_722636), Eve (Even-skipped, NP_523670), Exex (Extra-extra, NP_648164), Pb (Proboscipedia, NP_476669), SP1 (SP1, NP_727360), and Spps (SP1 like factor, NP_651232), Tca-Btd (NP_001107792) and Tca-Eve (NP_001034538). **b,c**, Time-lapse images from single z-slices of live *D. melanogaster* (**b**, n = 6) and *C. riparius* (**c**, n = 5) embryos expressing membrane markers; regions span ~10–75% EL (*D. melanogaster*) and ~15–85% EL (*C. riparius*). $T_0$ marks gastrulation onset. Colored lines highlight head-trunk boundaries - defined by the CF in *D. melanogaster* and by cell shape in *C. riparius*. Cyan dots track cells moving toward the boundary. Straightening occurs in both species, but *C. riparius* lacks the downstream cell shape changes and tissue flows seen during CF formation in *D. melanogaster*. Boxes indicate areas in d,e. Scale bars, 20 µm (b), 10 µm (c). **d,e**, Mesoscopic views at the head-trunk boundary in *D. melanogaster* (**d**) and *C. riparius* (**e**) at −3 and 0 min relative to gastrulation onset. Asterisks mark the CF region; arrows indicate apical area reduction in *D. melanogaster*, which is not seen in *C. riparius*. LUT bar shows cell area. Scale bars, 10 µm. Lower panels are also shown in Fig. 1m,n.

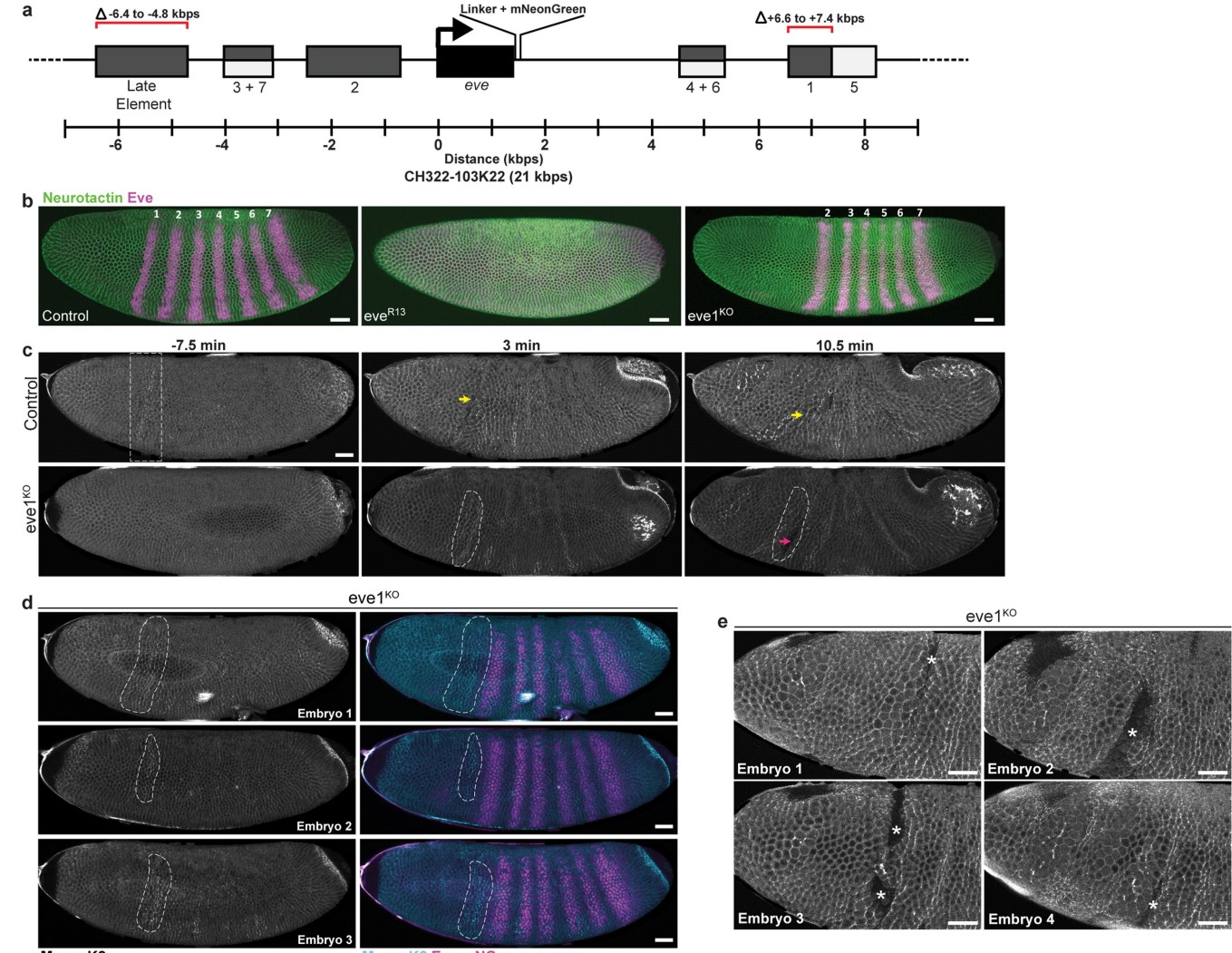

**Extended Data Fig. 4 | Phenotypic analysis of *eve1^KO* mutants. a**, Schematic representation of *eve^CH322-103K22-mNeonGreenΔst1ΔLE*. *eve* denotes the transcriptional region; numbers (1, 2, 3 + 7, 4 + 6 and 5) indicate stripe-specific enhancers; Late Element is the enhancer region that confers the seven-stripe pattern. The scale denotes genomic coordinates relative to the transcriptional start. **b**, Lateral surface projection of Neurotactin (as membranes, green) and Eve (magenta) immunofluorescence in a control (n = 5), *eve^R13* (n = 10) or *eve1^KO* (n = 14) embryo. Numbers are the Eve stripes. **c**, Time-lapse series of a control (top row, *MyoII-mKate2*/+, n = 8) or *eve1^KO* (bottom row, n = 4) embryo expressing MyoII-mKate2 showing a lateral surface projection. Dashed rectangle denotes planar polarized MyoII prior to CF initiation in controls; dashed outlines show the non-polarized MyoII patch at the head-trunk interface in *eve1^KO* embryos. Yellow arrows, CF; magenta arrow, head-trunk buckling. **d**, Lateral surface projection of three representative *eve1^KO* embryos (n = 4) expressing MyoII-mKate2 showing the MyoII patch (gray/cyan) that appears in the Eve stripe1 region, but does not overlap with the Eve stripe 2. Eve is visualized with the mNeonGreen tag (magenta). The Myosin patches are of variable sizes. Buckling typically coincides with the myosin patches, but does not occur immediately after Myosin accumulation, suggesting that the Myosin patches are unable to induce deformation. **e**, Lateral surface projection of four representative *eve1^KO* embryos (n = 4) expressing MyoII-mKate2 showing variable positions for the initiation of head-trunk buckling across the D-V axis (white asterisks).

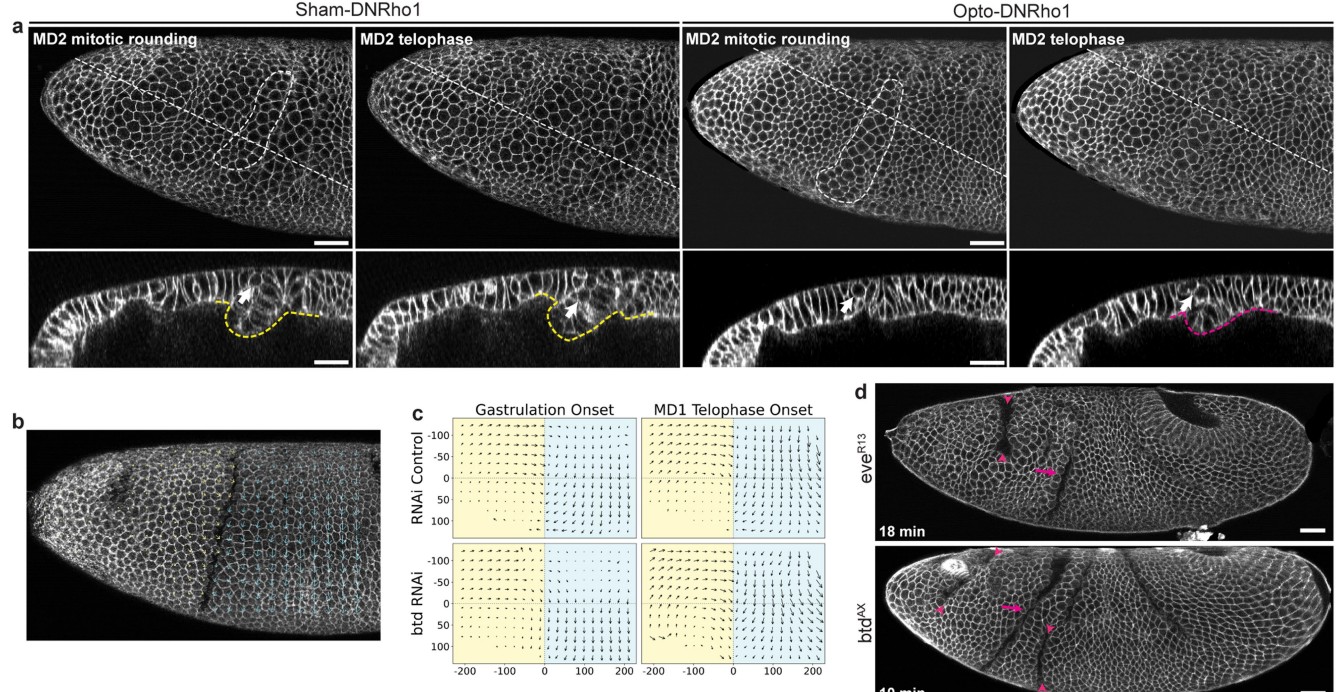

**Extended Data Fig. 5 | Phenotypic analysis following genetic and optogenetic ablation of the CF. a**, Lateral surface projection of a representative sham control (Sham-DNRho1) or a photoactivated (Opto-DNRho1) embryo expressing the Opto-DNRho1 system, visualized with a membrane marker (3xmScarlet-CaaX) showing the anterior half of the embryo that includes the head-trunk boundary (top rows) and a z-axis reslice (bottom rows) at two time points. White dashed outlines and white arrows, rounding or dividing MD2 cells; yellow outlines, CF; magenta outline, head-trunk buckling. Note in the sham control MD2 rounding occurs when the out-of-plane deformation of the CF was already present, while the onset of head-trunk buckling coincided with MD2 rounding when the CF was eliminated via optogenetic inhibition of DNRho1. **b**, Representative image of a control embryo superimposed with tissue flow field visualized with PIV. Yellow and cyan arrows are vector fields of the head and trunk regions, respectively. **c**, Tissue flow fields visualized with PIV in RNAi control (n = 4) and *btd* RNAi (n = 5) embryos at the onset of gastrulation and MD1 telophase. Yellow shaded rectangles, head region; blue shaded rectangles, trunk regions; x-origin, head-trunk boundary; y-origin, lateral midline; unit, pixel. **d**, Lateral surface projections of *eve*[R13] or *btd*[AX] embryos expressing Gap43-mCherry showing the presence of buckling that form elsewhere in the head region (magenta arrowheads), in addition to the head-trunk buckling (magenta arrows). Scale bar, 30 μm.

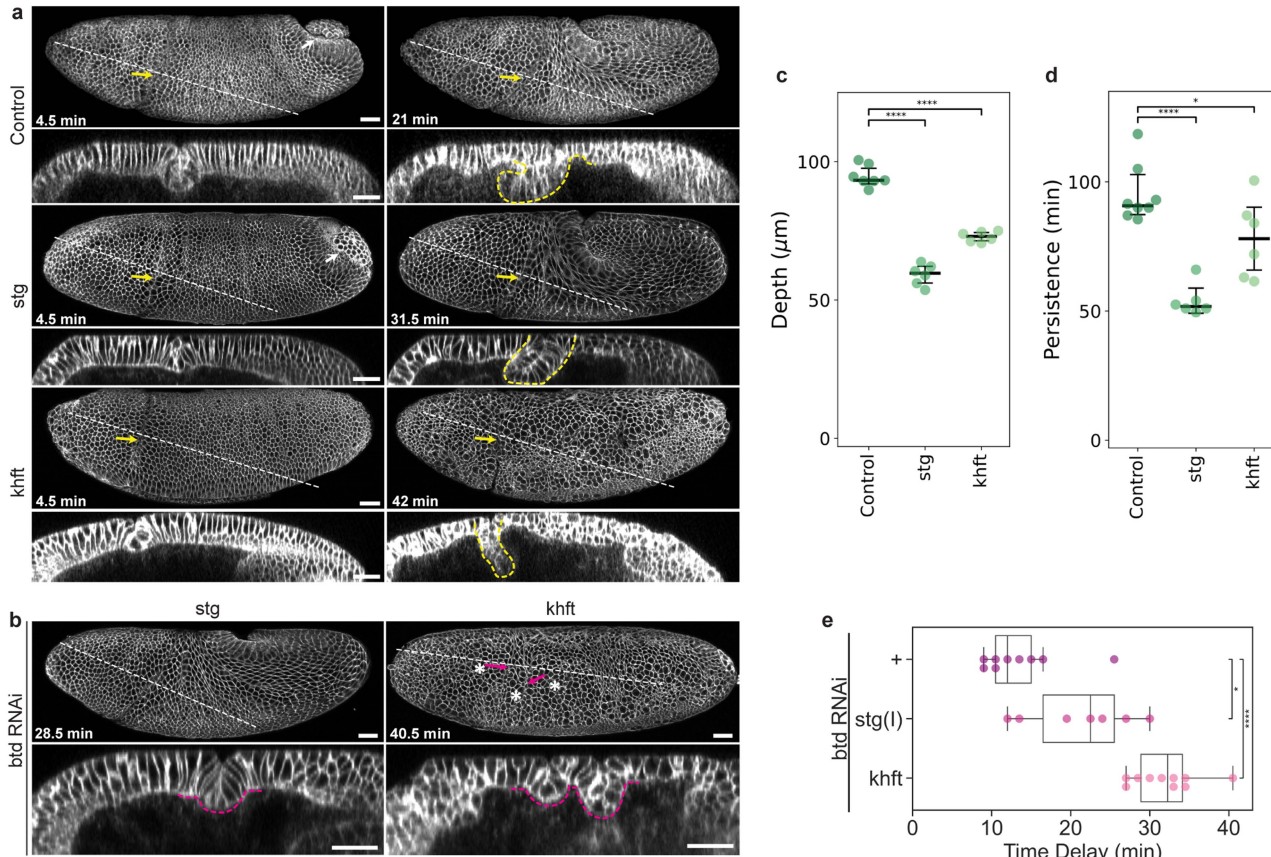

**Extended Data Fig. 6 | The effects of head mitosis and trunk convergent-extension on the CF and head-trunk buckling when the CF is lost.**
**a**, Time-lapse series of a representative control (*mat-tub-3xmScarlet-CaaX/+*, n = 7), *stg* mutant (n = 6), or *khft* (n = 6) mutant embryo, visualized with 3xmScarlet-CaaX showing a lateral surface projection (top rows) and a z-axis reslice (bottom rows). White arrows, PMG; yellow arrows and yellow dashed outlines, CF. Time is relative to the onset of gastrulation. Scale bars, 30 μm. **b**, Lateral surface projection (top panels) and z-axis reslice (bottom panels) of 3xmScarlet-CaaX showing late-stage buckling in *stg* (n = 7) or *khft* (n = 10) mutants (average buckling depth 40.31 ± 5.31 μm). Magenta arrows and magenta dashed outlines, head-trunk buckling; asterisks, late MDs (likely MD5,

6, and 11, based on their location, though the timing appears abnormal). Time is relative to the onset of gastrulation. Scale bars, 30 μm. **c**, **d**, Maximum depths (**c**) or durations (**d**) of the CF. Each dot represents an embryo; bold lines, median; whiskers, 95% CI. One-way ANOVA Tukey post-hoc test; **c**, ****, p < 0.0001 (*stg*, p = 0.000000; *khft*, p = 0.000000007 compared to control); **d**, ****, p < 0.0001 (p = 0.00001); *, p < 0.05 (p = 0.0334). Sample size: control=7, *stg* = 6, *khft* = 6. **e**, Timing of the onset of late buckling in the *stg*(l) and *khft* embryos that have been injected for *btd* RNAi, as compared to *btd* RNAi alone. Time is relative to the onset of gastrulation. One-way ANOVA Tukey post-hoc test; *, p < 0.05, (p = 0.0207); ****, p < 0.0001 (p = 0.0000003). Sample size: Control (*btd* RNAi alone)=9, *stg*(l) = 7, *khft* = 10.

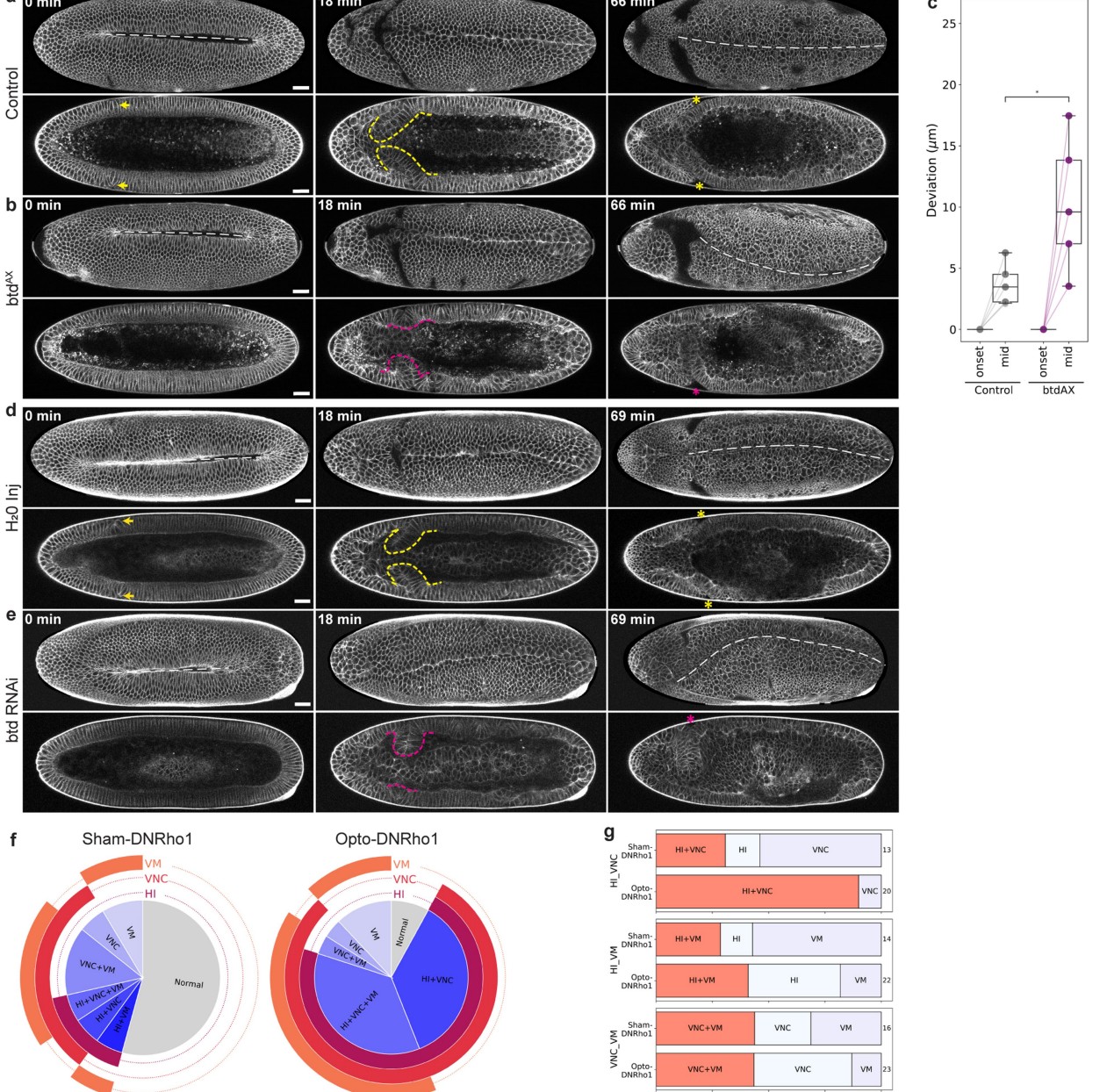

**Extended Data Fig. 7 | Genetic ablation of the CF using *btd* mutants or RNAi knockdown also causes midline distortion. a**, **b**, Time-lapse series of a control (**a**, Gap43-mCherry/+, n = 4), or *btd* mutant (**b**, *btd*^AX^, n = 5), visualized with Gap43-mCherry showing a ventral surface projection (top rows) and a single coronal section (bottom rows). **c**, Box plot showing mean deviation of the manually-marked ventral midline from the expected linear ventral midline position. Bold lines, median; boxes, interquartile range; whiskers, min-max range. Mann–Whitney U test (two-sided); *, p < 0.05, (p = 0.0317). Sample size: Control = 4, *btd*^AX^ = 5. **d**, **e**, Time-lapse series of a water injection control (**d**, H₂O inj, n = 5) or *btd* RNAi (**e**, n = 5) embryo, visualized with a 3xmScarlet-CaaX showing a ventral surface projection (top rows) and a single coronal section (bottom rows). White dashed lines, ventral midlines; yellow arrows and yellow dashed outlines, CF; yellow asterisks, bilaterally symmetric CFs; magenta arrows and magenta dashed outlines, head-trunk buckling; magenta asterisks, bilaterally asymmetric buckling. Time is relative to the onset of gastrulation. Scale bars, 30 µm. **f**, Pie charts showing the distribution of each phenotypic class observed in sham controls (Sham-DNRho1, n = 35), or photoactivated (Opto-DNRho1, n = 25) embryos expressing the Opto-DNRho1 system shown in Fig. 4d–f and the overlap between each of them. The innermost pie chart shows the individual phenotypic classes. Gray, normal; shades of blue, other phenotypes. Phenotype abbreviations: head involution defects (HI), ventral midline distortion (VM), ventral nerve cord condensation defects (VNC). The outer concentric circles show the combined percentage of embryos with HI (maroon), VNC (red) and VM (orange). **g**, Bar graphs comparing between sham controls (Sham-DNRho1) and photoactivated (Opto-DNRho1) embryos for the percentage co-occurrence of two phenotypes. The sample size is denoted at the end of each bar.

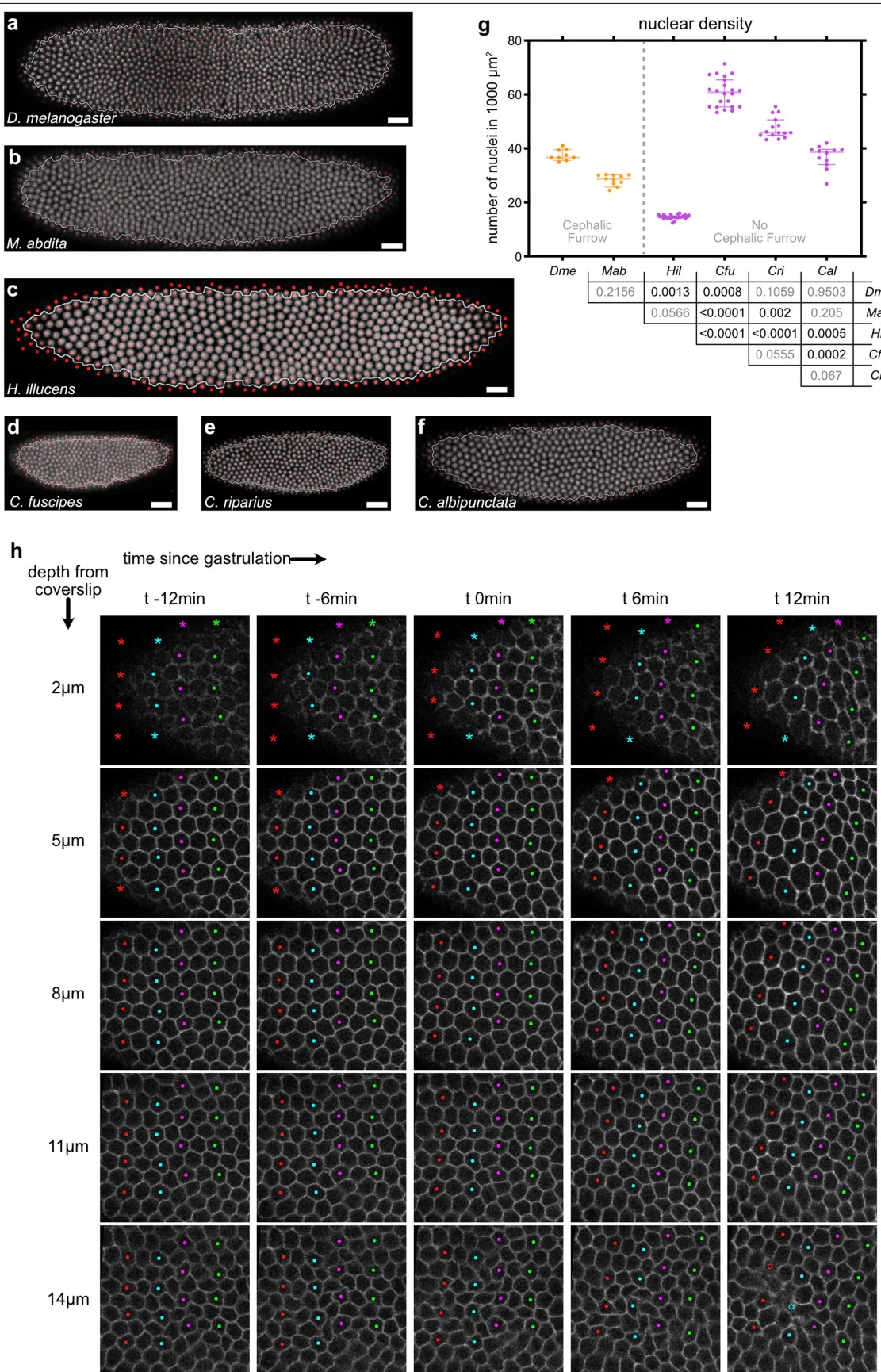

**Extended Data Fig. 8** | See next page for caption.

**Extended Data Fig. 8 | Blastoderm cell density does not correlate with presence or absence of the CF, and nuclear double-layering is not due to delamination or pseudostratification. a–f**, Representative images of fixed embryos stained with DRAQ5 (see Methods) and segmented nuclei (gray-scale; red dots mark individual nuclei). White boundaries enclose regions used to calculate nuclear density (red dots/area). Scale bars, 20 µm. **g**, Nuclear density distributions for six species, dashed vertical line separates species with and without the CF. Each dot represents an embryo; bold lines, median; whiskers, 95% CI. n = 9, 11, 25, 22, 15, 12 for *D. melanogaster, M. abdita, H. illucens, C. fuscipes, C. riparius, C. albipunctata*, respectively. Statistical significance determined using one-way ANOVA (non-parametric Kruskal-Wallis test with Dunn's uncorrected test for multiple comparisons), non-significant differences are indicated in gray. **h**, Time-lapse imaging of the early *C. riparius* head region shows cells retain columnar shapes until mitosis, without evidence for cone/frustum morphologies. This argues against pseudostratification or cell extrusion underlying nuclear double-layering (n = 5 embryos, 16 cells tracked). Images: time frames (columns), z-depths (rows), each panel 50 × 50 µm. Colored dots track cell positions; estimated positions are indicated when not visible due to embryo curvature (asterisk) or mitotic rounding (open circle).

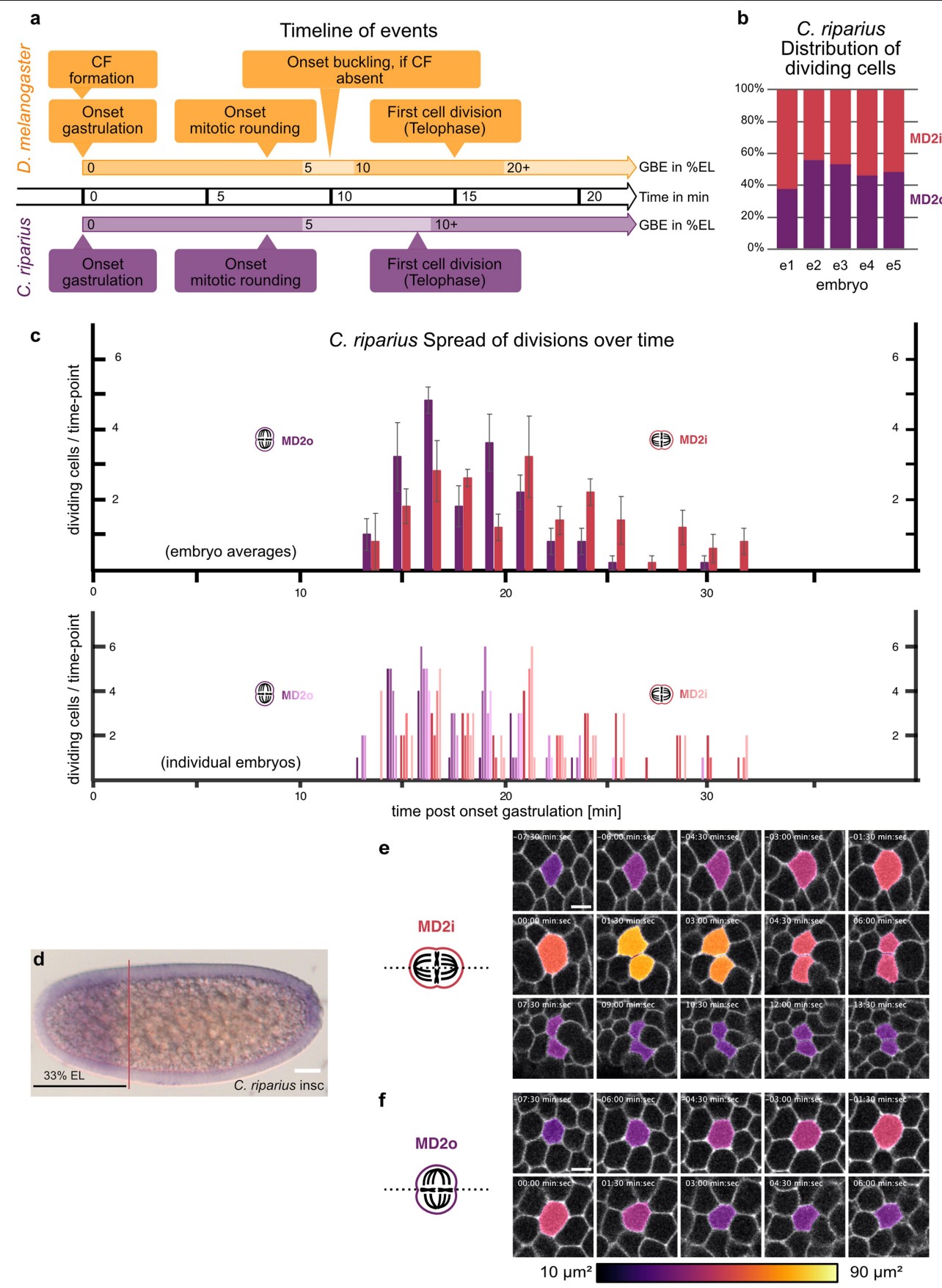

**Extended Data Fig. 9 |** See next page for caption.

**Extended Data Fig. 9 | Morphogenetic processes in *C. riparius* head ectoderm: relative chronology, temporal distribution of divisions, expression pattern of *Cri-Insc*, and cell apical area changes. a**, Timeline of early developmental events (yellow: *D. melanogaster*; magenta: *C. riparius*) in absolute time (in min) post gastrulation onset, defined as the onset of PMG in *D. melanogaster* and the onset of ventral-ward tissue flow in *C. riparius*. The onset of mitotic rounding was determined by the first cell displaying rounding in the domains analyzed (MD1 in *D. melanogaster*, MD2o in *C. riparius*). Complementary, the degree of GBE by % egg length (EL) in this timeframe is given for each fly. **b**, Distribution of *C. riparius* cells dividing out-of-plane (magenta/MD2o) and in-plane (red/MD2i) in the 5 embryos analyzed. **c**, Spread of divisions over time in *C. riparius* MD2o and MD2i in time post onset of gastrulation. Top panel, bars represent the average number of cells entering telophase within a 90 s window. Error bars, standard error. Bottom panel, bars show data from individual embryos (n = 5). **d**, In situ hybridization showing a broad expression pattern for *Cri-insc* in the head domain (n = 3). Red vertical line, anterior boundary of the *eve1* region at 33%EL. Scale bar, 25 μm. **e, f**, Time course of an event of in-plane (**e**) or out-of-plane division (**f**) with $T_0$ defined the same way as that in Fig. 5e, i.e., the onset of telophase. Cells are chosen from a representative embryo, **e** from MD2i and **f** from MD2o. The dashed lines in the schematics indicate the plane of the epithelium, and the LUT bar at the bottom indicates the color-code used for cell apical area. Scale bars 5 μm.

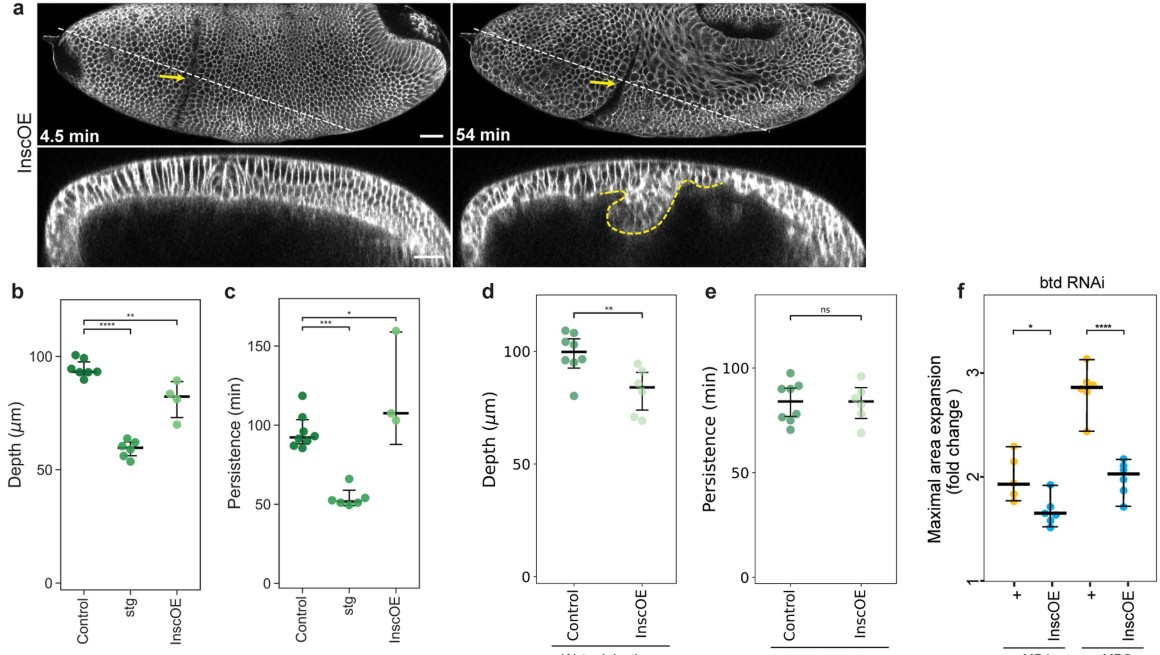

**Extended Data Fig. 10 | The effect of Insc overexpression on CF formation.**
**a**, Time-lapse series of a representative embryo of Insc overexpression in the head region (InscOE, n = 5), visualized with 3xmScarlet-CaaX showing a lateral surface projection (top rows) and a z-axis reslice (bottom rows). Yellow arrows and yellow dashed outlines, CF. Time is relative to the onset of gastrulation. The CF reaches the maximal depth at 54 min. Scale bars, 30 μm. **b**, **c**, Maximum depths (Sample size: control=7, *stg* = 6, InscOE=4) (**b**) or durations (Sample size: control=8, *stg* = 6, InscOE=3) (**c**) of the CF. Each dot represents an embryo; bold lines, median; whiskers, 95% CI. One-way ANOVA Tukey post-hoc test; **b**, ****, p < 0.0001 (p = 0.00000001); **, p < 0.01 (p = 0.0018); **c**, ***, p < 0.001

(p = 0.0003); *, p < 0.05 (p = 0.0371). **d**, **e**, Maximum depths (**d**) or durations (**e**) of the CF in water injected controls (n = 8) and water injected InscOE (n = 6) embryos. Each dot represents an embryo; bold lines, median; whiskers, 95% CI. Mann–Whitney U test (two-sided); **d**, **, p < 0.001 (p = 0.008); **e**, ns, not significant, p > 0.05 (p = 0.897). Control and *stg* measurements were replicated from Extended Data Fig. 6c,d. **f**, Maximal apical surface area expansion of MD1 and MD5 in control or InscOE embryos injected with *btd* RNAi (n = 6 respectively). Each dot represents an embryo; bold lines, median; whiskers, min and max. Two-sided t-test; *, p < 0.05 (p = 0.0166); ****, p < 0.0001.

# Reporting Summary

## Statistics

For all statistical analyses, confirm that the following items are present in the figure legend, table legend, main text, or Methods section.

| n/a | Confirmed | |
|---|---|---|
| ☐ | ☒ | The exact sample size (*n*) for each experimental group/condition, given as a discrete number and unit of measurement |
| ☐ | ☒ | A statement on whether measurements were taken from distinct samples or whether the same sample was measured repeatedly |
| ☐ | ☒ | The statistical test(s) used AND whether they are one- or two-sided<br>*Only common tests should be described solely by name; describe more complex techniques in the Methods section.* |
| ☒ | ☐ | A description of all covariates tested |
| ☒ | ☐ | A description of any assumptions or corrections, such as tests of normality and adjustment for multiple comparisons |
| ☐ | ☒ | A full description of the statistical parameters including central tendency (e.g. means) or other basic estimates (e.g. regression coefficient) AND variation (e.g. standard deviation) or associated estimates of uncertainty (e.g. confidence intervals) |
| ☐ | ☒ | For null hypothesis testing, the test statistic (e.g. *F*, *t*, *r*) with confidence intervals, effect sizes, degrees of freedom and *P* value noted<br>*Give P values as exact values whenever suitable.* |
| ☒ | ☐ | For Bayesian analysis, information on the choice of priors and Markov chain Monte Carlo settings |
| ☒ | ☐ | For hierarchical and complex designs, identification of the appropriate level for tests and full reporting of outcomes |
| ☒ | ☐ | Estimates of effect sizes (e.g. Cohen's *d*, Pearson's *r*), indicating how they were calculated |

*Our web collection on statistics for biologists contains articles on many of the points above.*

## Software and code

Policy information about availability of computer code

| | |
|---|---|
| Data collection | This study uses proprietary software to control the microscope hardware during image acquisition (data collection). However, the software itself has no bearing on the results. |
| Data analysis | For data analysis, this study uses custom-made ImageJ/FIJI (Version 2.16.0/1.54p) macros, Python (Version 3.12.5) scripts, Microsoft Excel (Version 16) and GraphpadPrism (Version 9) for data analysis; the macros and scripts used to facilitate data processing are available via Zenodo (doi:10.5281/zenodo.15870440). Other software used was Affinity Designer (2.6.3), Adobe Illustrator (29.5.1), and HandBrake (1.7.2). |

For manuscripts utilizing custom algorithms or software that are central to the research but not yet described in published literature, software must be made available to editors and reviewers. We strongly encourage code deposition in a community repository (e.g. GitHub). See the Nature Portfolio guidelines for submitting code & software for further information.

## Data

Policy information about <u>availability of data</u>

All manuscripts must include a <u>data availability statement</u>. This statement should provide the following information, where applicable:
- Accession codes, unique identifiers, or web links for publicly available datasets
- A description of any restrictions on data availability
- For clinical datasets or third party data, please ensure that the statement adheres to our <u>policy</u>

> Authors can confirm that all relevant data are included in the paper and/or its supplementary information files. Full resolution figures and files are available available via Zenodo (doi:10.5281/zenodo.15870440).

## Research involving human participants, their data, or biological material

Policy information about studies with <u>human participants or human data</u>. See also policy information about <u>sex, gender (identity/presentation), and sexual orientation</u> and <u>race, ethnicity and racism</u>.

| | |
|---|---|
| Reporting on sex and gender | N/A |
| Reporting on race, ethnicity, or other socially relevant groupings | N/A |
| Population characteristics | N/A |
| Recruitment | N/A |
| Ethics oversight | N/A |

Note that full information on the approval of the study protocol must also be provided in the manuscript.

# Field-specific reporting

Please select the one below that is the best fit for your research. If you are not sure, read the appropriate sections before making your selection.

☒ Life sciences         ☐ Behavioural & social sciences         ☐ Ecological, evolutionary & environmental sciences

For a reference copy of the document with all sections, see <u>nature.com/documents/nr-reporting-summary-flat.pdf</u>

# Life sciences study design

All studies must disclose on these points even when the disclosure is negative.

| | |
|---|---|
| Sample size | No statistical method was used to predetermine sample size. The sample size was based on previous studies in the field and indicated in the legends. |
| Data exclusions | No datasets were excluded from the analysis; all data resulting from valid experiments were included. Valid experiments were defined as experiments that were free of technical, instrumental, reagent and sample-related issues. |
| Replication | All experiments were highly reproducible, including those showing multiple classes of phenotypes under given experimental manipulations. Consistency across replicates is captured in each experiment by the statistics reported and reflected in the statistic tests performed. |
| Randomization | Sample randomization was performed on experiments involving externally administered perturbations (i.e. injections and photoactivation); sample randomization was not possible for experiments in which perturbations were introduced genetically, as all individuals of a given genotype were under such a perturbation. |
| Blinding | Blinding was not performed during data collection. It was infeasible in most experiments as knowledge of the experimental perturbations was either required for their execution or was a part of the data collection procedure. Blinding was not performed during data analysis as results typically were qualitatively or quantitatively distinct between experimental groups, or was unnecessary in cases where automatic imaging analysis methods were used, or infeasible when features of the phenotypes were apparent to the investigators which of the perturbations had been administered to the samples. Detailed methods and criteria for data analysis were extensively described in Methods and Supplementary Methods to ensure objectivity and reproducibility of results. |

# Reporting for specific materials, systems and methods

We require information from authors about some types of materials, experimental systems and methods used in many studies. Here, indicate whether each material, system or method listed is relevant to your study. If you are not sure if a list item applies to your research, read the appropriate section before selecting a response.

## Materials & experimental systems

| n/a | Involved in the study |
|-----|----------------------|
| ☐ | ☒ Antibodies |
| ☒ | ☐ Eukaryotic cell lines |
| ☒ | ☐ Palaeontology and archaeology |
| ☐ | ☒ Animals and other organisms |
| ☒ | ☐ Clinical data |
| ☒ | ☐ Dual use research of concern |
| ☒ | ☐ Plants |

## Methods

| n/a | Involved in the study |
|-----|----------------------|
| ☒ | ☐ ChIP-seq |
| ☒ | ☐ Flow cytometry |
| ☒ | ☐ MRI-based neuroimaging |

## Antibodies

| | |
|---|---|
| Antibodies used | Mouse monoclonal anti-Neurotactin (1:20, BP106, Developmental Studies Hybridoma Bank); rabbit polyclonal anti-Eve (1:500, gift from M. Biggin, Lawrence Berkeley National Laboratory, USA),  rat polyclonal anti-Btd (1: 500, gift from E. Wieschaus). The rabbit polyclonal anti-Eve and the rat polyclonal anti-Btd antibodies were provided as gifts and are not available as commercial products. |
| Validation | The non-commercial rabbit polyclonal anti-Eve and the rat polyclonal anti-Btd antibodies are custom-made antibodies generated and validated in the laboratories of M. Biggin and E. Wieschaus, respectively. |

## Animals and other research organisms

Policy information about studies involving animals; ARRIVE guidelines recommended for reporting animal research, and Sex and Gender in Research

| | |
|---|---|
| Laboratory animals | Embryos of Chironomus riparius, Clogmia albipunctata, Coboldia fuscipes, Drosophila melanogaster, Empis spec., Hermetia illucens, and Megaselia abdita were used in this study and all experiments were carried out according to the equivalent of Campos-Ortega staging for D. melanogaster between stage 5 and larval hatching. |
| Wild animals | The study did not involve wild animals. |
| Reporting on sex | The experimental procedure did not allow identifying the sex of the embryos at the time of the experiment. |
| Field-collected samples | The study did not involve samples collected from the wild. |
| Ethics oversight | No ethical approval or guidance was required for this study of invertebrate developmental processes. |

Note that full information on the approval of the study protocol must also be provided in the manuscript.

## Plants

| | |
|---|---|
| Seed stocks | N/A |
| Novel plant genotypes | N/A |
| Authentication | N/A |

