## [Peer Review File · Nature]

Divergent evolutionary strategies preempt tissue collision in gastrulation

Corresponding Author: Dr Steffen Lemke

Version 0:

Reviewer comments:

Referee #1

(Remarks to the Author)

This paper was submitted back-to-back with the study by Vellutini et al., and supports the same thesis: that the function of the cephalic furrow in fruit flies is to serve as a tissue sink that absorbs compressive stress due to germ band extension and anterior mitotic domains. Although the two papers come to the same conclusion, and contain some shared experiments (which is valuable as it shows reproducibility), they are not redundant, and this manuscript nicely complements Vellutini et al. by bringing in some aspects that could have been found to be missing from the other manuscript. Some of the highlights are:

- some highly specific ablations of the cephalic furrow either with very precise genetic changes (specific eve1 stripe ablation) or without any genetic change at all (via optogenetic inhibition of contractility)
- a broad taxonomic sampling, including 2 cyclorrhaphans (*Drosophila* and *Megaselia*), and 2 non-cyclorrhaphans (*Clogmia* and *Chironomus*)
- an extensive report of the variable morphology of ectopic folds in terms of number, shape, size, and location
- finally, I found the use of PIV as a tool to directly visualize tissue collision and divergence of the velocity fields at the level of the cephalic furrow to be a very nice touch, that makes immediately visible the mechanical conflict that the cephalic furrow resolves.

The paper appears overall compelling. A few suggestions for improvement are below:

1) I. 107-109: "In blastoderm stage embryos, Cri-btd is expressed in two separate domains in the head region (Fig. 1k); Cri-eve is expressed in six stripes, while the seventh pair-rule stripe comes up at the posterior pole after the onset of GBE (Fig. 1l). Importantly, the patterns do not overlap (Fig. 1k, l)." The evidence shown for non-overlap of eve and btd in *Chironomus* consists of two single WMISH aligned on the figure panel. Unfortunately, the boundaries of the two expression domains are somewhat close, and in the absence of double WMISH it is difficult to be fully convinced there's no overlap. If double WMISH is impossible in *Chironomus*, could the authors maybe attempt two simultaneous single WMISH with the same stain (NBT/BCIP) on the same embryos, to show that an expression-free gap remains between the two domains?

2) I. 337-339: 2 phenotypic classes of InscOE are defined: Class I (40% of embryos) with near-complete loss of buckles, Class II (60% of embryos) with head-trunk buckling. These classes seem to have been subjectively recognized and classified. I think the degree of buckling could be better quantified, for example by quantifying tortuosity (as in Vellutini et al.) This might reveal whether there is a small reduction of buckling (hard to appreciate by eye) even in "class I" embryos, for example.

3) I. 369-372: "our phylogenetic mapping reveals an abrupt and near concurrent emergence of the two morphogenetic traits: predominantly in plane division for the early mitotic cells in the head and active out-of-plane deformation at the head-trunk boundary". This is true for the cephalic furrow, but unclear for in-plane divisions based on the data presented. Outside Cyclorrhapha, the 2 genera studied (*Clogmia* and *Chironoma*) belong to 2 relatively closely related families, and thus are part of a monophyletic group separated from Cyclorrhapha by 3 intermediate branches without cephalic furrow, but for which division patterns in the head (in-plane or out-of-plane) are unknown. To make any statement about the origin of in-plane

divisions, some of these branches would need to be characterized. Otherwise, a lot of other situations are possible: intermediate branches (and the last common ancestor of flies) might show in-plane divisions, or might lack mitotic domains in the head, or mitotic domains might exist but not be active at the same time as germ band extension (thus preventing tissue collision), etc.

Formatting suggestion: Fig. 1a: could the species be mapped onto the phylogeny (eg by writing (c), (e), etc next to the family names at the tips of the tree)

Referee #2

(Remarks to the Author)

Summary

This MS examines a specific morphogenetic movement that occurs in some dipteran (flies & mosquitoes) embryos: the formation of a temporary trough of cells called the cephalic furrow. The formation and subsequent disappearance of this furrow has been well known from studies of *Drosophila* embryogenesis for decades, but it appeared to have no developmental function, as the structure is temporary and the cells that form it do not appear to adopt any specific fate that required the furrow to form. In this MS the authors perform a number of perturbations of furrow formation, and find that when they prevent it from forming, other, smaller and less regular furrows form in other regions of the embryo. These irregular ectopic furrows are also temporary

Major Comments

1. The hypothesis that formation of the cephalic furrow confers a fitness advantage and is therefore under selective pressure is at the heart of the paper, because the authors are claiming to have solved the problem of "why" the furrow forms. However, the evidence for this is weak. The two lines of evidence presented are (1) observations of "increased frequency of midline distortion or rotation" in embryos in which furrow formation was blocked with their optogenetic approach (Figure 3). However, there is no evidence provided that midline distortion results in decreased frequency of hatching or in worse outcomes at post-embryonic development, neither in this optogenetic experiment nor in any of the other genetic manipulations performed in the MS to compromise CF formation; (2) the assertion that because the furrow forms precisely and robustly in wild type embryos, it must confer fitness (page 7). This is an example of a very common yet incorrect application of the theory of evolution by natural selection: the mere existence of a biological trait is not evidence that it evolved due to natural selection. Rather, natural selection is just one of multiple different potential mechanisms of evolutionary change. Without compelling evidence for any biological function or evolutionary relevance of this morphogenetic process, the paper remains a well-written and well-presented comparative quantitative developmental biology manuscript showing interesting differences in cell behaviours between dipterans with and without cephalic furrows, but it does not rise to an unusual level of general interest. Related, on line 12 the subjective description of their finding that the CF is an evolutionary novelty as "surprising" and of transitions around CF presence as "seemingly abrupt" should be removed. This is related also to minor comment #6.
2. The sample size is very low (5 or fewer) for most of the experiments in this paper. There is no data presented to address the degree of wild type variance in CF or head-trunk buckle (ectopic fold) formation, so it is impossible to know what sample size would be necessary whether ectopic folds form more often than expected in cases of CF abrogation, but if it is indeed so rare that a sample size of 3 (as presented for many genetic conditions herein) is sufficient, then this evidence should be provided from assessments of wild type embryos, which in this laboratory system are easy to obtain in large numbers.
3. In Figure 1a, 42 families (of the nearly 200 in Diptera) are shown in the tree, only 12 of which appear to have available data on CF furrow formation. Why were these 42 chosen for display when most of them lack data on CF formation, and why are there 13 families (rather than 12) shown in the supplementary table (Fig1 Supp 4). The claim in the figure legend that once the CF appeared it was not secondarily lost, is not well supported given that there are so many missing data points on the tree.

Minor Comments

1. Please refer to species throughout with the Linnean abbreviation (e.g. *C. riparius*) rather than with the genus name (e.g. Chironomus)
2. The concept of "tissue tectonics" (page 6) has been used in other examples of similar epithelial buckling and furrowing events during development (e.g. Blanchard et al 2009 Nature Methods; Busby & Steventon 2021 Interface Focus; Adams, 2011 FASEB Journal). It would be useful for the authors to contextualize their observations and use of this term, with those of others - are the authors adding something new to this concept, or is this pre-existing concept a useful and accurate descriptor of the phenomena they report here?
3. page 9 lines 299-300: the meaning of "an in-plane dividing on average cell" is unclear.
4. page 10 line 333: the meaning of "qualitatively comparable" is ambiguous - please clarify.
5. page 11 line 372: please avoid referring to branches whose tips are extant organisms as "basal." "Early diverging" would be more appropriate. Related, on page 12 lines 408-409, no single species represents any other; please rephrase as "species belonging to early diverging branches" or similar.
6. page 11 lines 375 - 377: This sentence appears to state a hypothesis but is phrased as an assertion - please state the fact that this is not known, is more directly, for example "we speculate that," rather than using "likely" without explaining why the authors believe this is more likely than some other outcome of an early in-plane division origin. Also in this sentence, "stronger" than what? The same applies to the following sentence and to others in this paragraph - there is no apparent basis for claiming one scenario to be more likely than another; the authors can simply state that this is their speculation, and ideally explain the evidence-based logic behind favoring these hypotheses over others.
7. Figure 11: where is the 7th eve stripe?
8. Figure 3e,f: what proportion of the total midline length is the deviation shown here?
9. Figure 1 supplement 3: as these trees are unrooted, the evolutionary relationships of the *C. riparius* eve and btd candidate

sequences are not convincing. The in situ hybridization patterns lend support to the author's homology hypothesis for these sequences but these trees do not help - rooting the trees would help this.

Comments on the relationship of this MS to the accompanying Vellutini et al. MS

1. Dey et al point out that using *eve* or *btd* mutants to study the cephalic furrow is problematic because mutations in both genes cause severe patterning defects beyond the elimination of the furrow (page 4). However, this severe limitation is not acknowledged by Vellutini et al., who do use this approach.
2. Dey et al. use the terminology "head/trunk buckles" (page 4) while Vellutini et al. call them "ectopic folds." As these MSS are clearly intended to be companion pieces examining the same biological problem, it would be better to use consistent terminology in both.
3. Dey et al. observe late onset of ectopic folds in *stg* mutant embryos (their class I phenotype) but this is not reported in Vellutini et al., who examined embryos of the same genotype. Please explain this discrepancy.

Referee #3

(Remarks to the Author)

This manuscript describes a multipronged approach to the structure, and in particular the function, of the Cephalic Furrow (CF), a seemingly unimportant multicellular pattern that arises, transiently, in the anterior third of the early *Drosophila* embryo. As it happens the study turns out a solid and original piece of work on the role of mechanics in pattern formation and the evolution of the mechanisms adopted by constrained cellular fields to cope with mechanical stresses during morphogenesis. It is an original and insightful piece of work that takes advantage of the detail in which one can probe mechanism in *Drosophila*, and extrapolates it to related insects, to understand the evolution of fundamental issues in developmental biology.

Dey et al start by looking in detail at the dynamics of the organization of the CF in *Drosophila* and probe it through a range of techniques and mutants and great attention to detail. The authors propose that in Cyclorrhaphan flies the emergence of the buttonhead head-trunk boundary expression domain, overlapping with the first stripe of even-skipped is a central event leading to the evolution of cephalic furrow formation in Diptera. In *Drosophila* they probe mechanical and morphogenetical instabilities that arise in the absence of cephalic CF formation using genetic and optogenetic perturbations and sophisticated cell and tissue dynamics analysis. They observe cell divisions associated with this event and suggest that the role of the CF is to cope with stresses associated with the strong movements during early events of gastrulation in Cyclorrhaphan Diptera.

To go beyond the tests that they do, which involve a surgical removal of the CF through a rescue construct for *eve* that lacks the stripe 1 enhancers, they look at other insects, in particular non-Cyclorrhaphan flies that represent a different branch and lack CF and find that they have evolved a different mechanism to cope with the same challenges, in this case, out of plane divisions. In non-Cyclopharran flies (*Chironomus*) they characterised the orientation of head cell divisions and concomitant cell shape changes and explore the role of out-of-plane divisions in reducing compressive stress. They report that both mechanisms are present in a representative species of the most basal branches of Cyclopharran flies (*Megaselia*) leaving open the question on how these morphogenetic control mechanisms evolved.

The work is a rare piece integrating development, evolution and cell biology and provides a very detailed example of the role of mechanobiology in pattern formation which will serve as a basis to consider the evolution of developmental mechanisms.

It is difficult to find a major flaw, though it would have been good to hear if other transient folds that appear also during gastrulation have a similar function.

The manuscript is long which helps in the detail but this is maybe good.

Version 1:

Reviewer comments:

Referee #1

(Remarks to the Author)

I am fully satisfied with the revision.

While I could have been content with the addition of a verbal caveat regarding the 'abrupt and near concurrent' emergence of in-plane divisions and cephalic furrow formation, the authors went above and beyond, by studying four additional fly species, including three that were collected *de novo* in the wild - now supporting their point with a much broader taxonomic sample. I also thank them for establishing HCR to conclusively rule out *btd/eve* co-expression in *Chironomus*, and performing more detailed quantification of phenotypes. Finally, I strongly appreciated the additional observations of head involution and ventral nerve cord condensation defects documented in response to Reviewer 2. I think the author's argument for functionality of the cephalic furrow (and for the specific function they hypothesise) is now compelling.

Given how lengthy, effort-intensive, and extensive the first revision was, I feel it would be unreasonable to ask for more data. Congratulations on a thorough and compelling paper.

Referee #2

(Remarks to the Author)

Summary:

The authors have made a good faith attempt to address all concerns raised in the previous round of review, and I find this MS improved as a result. The most notable improvements are the addition of double HCR to address the btd/eve expression pattern in *C. riparius* and the attempts to find deleterious effects resulting from compromised CF formation in *D. melanogaster*, to support the authors' main claim that the CF is an adaptive evolutionary novelty. Despite these changes to the MS, many of which have improved an already strong MS, I remain of the opinion that the evidence that the cephalic furrow confers an evolutionary advantage that explains "why" it evolved, is not strong. I find their hypothesis for the divergent cell biological mechanisms regulating CF formation, and underlying the differences in CFs between dipterans, to be strongly supported by their experiments. However, I disagree with the authors regarding their strong conclusion that CF evolution was an adaptive response that solved a mechanical problem in development.

Major comments:

1. Regarding the assertion that the presence of the CF, and that fact that it appears in all individuals of species that have a CF (which is how I believe the authors define "robustness" in this context), is evidence that it evolved by natural selection and confers an adaptive benefit: I appreciate the clarification of writing in many instances, and acknowledge that the authors are trying to make an argument for evidence of stabilizing selection of this trait. The expanded taxon sampling and the addition of the double HCR for *C. riparius* are strong additions that are relevant to this claim. Despite the expanded taxon sampling, in my view I don't believe that the authors can reject the hypothesis of loss of CF in some lineages simply because of the long evolutionary distances and wide radiation of the clade. The authors argue against loss "because then we would expect to see at least some variation or an occasional loss of CF formation." The question is whether or not we would expect that sampling 6 out of the over 70,000 described species of *Cyclorhapha* would happen to reveal at least one such loss. Through no fault of their own given the large evolutionary radiation - I am not suggesting that the authors sample all 70,000 species! - the authors cannot reject this null hypothesis, and they should at a minimum acknowledge this.
2. Regarding the claim that the CF is "beneficial" based on observing midline distortions alone: I appreciate that the authors extended their analysis of embryos in which they had disrupted the CF, to assess whether later stages of embryogenesis and/or embryonic survival at least to hatching were impacted. This is a strong addition to the MS. However, as for point (1) above, I find that the authors have interpreted these data too strongly. I can see in their modified Figure 3 shows numerically greater incidence of head involution and ventral nerve cord defects in the CF-disrupted embryos relative to shams (as a note of rigor, the authors should additionally perform a statistical comparison of these control and experimental distributions and provide the results in the MS). The authors interpret this as evidence that CF disruption has specific negative impacts on head involution and ventral cord formation/morphogenesis. However, the fact that the genetic background they are using to induce specific CF disruption has such a high level of embryonic lethality, and that even sham embryos show high incidence of head involution defects and ventral nerve cord defects, suggests an alternative interpretation of these data to the one favored by the authors: namely, that these embryos already have a tendency to have problems in these specific developmental processes (among others), such that they are dealing with a sensitized background that is easy to "push over the edge" into having a higher incidence of the problems they already started with, by subjecting them to any kind of stress. I appreciate that the authors are working with this genetic background because it allows them to more specifically interfere with CF formation than the genetic mutants used in the companion study, whose interpretation was potentially complicated by pleiotropy. However, the additional experiments that the authors have performed for the revised MS have revealed that their genetic background, too, contains pleiotropic effects that are directly related to the developmental defects they would like to claim are caused by CF disruption. Glossing over these problems by calling them "technical difficulties" in the main text (line 382) is misleading. Thus, I do not find that the case for an adaptive benefit of CF formation to be any stronger in this revised MS, than in the original, although I reiterate that I appreciate the authors' expanded investigation into additional phenotypes in the CF-disrupted embryos.

Minor comments:

1. I appreciate that the authors have used the accepted Linnean form of abbreviated genus/full species, after the first mention of the species with its full binary species name. However, the authors do not need to put the genus abbreviation/full species into parentheses after the first incidence of species mention. For example, in line 65, the authors can simply say "*Drosophila melanogaster*" and then proceed to use "*D. melanogaster*" in line 74, without having put that abbreviation into parentheses in line 65.

Referee #3

(Remarks to the Author)

The revision of the manuscript by Dey et al. has strengthened the argument that they put forward. On its own it makes a strong case for an evolutionary divergence in the way a tissue in the early embryo copes with mechanical stresses. The additional optogenetic experiments are particularly enlightening. The response to the reviewers has resulted in a number of additional experiments and observations that make the case clear. The summary figure is much appreciated as it helps conceptualize the issues at stake and the solution. Together with Vellutini et al. make an important point for the field of evolutionary and developmental biology.

Alfonso Martinez Arias

Response to the referees

We would like to thank the editor and the referees for their positive appreciation of our work and their constructive comments. We have attempted and are able to address all of their concerns. These efforts have confirmed and strengthened our conclusions, and all together substantially improved our manuscript.

To facilitate the review of our revised manuscript, we provide below a summary of major changes that we made to the manuscript, followed by a detailed point-by-point response to the referee's concerns.

Summary of key revisions.

- Extended phylogenetic sampling: We have now collected data from three additional non-cyclorrhapha families – Scatopsidae (*Coboldia fuscipes*), Stratiomyidae (*Hermetia illucens*), and Empididae (*Empis sp.*). For these species, we present time-lapse DIC imaging of embryo-scale morphogenetic movements, confirming the absence of a cephalic furrow or tissue buckles during gastrulation. Additionally, for *H. illucens* and *C. fuscipes*, we provide new data on out-of-plane divisions, early gastrulation stage tissue morphology, and blastoderm cell density (quantified from fixed embryos with nuclear staining).
- Gene expression analysis: To simultaneously visualize *btd* and *eve* expression patterns and confirm the absence of overlap in their expression domains, we performed Hybridization Chain Reaction (HCR)-based *in-situ* hybridization in *C. riparius* embryos.
- Assessing the impact of cephalic furrow loss: To evaluate whether the absence of the cephalic furrow affects embryonic development, we performed additional optogenetic inhibition experiments, followed by long-term (≥ 18 -hour) live imaging. Since direct assessment of hatching rates was not feasible due to system-wide background lethality, we instead used successful development as a proxy for organismal fitness. We consistently observed severe defects in late-stage head involution and ventral nerve cord formation, indicating that CF loss has significant negative consequences for embryonic development.
- Increased sample sizes and expanded analyses: We repeated and expanded our experiments on cephalic furrow formation and head-trunk buckling in *D. melanogaster*, particularly in *btd* RNAi and the *insc*-OE manipulations of cell division orientation, alongside appropriate controls. With these new data, we extended our analyses of depth and persistence for cephalic furrow and head-trunk buckle.
- Enhanced statistical power: We increased sample sizes for area expansion analyses in the mitotic domains of head ectoderm, incorporating additional *D. melanogaster* and *C. riparius* embryos, thereby improving the statistical strength of our conclusions.
- New conceptual model figure: To better illustrate the evolutionary divergence in strategies to prevent head-trunk tissue collision in cyclorrhaphan and non-cyclorrhaphan flies, we have added a new model figure summarizing our findings.

Below are our detailed response to the referees' comments; to facilitate readability, we used the following font emphases/colors:

Comments from the referees

Our response to their comments

Changes to the manuscript

Referee #1 (Remarks to the Author):

This paper was submitted back-to-back with the study by Vellutini et al., and supports the same thesis: that the function of the cephalic furrow in fruit flies is to serve as a tissue sink that absorbs compressive stress due to germ band extension and anterior mitotic domains. Although the two papers come to the same conclusion, and contain some shared experiments (which is valuable as it shows reproducibility), they are not redundant, and this manuscript nicely complements Vellutini et al. by bringing in some aspects that could have been found to be missing from the other manuscript. Some of the highlights are:

- some highly specific ablations of the cephalic furrow either with very precise genetic changes (specific eve1 stripe ablation) or without any genetic change at all (via optogenetic inhibition of contractility)**
- a broad taxonomic sampling, including 2 cyclorrhaphans (Drosophila and Megaselia), and 2 non-cyclorrhaphans (Clogmia and Chironomus)**
- an extensive report of the variable morphology of ectopic folds in terms of number, shape, size, and location**
- finally, I found the use of PIV as a tool to directly visualize tissue collision and divergence of the velocity fields at the level of the cephalic furrow to be a very nice touch, that makes immediately visible the mechanical conflict that the cephalic furrow resolves.**

We are very grateful to the referee for their positive evaluation of our work.

The paper appears overall compelling. A few suggestions for improvement are below:

1) l. 107-109: "In blastoderm stage embryos, Cri-btd is expressed in two separate domains in the head region (Fig. 1k); Cri-eve is expressed in six stripes, while the seventh pair-rule stripe comes up at the posterior pole after the onset of GBE (Fig. 1l). Importantly, the patterns do not overlap (Fig. 1k, l)." The evidence shown for non-overlap of eve and btd in Chironomus consists of two single WMISH aligned on the figure panel. Unfortunately, the boundaries of the two expression domains are somewhat close, and in the absence of double WMISH it is difficult to be fully convinced there's no overlap. If double WMISH is impossible in Chironomus, could the authors maybe attempt two simultaneous single WMISH with the same stain (NBT/BCIP) on the same embryos, to show that an expression-free gap remains between the two domains?

This is a very valid point. Following the suggestion of the referee, we now established HCR in *Chironomus* and visualized the expression of *Cri-eve* and *Cri-btd* in the same embryo using multi-color confocal imaging. We counterstained DNA with DRAQ5 to visualize the nuclei, which allowed us to estimate the spatial separation of the *btd* and *eve* expression with single-cell resolution in *C. riparius* at two stages, i.e., just before and after the onset of gastrulation. These analyses have replaced our previous histochemical in situ hybridizations and are presented in updated panels of the main Figure 1 (Fig. 1k, l). The results demonstrate that *btd* and *eve* in *C. riparius* do not overlap before the onset of gastrulation and are separated by a gap of about 1-2 nuclei after the onset of gastrulation.

To reflect the more specific findings resulting from double WMISH in the main text, we refer to the updated figure in the revised text as follows:

(P3, L93-99): In the blastoderm stage, *Cri-btd* is expressed in two separate domains in the head region, while *Cri-eve* is expressed in six stripes [Fig. 1k, l; note the seventh *Cri-eve* stripe comes up at the posterior pole only after the onset of gastrulation, similar to what has been previously described⁵⁴ for the moth midge *Clogmia albipunctata* (*C. albipunctata*; Fig. 1a, Psychodidae)]. Importantly, the *Cri-btd* and *Cri-eve* expression patterns do not overlap (Fig. 1k, l), as we can see a gap of 1-2 nuclei between their expression domains. These data suggest non-cyclorrhaphan flies lack the positional code necessary for CF initiation.

2) I. 337-339: 2 phenotypic classes of InscOE are defined: Class I (40% of embryos) with near-complete loss of buckles, Class II (60% of embryos) with head-trunk buckling. These classes seem to have been subjectively recognized and classified. I think the degree of buckling could be better quantified, for example by quantifying tortuosity (as in Vellutini et al.) This might reveal whether there is a small reduction of buckling (hard to appreciate by eye) even in “class I” embryos, for example.

We thank the referee for the suggestion to re-assess the classification of phenotypes in *insc* overexpressed embryos with *btd* RNAi knockdown (InscOE+*btd*RNAi). First, to increase the sample size, we have repeated the experiment. Then, to extend and improve our analysis, we quantified the depth of buckling as well as the persistence of buckling over time.

Following these analyses, we observed two classes of phenotypes, as described in the original manuscript. Specifically, one class of embryos shows a near complete loss of buckles, which is what we called Class I embryos. The other class of embryos (Class II) does show notable buckles. When compared to the control embryos with *btd* RNAi, these buckles have a reduced depth, although the persistence appears comparable. These data are presented in an updated main Fig. 4 (Fig. 4n, o) as well as supplemental Movie S4 (Fig 4 Movie S4).

To reflect the more specific findings resulting from our extended analysis in the main text, we refer to the updated figure in the revised text as follows:

(P11, L376-381): [...], we examined the effect of Insc^{OE} in embryos in which the CF is blocked by *btd* RNAi. We found two classes of phenotypes: 7 out of 13 (54%) of the embryos (Class I) showed a complete loss of head-trunk buckle; 6 out of 13 (46%) of embryos (Class II) undergo head-trunk buckling with shallower buckles than those seen in *btd* RNAi alone (Fig. 4m-o; Fig. 4 Movie S4). Embryos with the Class I phenotype tend to form smaller buckles either in the region between MD1 and MD5 or a deeper anterior dorsal fold, or both (Fig. 4 Movie S4).

3) I. 369-372: “our phylogenetic mapping reveals an abrupt and near concurrent emergence of the two morphogenetic traits: predominantly in plane division for the early mitotic cells in the head and active out-of-plane deformation at the head-trunk boundary”. This is true for the cephalic furrow, but unclear for in-plane divisions based on the data presented. Outside Cyclorrhapha, the 2 genera studied (*Clogmia* and *Chironoma*) belong to 2 relatively closely related families, and thus are part of a monophyletic group separated from Cyclorrhapha by 3 intermediate branches without cephalic furrow, but for which division patterns in the head (in-plane or out-of-plane) are unknown. To make any statement about the origin of in-plane divisions, some of these branches would need to be characterized. Otherwise, a lot of other situations are possible: intermediate branches (and the last common ancestor of flies) might show in-plane divisions, or might lack mitotic domains in the head, or mitotic domains might exist but not be active at the same time as germ band extension (thus preventing tissue collision), etc.

We are grateful for the referee’s comment – it motivated us to substantially expand our sampling of species among non-cyclorrhaphan flies, with a particular focus on closing the outlined gap between families outside of Neodiptera (here: *Clogmia albipunctata* and *Chironomus riparius*) and those constituting the early diverging branches of Cyclorrhapha (here: *Megaselia abdita*). This group of non-cyclorrhaphan Neodiptera is characterized by fly families in four monophyletic branches, i.e., (1) Cecidomyiidae, Sciaridae, Keroplatidae, Bibionidae, Scatopsidae, Anisopodidae, (2) Tabanidae, (3) Therevidae, Asilidae, Bombyliidae, Acroceridae, Stratiomyidae, Xylophagidae, and (4) Empididae. We extended our analysis to include the monophyletic branches 1, 3, and 4, and analyzed species of the family of Scatopsidae (*Coboldia fuscipes*), Stratiomyidae (*Hermetia illucens*), and Empididae (*Empis* sp. and *Tachypeza nubila*).

To obtain specimen *H. illucens*, we were able to use a recently established lab culture. For *C. fuscipes*, *Empis* sp. and *T. nubila*, lab cultures were not available, and individuals were caught as adult flies from the wild. For *C. fuscipes*, these adults could be used to establish a transient lab culture. For *T. nubila*, we relied on repeatedly catching adults, first from hills near Heidelberg, and then the botanical garden of Hohenheim University, while *Empis* sp. adults were caught during our excursion to Pula (Croatia), which severely limited the number of embryos available for analysis.

To determine whether or not a CF was formed in embryos of *C. fuscipes*, *H. illucens*, *Empis* sp., and *T. nubila*, we used DIC optics timelapse recordings. In none of the recordings (*C. fuscipes*, n = 6; *H. illucens*, n = 21; *Empis* sp., n = 3) we could observe CF formation (updated Fig. 1 Supplement 1). In *T. nubila*, absence of CF could be confirmed up until the onset of gastrulation; after that, observation of embryonic development using DIC optics was made difficult as the embryos turned opaque after the onset of morphogenetic movements (n=7). Taken together, our new analyses confirm that non-cyclorrhaphan flies do not form a CF.

To determine the plane of divisions in the head, embryos were fixed and stained using DAPI or DRAQ5. For *C. fuscipes* and *H. illucens*, the two species for which we could establish at least a transient lab culture, sufficient embryos were available to sample stages at the onset of gastrulation and test for the presence of out-of-plane divisions. Unfortunately for *Empis* sp., we were limited by our time during the Pula excursion, and could not find enough

adults to get embryos for fixation and staining. For *T. nubila*, we could only fix a small number of embryos. These embryos unfortunately were too young, as we could not determine the developmental stages from the earlier time lapse recordings, as the embryos turned opaque. This severely prevents us from assaying the orientation of head divisions. Nonetheless, our extended sampling has allowed us to better assess whether the absence of CF formation consistently coincides with out-of-plane divisions in the head across non-cyclorrhaphan flies. We were able to confirm out-of-plane divisions in the head of *C. fuscipes* and *H. illucens*, thereby adding two more branches in the dipteran phylogeny that suggest that out-of-plane division in the head represents a common strategy for the non-cyclorrhaphan flies to avoid tissue collisions at the head-trunk boundary in place of CF formation.

Taken together, our extended sampling improves our phylogenetic coverage to better support our claims. For the newly sampled branches, we have now included additional panels in Figure 1 (Fig. 1e, f) with images showing the absence of CF during early gastrulation in fixed embryos, along with additional panels in Fig. 1 Supplement 1 with montages of timelapse recordings. We have also updated Fig. 1 Supplementary Table 1 to indicate additional evidence for the absence of CF in non-Cyclorrhapha and included additional panels in Fig. 4a showing images indicative of out-of-plane dividing nuclei.

To include the information on extended species sampling, we added the following section to “Experimental Animals” in the Methods parts:

(P48, L888-903): The laboratory cultures of *H. illucens* were established from existing cultures in the Schmidt-Ott lab (University of Chicago, USA). To collect *H. illucens* embryos, females were decapitated to trigger egg laying at the desired time.

A transient lab culture of *C. fuscipes* was established from wild-caught adults found near municipal compost plant of the city of Heidelberg. *C. fuscipes* adults prefer to lay eggs in small cavities. To collect embryos, old culture plates of *C. albipunctata* were used as they conveniently have such cavities. Before using these for *C. fuscipes* egg collection, the plates were decontaminated by first freezing them at -20°C for at least a week, followed by thawing them overnight at room temperature.

We could not establish a lab culture for any of the species from the family Empididae, as we couldn't optimise the culture conditions. During our excursion in Pula (Croatia) we managed to catch a few adults of an *Empis* species (most likely *Empis pennipes*, referred to as *Empis* sp. throughout the text). As a result, we resorted to repeatedly catching adults during that time from the wild, and then decapitated the females to trigger egg laying. We managed to catch another fly species from the same family (*Tachypeza nubila*) on the hills near Heidelberg and then in the botanical garden of Hohenheim University. However, these embryos turn opaque during early gastrulation, precluding us from analyzing them properly (data not shown).

Formatting suggestion: Fig. 1a: could the species be mapped onto the phylogeny (eg by writing (c), (e), etc next to the family names at the tips of the tree)

This is an excellent suggestion! Especially now with increased sampling, the figure benefits from an indication of which branches have been sampled, and where to find the

corresponding data panels in the figure. The revised version of Fig. 1a has been changed accordingly. The figure legend for panel a now reflects these changes as follows:

(P15, L481-483): Highlights indicate families for which species were evaluated (light) and studied (dark) in this work. Letters in parentheses refer to panels on the right side of this figure.

Referee #2 (Remarks to the Author):

Summary

This MS examines a specific morphogenetic movement that occurs in some dipteran (flies & mosquitoes) embryos: the formation of a temporary trough of cells called the cephalic furrow. The formation and subsequent disappearance of this furrow has been well known from studies of Drosophila embryogenesis for decades, but it appeared to have no developmental function, as the structure is temporary and the cells that form it do not appear to adopt any specific fate that required the furrow to form. In this MS the authors perform a number of perturbations of furrow formation, and find that when they prevent it from forming, other, smaller and less regular furrows form in other regions of the embryo. These irregular ectopic furrows are also temporary.

We thank the referee for this succinct summary.

Major Comments

1. The hypothesis that formation of the cephalic furrow confers a fitness advantage and is therefore under selective pressure is at the heart of the paper, because the authors are claiming to have solved the problem of “why” the furrow forms. However, the evidence for this is weak. The two lines of evidence presented are (1) observations of “increased frequency of midline distortion or rotation” in embryos in which furrow formation was blocked with their optogenetic approach (Figure 3). However, there is no evidence provided that midline distortion results in decreased frequency of hatching or in worse outcomes at post-embryonic development, neither in this optogenetic experiment nor in any of the other genetic manipulations performed in the MS to compromise CF formation; (2) the assertion that because the furrow forms precisely and robustly in wild type embryos, it must confer fitness (page 7). This is an example of a very common yet incorrect application of the theory of evolution by natural selection: the mere existence of a biological trait is not evidence that it evolved due to natural selection. Rather, natural selection is just one of multiple different potential mechanisms of evolutionary change. Without compelling evidence for any biological function or evolutionary relevance of this morphogenetic process, the paper remains a well-written and well-presented comparative quantitative developmental biology manuscript showing interesting differences in cell behaviours between dipterans with and without cephalic furrows, but it does not rise to an unusual level of general interest.

We thank the referee for highlighting these weak spots in our reasoning. Their excellent comments motivated us to come up with a new set of experiments and clarify our writing,

both of which we believe significantly improved the manuscript and substantially strengthened our claim. Below we will address each of the two questions separately.

(1) We agree with the impression that the arguments presented thus far could be seen as a somewhat weak support for a fitness advantage conferred by CF formation. The association of increased frequency of midline distortion or rotation after blocking CF with a decrease in embryonic fitness in *Drosophila melanogaster* may be, e.g., questioned against the background that midline distortions and rotations have been reported for gastrulation movements in some non-*Drosophila* insects, which could be used to argue that the influence of midline distortions on embryonic fitness in *Drosophila* is not precisely defined.

To address such concerns, our first attempt was to use the hatching rate after optogenetic perturbations as a more accurate proxy for organismal fitness. The idea was that a significant decrease in fitness should be reflected in a lower hatching rate. With this approach, however, we faced technical limitations, as the specific genotype of the embryos required for our optogenetic perturbation had a rather low baseline hatching rate (approx 36%) even without photostimulation or imaging, which contrasts with a 91% hatching rate for wild-type embryos from Oregon-R when subject to the same experimental procedure. Such a low hatching rate can have diverse and unrelated reasons, so relying on them, one risks a bias that would be difficult to gauge.

To describe these initial benchmarking experiments, we added a dedicated paragraph in the “Methods” section as follows:

(P53, L1046-1056): To approximate the fitness of embryos that fail to form the CF, we initially planned to use the experimental design 3) described above to eliminate the CF, followed by scoring the embryonic hatching rate. This approach was abandoned after benchmarking experiments, where the opto-DNRho1 embryos were mounted following a standard protocol – dechorionated, affixed to a glass bottom dish with the heptane glue, and immersed under a mixture of Halocarbon oil 700 and 27 in the ratio 3:1 – and kept in a moisturized chamber at 22°C in the dark for 24-48 hours without imaging or any photostimulation, showed a low hatching rate (36%; number of trials =6; 5 embryos/trial), contrasting with wild-type (Oregon R) embryos that scored a 91% hatching rate (number of trials =4; 10 embryos/trial). These data suggest that the currently available tool for optogenetic inhibition of the CF, i.e. the flies that carry the opto-DNRho1 system, has high background lethality even without photostimulation, thus unsuitable for the evaluation of embryonic lethality following CF inhibition.

To overcome these technical limitations, we decided to perform long-term imaging experiments to monitor the complete course of embryonic development following a 1-hour optogenetic perturbation that blocks CF formation at the onset of gastrulation. We were able to image these embryos for at least 18 hours and achieved a sizable dataset both for the sham controls that form the CF normally (n=35) and the photostimulated embryos that fail to form the CF (n=25).

In this dataset, we observed two late-stage defects with high frequency, in addition to the midline distortion that we reported previously. Specifically, embryos whose CF was blocked during the onset of gastrulation showed a much higher frequency of complete or partial failure of head involution. There is also a higher frequency of incomplete or disrupted or splayed ventral nerve cord during its condensation. As has been extensively documented before (reviewed, e.g., by VanHook and Letsou (2008) and Karkali and Martín-Blanco

(2024), see also below in text quoted from the revised manuscript), head involution ensures proper internalization of embryonic head segments that include several vital organs such as the mouth, while proper ventral nerve cord condensation is necessary for the assembly and wiring of the central nervous system. Their defects in embryos without CF formation, therefore, provide unequivocal evidence that loss of CF has deleterious effects on embryonic development.

To better understand how these late developmental phenotypes compared to earlier midline distortions as a proxy for how strongly development (and, by extension, embryonic fitness) was affected by perturbation of CF formation, we collectively analyzed the effects of optogenetic CF perturbation on midline distortions, ventral nerve cord condensation, and head involution. Our results demonstrate that using midline distortion substantially underestimated the effects of CF abrogation on embryonic development and embryonic fitness, indicating that the effects of CF loss worsen as development proceeds, and midline distortion is too early and possibly too conservative as a measure to accurately report any fitness consequence of the CF loss. Uniquely, we observed increased co-occurrence of head involution and ventral nerve cord defects in the photoactivated embryos, suggesting that they are linked. At the moment, we do not have further mechanistic insights into how the lack of CF formation leads to these late-stage defects. However, given their high co-occurrence, which we intend to follow up on in our future studies, we propose that these phenotypes are strong indicators that suggest that the loss of CF formation reduces organismal fitness.

Our new and extended analyses are presented in Fig. 3d-f, Fig. 3 Movie S2 and described in the main text as follows:

(P8-9, L274-290): Although midline distortions are a clear deviation from normal embryonic development in *D. melanogaster*, we could not be certain that they have a deleterious effect on organismal fitness. Unfortunately, technical difficulties prevent us from assessing embryonic survival as a direct measure of fitness (see Methods). As a proxy, we sought to examine whether CF loss causes further, possibly more severe, developmental abnormalities that could be detrimental to the function or viability of the organism. We imaged the embryos beyond the normal timeframe of CF formation and retraction, and monitored their development for at least 18 hours after optogenetic inhibition of CF formation at the onset of gastrulation. Strikingly, in addition to ventral midline distortions, at later stages we observed an increased frequency of head involution (HI) and ventral nerve cord (VNC) defects (Fig. 3d-f, Fig. 3 Movie S2). While we observed a substantial increase of the co-occurrence of HI and VNC defects, these occur independently of ventral midline distortions (VF), suggesting that they represent a separate phenotypic repertoire (Fig. 3 Supplement 1f, g). These late defects suggest that embryonic abnormalities following CF loss extend far beyond the gastrulation stages, while focusing on early gastrulation processes substantially underestimates the effects of CF abrogation on embryonic development. As HI ensures proper internalization of embryonic head segments that include several vital organs such as the mouth^{30,31}, while proper VNC condensation is necessary for the assembly and wiring of the central nervous system³², these data suggest that loss of the CF has deleterious effects on fitness.

We discuss the implication of these data in Discussion.

(P13, L445-451): These data highlight the possibility that CF might have evolved for a function related to head morphogenesis, thus accounting for the severe defects in HI

following optogenetic ablation of the CF. Alternatively, the wide-ranging phenotypic consequences associated with CF ablation, including defects in HI and VNC, may be attributed to the improper release of compressive stress or deleterious tissue deformation resulting from mechanical instabilities, despite that these phenotypes manifest in organogenesis processes that occur hours after the normal timing of CF retraction.

(2) We agree that the mere existence of a trait is not sufficient evidence to conclude that it evolved via natural selection or that it confers fitness, and we acknowledge the need to clarify our reasoning regarding the evolutionary significance of CF formation.

We obviously failed to sufficiently explain the argument we would like to make about the evolution of CF formation. Our argument is actually based not on the existence of the CF as a trait as such. Instead, we would like to discuss CF evolution in light of its conservation, robustness, and apparent irreversibility across independent lineages. The evidence we present suggests that CF formation is maintained by stabilizing selection.

- The CF forms precisely and robustly across wild-type embryos, as supported by previous literature (#9 - Eritano et al., (2020); #14 - Liu et al., (2013)) as well as our own observations (Fig. 1, Fig. 1 Supplement 1, 2). This is consistent with a trait that is subject to stabilizing selection to maintain its optimal functionality.
- The CF appears in multiple independent lineages, i.e., in two paraphyletic non-schizophoran branches as well as three major, paraphyletic schizophoran branches (Fig. 1, Fig. 1 Supplementary table 1), indicating that it is an evolutionarily conserved feature.
- Within the limits of our sampling, the CF has not been lost since it originated in the stem group of Cyclorrhapha about 150 million years ago (6/6 sampled cyclorrhaphan species form a CF), which argues against drift or neutral evolution, as traits without functional significance tend to drift in their properties or be lost over such evolutionary time scales.

Taken together, we interpret this combination of conservation, robustness, and irreversibility as indicative of ongoing stabilizing selection. Stabilizing selection, by its nature, suggests that the trait confers a fitness advantage, as it minimizes deviations from an optimal phenotype.

While our interpretation aligns with evolutionary theory, our evidence may not allow us to formally exclude other potential mechanisms, such as phylogenetic inertia or indirect selection. However, we consider them less parsimonious for the following reasons:

Phylogenetic inertia is maintaining a trait without strong selection that acts against the loss of it. We have rejected this possibility because then we would expect to see at least some variation or an occasional loss of CF formation. Traits that are selectively neutral or weakly constrained tend to accumulate variation or be lost over time. The fact that CF formation remains precise and robust argues against pure phylogenetic inertia.

Linked selection would mean that CF is maintained because of genetic linkage (maintained synteny) of its genetic regulation to other key genes. We considered this scenario unlikely because Diptera have been characterized by rapidly deteriorating synteny due to a high rate of genome reshuffling.

To address this issue and improve clarity, we have now revised the manuscript to ensure that our argument is framed explicitly around the evidence for stabilizing selection and not the mere existence of the trait.

The corresponding section now reads:

(P8, L254-257): The precision and robustness with which the CF is formed^{9,14}, combined with its deep evolutionary conservation across independent lineages and the absence of documented losses since its origin in the stem group of Cyclorrhapha, suggests that its spatial patterning and temporal dynamics are subject to stabilizing selection. This suggests that CF formation provides a fitness advantage.

Related, on line 12 the subjective description of their finding that the CF is an evolutionary novelty as "surprising" and of transitions around CF presence as "seemingly abrupt" should be removed. This is related also to minor comment #6.

Thank you. We have removed these expressions.

2. The sample size is very low (5 or fewer) for most of the experiments in this paper. There is no data presented to address the degree of wild type variance in CF or head-trunk buckle (ectopic fold) formation, so it is impossible to know what sample size would be necessary whether ectopic folds form more often than expected in cases of CF abrogation, but if it is indeed so rare that a sample size of 3 (as presented for many genetic conditions herein) is sufficient, then this evidence should be provided from assessments of wild type embryos, which in this laboratory system are easy to obtain in large numbers.

We thank the referee for this observation. The invariance of wild-type CF formation in *Drosophila* is well-documented in the literature, including Ref#9 and Ref#14 (Eritano et al., (2020); Liu et al., (2013)), where hundreds of embryos were imaged and analyzed. Given the high consistency (100%) of CF formation, we were confident in the phenotypes with genetic or optogenetic manipulation that we reported in the manuscript despite low sample sizes.

We nonetheless repeated all experiments involving *btd* RNAi injection in wild-type as well as mutants that disrupt mitosis and germband extension. All newly performed experiments consistently confirmed our previous findings. These efforts have allowed us to report much higher sample sizes in this updated version of the manuscript.

We also note that in embryos that were genetically or optogenetically manipulated to block CF formation, we consistently observed the formation of head-trunk buckles. To reflect the fully penetrant nature of the phenotype, we are now using the term "fully penetrant head-trunk buckle" (P5, L155).

3. In Figure 1a, 42 families (of the nearly 200 in Diptera) are shown in the tree, only 12 of which appear to have available data on CF furrow formation. Why were these 42 chosen for display when most of them lack data on CF formation, and why are there 13 families (rather than 12) shown in the supplementary table (Fig1 Supp 4). The claim

in the figure legend that once the CF appeared it was not secondarily lost, is not well supported given that there are so many missing data points on the tree.

We thank the referee for this observation. The tree shows all major fly families and has been published as such by Wiegmann et al. 2011, who is essentially the world leading authority on the dipteran phylogeny. By showing this tree rather than one with all fly families, or one with only the families that have available literature references, we aimed to strike a justified balance between comprehensiveness and reasonable data representability. Showing the major families, marked with the respective character states, also reveals that the branches we sampled scatter across the tree and are not clustered. Even if absolute certainty in phylogenetics is rarely achieved because comprehensive sampling is often infeasible, we feel that the shown tree is a truthful and unfiltered representation of the current state of our sampling.

While we decided to use the Wiegmann tree with only major families for representation purposes, we did find some classic literature for 'Simuliidae', a family that presumably has once been of interest due to its hematophagous behavior. Despite not being a major fly family, we included it in the supplementary table for the sake of completion.

The statement about loss of CF formation has been removed; the legend for panel a now reads:

(P15, L479-481): Phylogenetic tree showing major fly families (adapted from Wiegmann et al., 2011) marking the presence (filled boxes) or absence (empty boxes) of a CF in embryos belonging to various fly species, indicating that the CF appeared in the stemgroup of *Cyclorrhapha* (arrow).

Minor Comments

We thank the referee for these minor comments. Below is our point-by-point response.

1. Please refer to species throughout with the Linnean abbreviation (e.g. *C. riparius*) rather than with the genus name (e.g. *Chironomus*)

All species have been changed to the Linnean abbreviation, except in places where processes common to a given lineage were mentioned, e.g., *Drosophila* gastrulation.

2. The concept of "tissue tectonics" (page 6) has been used in other examples of similar epithelial buckling and furrowing events during development (e.g. Blanchard et al 2009 Nature Methods; Busby & Steventon 2021 Interface Focus; Adams, 2011 FASEB Journal). It would be useful for the authors to contextualize their observations and use of this term, with those of others - are the authors adding something new to this concept, or is this pre-existing concept a useful and accurate descriptor of the phenomena they report here?

We thank the referee for pointing this out. We have now referenced the previous studies where the term 'tissue tectonics' was introduced. We have additionally explained and elaborated how our use of the term expands on the existing concepts. The texts associated reads as follows:

(P6, L202-211): Based on these analyses, we propose that buckling at the head-trunk boundary arises from 'tissue tectonic collision' driven by the converging flows of expanding head and trunk 'tissue plates'. In the absence of the genetically programmed mechanical sink, i.e. the CF, tissue collision gives rise to increased compressive stress, thereby resulting in buckling. Note the term 'tissue tectonics' was previously introduced to describe in-plane epithelial deformation via cell shape changes and rearrangements (Blanchard et al 2009) and higher-level regulation of developmental timing through spatial displacement of signalling and responding tissues (Busby and Steventon 2021). Our adoption of the term expands its use to describe epithelial out-of-plane deformation resulting from tissue collision, drawing on an analogy to Earth's tectonic plates, which collide at their convergent boundaries to form mountain ranges and deep-sea trenches.

Note we omitted Adams 2011 citation as it is redundant with Blanchard et al 2009.

3. page 9 lines 299-300: the meaning of "an in-plane dividing on average cell" is unclear.

We thank the referee for pointing out this typo. This has been corrected and it reads:

(P10, L337-338): [...] an in-plane dividing cell, on average, reaches an apical area [...]

4. page 10 line 333: the meaning of "qualitatively comparable" is ambiguous - please clarify.

We thank the referee for the request. We have removed the ambiguous expression, given that quantitative data were already provided. The sentence has now been rephrased.

(P11, L370-372): For both MD1 and MD5, Insc^{OE} results in a reduced area expansion, i.e. from 2-fold to 1.7-fold in MD1 (Fig. 4j, l) and from 2.8-fold to 2-fold in MD5 (Fig. 4k, l), thus similar to the reduction of area expansion observed in *C. riparius* MD2o, when compared to MD2i.

Additionally, to further increase the strength of this quantitative dataset, as well as the corresponding dataset in *C. riparius*, we also increased the number of embryos analyzed from 3 to 6, and 3 to 5, respectively and performed statistical tests. We updated Figure 4 (Fig. 4e, f, j-l) and Figure 4 Supplement 3 (Fig. 4 Supplement 3b, c) correspondingly.

5. page 11 line 372: please avoid referring to branches whose tips are extant organisms as "basal." "Early diverging" would be more appropriate. Related, on page 12 lines 408-409, no single species represents any other; please rephrase as "species belonging to early diverging branches" or similar.

Thank you for pointing out imprecisions in our phrasing. We have replaced the occurrences of 'basal' with 'early diverging' and also aimed to revise instances in which our wording may have been interpreted as conflating a single species with the entire lineage it belongs to.

E.g. (P12, L411-412): Notably, species belonging to the earliest-diverging branches of Cyclorhapha, such as *M. abdita*, contain both of these traits, raising the intriguing question regarding which of two traits evolved first.

We hope that this phrasing avoids the implication that a single species "stands in for" an entire lineage while still highlighting that it shares traits associated with early-diverging branches.

6. page 11 lines 375 - 377: This sentence appears to state a hypothesis but is phrased as an assertion - please state the fact that this is not known, is more directly, for example "we speculate that," rather than using "likely" without explaining why the authors believe this is more likely than some other outcome of an early in-plane division origin. Also in this sentence, "stronger" than what? The same applies to the following sentence and to others in this paragraph - there is no apparent basis for claiming one scenario to be more likely than another; the authors can simply state that this is their speculation, and ideally explain the evidence-based logic behind favoring these hypotheses over others.

We thank the referee for pointing out the need to better indicate that we are stating a hypothesis. We have rephrased the sentence accordingly, which now reads:

(P12, L413-415): In the first of the two hypothetical scenarios, where in-plane division arose first, the increased spatial demand and the prolonged surface expansion associated with in-plane division would have exerted strong compressive stress on the head-trunk boundary, increasing the likelihood that it would buckle.

Also in the following paragraph, we now clearly state that we are presenting a hypothesis. It now starts as follows:

(P13, L432-435): In the second hypothetical scenario, where the CF evolved first, arising prior to the conversion of predominant out-of-plane divisions to in-plane divisions in the head, compressive stresses in the head may have been initially low or absent, and the CF may have arisen for a different, currently unknown function.

7. Figure 1l: where is the 7th eve stripe?

Thank you for pointing out the need for further explanations. *C. riparius* shows the 7th stripe a little later than *Drosophila*, i.e., after the onset of gastrulation. The revised version of the manuscript now contains images for *C. riparius* at stages before and after the onset of gastrulation to demonstrate the expression domains of *btd* and *eve* (Fig. 1k, l) as evidenced by HCR-based *in-situ* hybridization. And, even though that's not the main point, these images also illustrate the appearance of the 7th *eve* stripe during early gastrulation.

(P3, L93-97): In the blastoderm stage, *Cri-btd* is expressed in two separate domains in the head region, while *Cri-eve* is expressed in six stripes [Fig. 1l, k; note the seventh *Cri-eve* stripe comes up at the posterior pole only after the onset of gastrulation, similar to what has been previously described for the moth midge *Clogmia albipunctata* (*C. albipunctata*; Fig. 1a, Psychodidae)].

8. Figure 3e,f: what proportion of the total midline length is the deviation shown here?

In the revised manuscript, these plots are now presented in Figure 3c and Figure 3 Supplement 3c. As now mentioned in the methods section, the entire midline length visible

on the ventral side is considered. The plotted deviation in μm is the mean distance from the actual midline, along the entire length of the midline, to the expected midline position estimated based on a linear fit to the midline positions observed at the onset of gastrulation. To further clarify the text, the method section now reads as follows.

(P56, L1158-1167): For an estimate of deviation from the expected linear ventral midline, the ventral midlines visible on the ventral side were marked manually using the segmented line tool in FIJI at the onset of gastrulation ('onset') and a later time point ('mid') when the midline exhibits maximal distortion. The resultant ROIs were converted into points using built-in macro function `Roi.getContainedPoints` in FIJI, and the coordinates were analyzed using a Python code. To compute deviation, a linear fit to the midline coordinate was first generated as the expected midline for the onset stage, which was then translated to the linear fit for the midline coordinates marked at the 'mid' stage, while preserving the slope, as the expected midline for the mid stage. For each stage, the mean distance from the marked midline coordinates to the expected midline was calculated and plotted as 'Deviation' in Fig. 3c, f.

9. Figure 1 supplement 3: as these trees are unrooted, the evolutionary relationships of the *C. riparius* eve and btd candidate sequences are not convincing. The in situ hybridization patterns lend support to the author's homology hypothesis for these sequences but these trees do not help - rooting the trees would help this.

Thank you for pointing out the need to include an outgroup: we have generated new trees (Figure 1 supplement 3) that now include orthologs of *btd* and *eve* from *Tribolium castaneum* (Coleoptera) to support the homology of the *C. riparius* eve and *btd* sequences.

Comments on the relationship of this MS to the accompanying Vellutini et al. MS

1. Dey et al point out that using eve or btd mutants to study the cephalic furrow is problematic because mutations in both genes cause severe patterning defects beyond the elimination of the furrow (page 4). However, this severe limitation is not acknowledged by Vellutini et al., who do use this approach.

We thank the referee for this comment and we agree this needs further clarification. Our argument that using *eve* or *btd* mutants is problematic has actually been made with the question in mind of CF function and the potential impact on embryonic fitness. If CF formation is abrogated by loss-of-function mutants or knockdown of *eve* or *btd*, the interpretation of late gastrulation phenotypes or head development is complicated because it is not possible to ascertain whether the resultant phenotypes are due to defective patterning associated with the loss of *eve* or *btd*, or due to the loss of cell shape changes and tissue deformation that normally lead to CF formation.

For studying the immediate effects of tissue collision at the head trunk boundary, which in the absence of CF furrow formation results in epithelial buckling, the use of *eve* or *btd* mutants or knockdown is less problematic. This is illustrated in the phenotypic analysis in Fig. 2a-i, where we showed that with respect to head-trunk buckling, the phenotypes of *eve1KO*, *btd*, *eve*, and optogenetic inhibition of CF formation are all comparable.

This compatibility in phenotypes, in our view, perfectly justifies the use of *eve* and *btd* mutants to study the short-term phenotypic effect of loss of CF in the context of early gastrulation. It also justifies the use of *btd* mutants or RNAi as a means to block CF formation in conjunction with other genetic manipulations, an approach we undertook in Fig. 2k-o, and 4j-o, where it has been technically challenging to combine *eve1KO* or optogenetic inhibition of CF with other genetic manipulations.

Vellutini et al. should probably present their own view on this, but we feel that it is well justified to use *eve* and *btd* mutants to uncover the immediate, short-term phenotypic consequence of an increase of mechanical instabilities in the early gastrula when the CF is not formed - in particular in light of our results with *eve1KO* and optogenetic inhibition. Vellutini et al went on to use *eve* and *btd* mutants to show that reducing compressive stress can ameliorate mechanical instabilities, which again is well justified in the context of investigating the cause of such short-term mechanical instabilities. Furthermore, both studies reach similar conclusions with respect to the function of the CF in releasing compressive stress further validating our use and description of the *eve* and *btd* mutants.

In regard to analysing the long-term phenotypic effect and fitness consequence of the loss of CF, given that both *eve* and *btd* are expressed in a large number of cells and throughout all stages of embryonic development, we do feel that the phenotypic consequences of *eve* and *btd* mutants include effects that reflect expression and function unrelated to the specification of the CF. In fact, we observed widespread cell death even in the *eve1KO* embryos, when imaging them for about 16-18 hrs to assess long-term phenotypic effects. Thus, the use of *eve* and *btd* mutants could complicate the analysis of phenotypes associated with the loss of the CF, especially those that occur at late developmental stages, and the assessment of the ultimate fitness consequence of CF loss. Therefore, in Fig. 3 where we explored the fitness consequences of loss of CF, we based our analysis primarily on embryos with optogenetic inhibition of CF.

2. Dey et al. use the terminology “head/trunk buckles” (page 4) while Vellutini et al. call them “ectopic folds.” As these MSS are clearly intended to be companion pieces examining the same biological problem, it would be better to use consistent terminology in both.

We thank the referee for this suggestion - given how the two manuscripts converged on the same fundamental mechanism of mechanical instabilities, we agree that consistent terminologies should be used wherever applicable. However, we could not convince ourselves that this would do justice to the minor, but notable differences in the phenomena reported in each of the manuscripts. In general, it is our impression that it will be more honest, and in fact intellectually beneficial, to maintain the side-by-side reporting of slightly deviating findings with slightly different terminology, as they could provide crucial insights into specific aspects of this general phenomenon of mechanical instabilities. Fully unifying the terminologies may risk losing these detailed nuances that are important not only for the understanding of the phenomenon, but also for future studies of this and related processes in other contexts.

Specifically, when we use the term “head/trunk buckle” to describe tissue dynamics and topology in the absence of CF formation, we refer to the one deep epithelial deformation that we observed at the head/trunk boundary precisely at the position where the CF would form

by molecular patterning and cell-mechanical refinement. This is the phenotype we have mainly focused on in our analyses. In addition to the head-trunk buckle, we did observe additional, smaller, and more variable deformations in *btd* or *eve* mutant or *btd RNAi+ InscOE* embryos (Fig. 3 Supplement 2 d, Fig. 4 Movie S4). To describe these, we write:

(P5, L153-158): In addition to the head-trunk buckle, we observed buckling-like deformations elsewhere in the head region (Fig. 2 Supplement 2d), consistent with data reported in Vellutini et al. These deformations are more variable and occur at lower frequencies, contrasting with the fully penetrant head-trunk buckle. All of these buckles are likely manifestations of compressive stress induced mechanical instabilities (see below). In the following analysis, however, we will focus primarily on the head-trunk buckle to reveal the functional role and evolutionary origin of the CF.

Mechanical instabilities, such as buckling, are inherently stochastic and highly sensitive to small mechanical inhomogeneities. This sensitivity likely contributes to the variability in the number and position of buckles observed in our study and in Vellutini et al. However, while mechanical inhomogeneities may influence where small buckles emerge, the primary driver of increased compressive stress in our system is not stochastic noise; rather, it is the spatially patterned tissue collision between the expanding head and trunk, as revealed in our tissue flow field analysis (Fig. 2j).

This interpretation is strengthened by theoretical considerations. In a scenario where uniaxial stress is applied to a homogeneous tissue sheet with all but intrinsic mechanical fluctuations, one would expect uniformly undulating and variable buckling patterns. In contrast, our findings reveal a dominant buckle forming consistently at the head-trunk boundary, suggesting that its position is not purely determined by random inhomogeneities. Instead, the frequent emergence of buckling precisely at this boundary suggests an additional influence: pre-existing mechanical patterning. Indeed, prior studies have shown that compression-driven instabilities can appear spatially organized when guided by patterned mechanical cues (Ref#45 and #46).

To explain these conceptual ideas, we now include the following paragraph in Discussion:

(P13, L455-463): The buckling transitions reported here and in Vellutini et al. when the CF is ablated add to a growing list of epithelial morphogenesis events resulting from mechanical instabilities⁴⁴. Such instabilities are intrinsically stochastic and sensitive to small perturbations, which may explain the variability in the number and positions of buckles observed in our study and that of Vellutini et al.. However, spatially patterned mechanical information is also known to influence where buckling occurs^{45,46}. The frequent emergence of buckling at the head-trunk boundary suggests that its position is not random, but rather shaped by the spatially patterned ‘tissue tectonic collision’ between the expanding head and trunk tissues. Thus, the CF likely evolved in alignment with this preexisting mechanical patterning, providing a solution to mitigate tissue collision during the evolution of dipteran gastrulation (Fig. 5).

While we remain open to the possibility of streamlining terminology between the two manuscripts, so far we have preferred the more honest option of maintaining the different terms.

3. Dey et al. observe late onset of ectopic folds in *stg* mutant embryos (their class I phenotype) but this is not reported in Vellutini et al., who examined embryos of the same genotype. Please explain this discrepancy.

We thank the referee for pointing this out. Indeed, there is a discrepancy in the data when compared to Vellutini et al.: Vellutini et al. used double mutants of *btd* and *stg* for this experiment, while we used *btd* double-strand RNA injections into the *stg* mutant embryos to create the double loss-of-function condition. After repeating these experiments several times, including efforts made previously and more recently during our revision, we consistently observed that the severity of *btd* knockdown shows variabilities, despite increasing the concentration of the *btd* double-strand RNA to what is technically possible in our hands. Therefore, either our results are related to the incomplete knockdown of the RNAi methodology, or the genetic background of the double mutant flies that Vellutini et al. used has additional inhibitory effects on buckling, or the subtle differences in phenotype stem from differences related to imaging modalities and respective sample preparation (inverted confocal microscope vs. light sheet microscope). We have not been able to further resolve this because we were not able to generate the *btd stg* double mutant stocks that contain the necessary fluorescent markers for genotyping and imaging. We have added the following lines in the methods section to comment on these discrepancies:

(P51, L976-983): A note on the efficacy and variability of phenotypic effects associated with the *btd* dsRNA injection: The efficacy of all *btd* dsRNA preps were verified based on phenotypes. Specifically, the blocking of CF formation and the resultant occurrence of head-trunk buckling were comparable to those observed in the *btd^{AX}* homozygous embryos. When combined with *stg* mutant or *InscOE*, we consistently observed two classes of phenotypes (Fig. 2k, l and Fig. 4n, o) despite raising the concentrations of *btd* dsRNA by ~10 fold. Technical difficulties prevented us from executing these experiments using the *btd^{AX}* mutant to resolve differences between our data and those of Vellutini et al. regarding *btd stg* double loss of function.

Referee #3 (Remarks to the Author):

This manuscript describes a multipronged approach to the structure, and in particular the function, of the Cephalic Furrow (CF), a seemingly unimportant multicellular pattern that arises, transiently, in the anterior third of the early *Drosophila* embryo. As it happens the study turns out a solid and original piece of work on the role of mechanics in pattern formation and the evolution of the mechanisms adopted by constrained cellular fields to cope with mechanical stresses during morphogenesis. It is an original and insightful piece of work that takes advantage of the detail in which one can probe mechanism in *Drosophila*, and extrapolates it to related insects, to understand the evolution of fundamental issues in developmental biology.

Dey et al start by looking in detail at the dynamics of the organization of the CF in *Drosophila* and probe it through a range of techniques and mutants and great attention to detail. The authors propose that in Cyclorrhaphan flies the emergence of the buttonhead head-trunk boundary expression domain, overlapping with the first stripe of even-skipped is a central event leading to the evolution of cephalic furrow formation in Diptera. In *Drosophila* they probe mechanical and morphogenetical instabilities that arise in the absence of cephalic CF formation using genetic and

optogenetic perturbations and sophisticated cell and tissue dynamics analysis. They observe cell divisions associated with this event and suggest that the role of the CF is to cope with stresses associated with the strong movements during early events of gastrulation in Cyclorrhaphan Diptera.

To go beyond the tests that they do, which involve a surgical removal of the CF through a rescue construct for eve that lacks the stripe 1 enhancer, they look at other insects, in particular non-Cyclorrhaphan flies that represent a different branch and lack CF and find that they have evolved a different mechanism to cope with the same challenges, in this case, out of plane divisions. In non-Cyclopharran flies (Chironomus) they characterised the orientation of head cell divisions and concomitant cell shape changes and explore the role of out-of-plane divisions in reducing compressive stress. They report that both mechanisms are present in a representative species of the most basal branches of Cyclopharran flies (Megaselia) leaving open the question on how these morphogenetic control mechanisms evolved.

The work is a rare piece integrating development, evolution and cell biology and provides a very detailed example of the role of mechanobiology in pattern formation which will serve as a basis to consider the evolution of developmental mechanisms.

It is difficult to find a major flaw, though it would have been good to hear if other transient folds that appear also during gastrulation have a similar function.

The manuscript is long which helps in the detail but this is maybe good.

We thank the referee for a comprehensive summary of our study.

The point raised about other transient folds serving similar purposes during gastrulation is a valid point and something that we are also interested to look into in the future. Dorsal folds, which are two transient folds that appear on the dorsal surface of the embryo a little later after the initiation of the cephalic furrow, are a good candidate to study this. However, at the moment there is a lack of spatiotemporally specific genetic tools to ablate dorsal folds specifically, nor is there an optogenetic method to precisely inhibit dorsal folds mechanically, which limits the possibility of making clear conclusions about it working as a tissue sink. We agree it could be helpful to comment on this and have now added a few sentences in Discussion.

(P14, L471-475): That the CF likely evolved for and performs a mechanical function raises the possibility that similar principles may apply to other transient epithelial folds, such as the dorsal folds in *Drosophila* gastrulation⁴⁸. It thus might be of interest to explore the possibility that mechanical stress management represents a general functional feature of transient epithelial folding across organisms and developmental contexts.

Response to referees

Below are our detailed responses to the referees' comments; to facilitate readability, we used the following font emphases/colors:

Comments from the referees

Our response to their comments

Changes to the manuscript

Referee #1

I am fully satisfied with the revision.

While I could have been content with the addition of a verbal caveat regarding the 'abrupt and near concurrent' emergence of in-plane divisions and cephalic furrow formation, the authors went above and beyond, by studying four additional fly species, including three that were collected de novo in the wild - now supporting their point with a much broader taxonomic sample. I also thank them for establishing HCR to conclusively rule out btd/eve co-expression in Chironomus, and performing more detailed quantification of phenotypes. Finally, I strongly appreciated the additional observations of head involution and ventral nerve cord condensation defects documented in response to Reviewer 2. I think the author's argument for functionality of the cephalic furrow (and for the specific function they hypothesise) is now compelling.

Given how lengthy, effort-intensive, and extensive the first revision was, I feel it would be unreasonable to ask for more data. Congratulations on a thorough and compelling paper.

We thank the referee for their appreciation of our work.

Referee #2

Summary: The authors have made a good faith attempt to address all concerns raised in the previous round of review, and I find this MS improved as a result. The most notable improvements are the addition of double HCR to address the btd/eve expression pattern in *C. riparius* and the attempts to find deleterious effects resulting from compromised CF formation in *D. melanogaster*, to support the authors' main claim that the CF is an adaptive evolutionary novelty. Despite these changes to the MS, many of which have improved an already strong MS, I remain of the opinion that the evidence that the cephalic furrow confers an evolutionary advantage that explains "why" it evolved, is not strong. I find their hypothesis for the divergent cell biological mechanisms regulating CF formation, and underlying the differences in CFs between dipterans, to be strongly supported by their experiments. However, I disagree with the authors regarding their strong

conclusion that CF evolution was an adaptive response that solved a mechanical problem in development.

We thank the referee for their appreciation of our work. We acknowledge the concern that an adaptive response to solve a mechanical problem in development is not the only possible explanation for the evolution of the CF. It has not been our intention to portray the evolution of CF formation in this manner. To clarify this, we have revised the manuscript accordingly and in response to the comments below.

Major comments:

1. Regarding the assertion that the presence of the CF, and that fact that it appears in all individuals of species that have a CF (which is how I believe the authors define "robustness" in this context), is evidence that it evolved by natural selection and confers an adaptive benefit: I appreciate the clarification of writing in many instances, and acknowledge that the authors are trying to make an argument for evidence of stabilizing selection of this trait. The expanded taxon sampling and the addition of the double HCR for *C. riparius* are strong additions that are relevant to this claim. Despite the expanded taxon sampling, in my view I don't believe that the authors can reject the hypothesis of loss of CF in some lineages simply because of the long evolutionary distances and wide radiation of the clade. The authors argue against loss "because then we would expect to see at least some variation or an occasional loss of CF formation." The question is whether or not we would expect that sampling 6 out of the over 70,000 described species of *Cyclorhapha* would happen to reveal at least one such loss. Through no fault of their own given the large evolutionary radiation - I am not suggesting that the authors sample all 70,000 species! - the authors cannot reject this null hypothesis, and they should at a minimum acknowledge this.

We thank the referee for pointing out the phrasing that could be viewed as overinterpretation. We did not intend to exclude the possibility that CF formation has been secondarily lost, and in the revised version of the manuscript, we have made sure that there is no sentence saying so. It was also not our intention to portray CF formation as a process that could have only evolved by natural selection and must confer an adaptive benefit, and we removed the statements that could possibly be interpreted as such.

Specifically, while revising and shortening the manuscript, we have paid particular attention to the following sections of our first revision:

ll 254-259: "The precision and robustness with which the CF is formed, combined with its deep evolutionary conservation across independent lineages and the absence of documented losses since its origin in the stem group of *Cyclorhapha*, suggests that its spatial patterning and temporal dynamics are subject to stabilizing selection. This suggests that CF formation provides a fitness advantage. To test this, we asked whether head-trunk buckling has a deleterious effect on embryonic development."

In the revised version, all references to the origin of CF formation, selection, and fitness have been removed. It now reads as follows (ll 219-220): “We next asked whether CF loss and head-trunk buckling have deleterious effects on embryonic development.”

ll 274-277: Although midline distortions are a clear deviation from normal embryonic development in *D. melanogaster*, we could not be certain that they have a deleterious effect on embryonic organismal fitness. Unfortunately, technical difficulties prevent us from assessing embryonic survival as a direct measure of fitness (see Methods). As a proxy, we sought to examine whether CF loss causes further, possibly more severe, developmental abnormalities that could be detrimental to the function or viability of the organism.

In the revised version, all references to fitness have been removed. It now reads as follows (ll 232-236): “Although VM distortions are a clear deviation from normal embryonic development in *D. melanogaster*, it is insufficient to indicate deleterious effects on embryonic development. To examine whether CF loss causes further, possibly more severe, developmental abnormalities detrimental to the function or viability of the organism, we monitored embryonic development for at least 18 hours after a one-hour optogenetic inhibition of CF at gastrulation onset.”

ll 288-291: “As HI ensures proper internalization of embryonic head segments that include several vital organs such as the mouth, while proper VNC condensation is necessary for the assembly and wiring of the central nervous system, these data suggest that loss of the CF has deleterious effects on embryonic fitness”.

In the revised version, we focus on “embryonic development” rather than “fitness”. Furthermore, we refer to Discussion for a more detailed view on their implications. The section now reads as follows (ll 242-246): “As HI ensures proper internalization of embryonic head segments, including vital organs such as the mouth^{34,35}, and VNC condensation promotes proper assembly and wiring of the central nervous system³⁵, these data suggest that CF loss is deleterious to embryonic development (see also Discussion and Supplementary Notes 3).”.

The section of Discussion now is the only place where we discuss more broadly the evolutionary implications of our findings, which include the possibility that the CF might have evolved as an adaptive response under the influence of mechanical stress, all while emphasizing that further work will be required to conclusively address this issue. We believe that raising this possibility is well justified, providing a crucial conceptual perspective that we hope the field as a whole will continue to explore going forward. The corresponding section now reads (ll383-387): “These findings are consistent with the hypothesis that the CF emerged under selection to mitigate the rise of mechanical instabilities at the head-trunk boundary during gastrulation (Fig. 6h). However, direct evidence for adaptation is lacking presently without explicit fitness measurements following CF ablation (see Supplementary Notes 3).”.

2. Regarding the claim that the CF is "beneficial" based on observing midline distortions alone: I appreciate that the authors extended their analysis of embryos in which they had disrupted the CF, to assess whether later stages of embryogenesis and/or embryonic survival at least to hatching were impacted. This is a strong addition to the MS. However, as for point (1) above, I find that the authors have interpreted these data too strongly. I can see in their modified Figure 3 shows numerically greater incidence of head involution and ventral nerve cord defects in the CF-disrupted embryos relative to shams (as a note of rigor, the authors should additionally perform a statistical comparison of these control and experimental distributions and provide the results in the MS).

We thank the referee for the appreciation of our new optogenetic dataset and the suggestion to perform a statistical comparison between control and experimental distributions. They are now provided in the manuscript as part of Figure 4f as follows: "f, Pie charts showing the percentage of embryos defective for ventral furrow deviation (VM), HI and VNC. Time is relative to the onset of gastrulation. Chi-square test of independence with Bonferroni correction; HI, $p=0.000039$; VNC, $p=0.0004$; VM, $p=0.0005$. Scale bars, 30 μm ."

The authors interpret this as evidence that CF disruption has specific negative impacts on head involution and ventral cord formation/morphogenesis. However, the fact that the genetic background they are using to induce specific CF disruption has such a high level of embryonic lethality, and that even sham embryos show high incidence of head involution defects and ventral nerve cord defects, suggests an alternative interpretation of these data to the one favored by the authors: namely, that these embryos already have a tendency to have problems in these specific developmental processes (among others), such that they are dealing with a sensitized background that is easy to "push over the edge" into having a higher incidence of the problems they already started with, by subjecting them to any kind of stress. I appreciate that the authors are working with this genetic background because it allows them to more specifically interfere with CF formation than the genetic mutants used in the companion study, whose interpretation was potentially complicated by pleiotropy. However, the additional experiments that the authors have performed for the revised MS have revealed that their genetic background, too, contains pleiotropic effects that are directly related to the developmental defects they would like to claim are caused by CF disruption. Glossing over these problems by calling them "technical difficulties" in the main text (line 382) is misleading. Thus, I do not find that the case for an adaptive benefit of CF formation to be any stronger in this revised MS, than in the original, although I reiterate that I appreciate the authors' expanded investigation into additional phenotypes in the CF-disrupted embryos.

As stated in response to point 1: it was not our intention to portray CF formation as a process that could have only evolved by natural selection and with adaptive benefit. In the revised version, we removed the statements that could possibly be interpreted in that direction.

With regard to the optogenetic DNRho1 experiments, we agree with the referee that the fly line we used represents a genetic background that is sensitive to stress, given that procedures of embryo handling and mounting alone, without imaging or photoactivation, can already cause high lethality. However, we respectfully disagree with the interpretation that the observed phenotypes upon optogenetic CF inhibition are the result of generalized stress or non-specific perturbation.

In the experiments where we observed more than a two-fold increase in the incidences of late embryonic defects, the only variable that distinguishes optogenetically stimulated embryos from sham controls was a one-hour, spatially confined blue-light illumination targeting the prospective CF region. This mild illumination protocol is not known to cause developmental defects. Consistent with prior studies that show that this optogenetic setup specifically causes local inhibition of myosin contractility, this targeted perturbation led to the inhibition of CF formation, followed by buckling at the head-trunk boundary that we know through our study was due to the accumulation of compressive stress resulting from the loss of the CF. The perturbation left all other developmental processes intact, however.

If the observed outcomes stemmed from generalized stress, we would expect a wider range of phenotypes and earlier developmental disruptions, rather than the reproducible and specific defects restricted to head involution and ventral nerve cord condensation that emerged several hours after the perturbation. Moreover, these late-stage, spatially-temporally restricted defects differ fundamentally from the pleiotropic effects of classic mutations in broadly expressed patterning genes, such as *eve* or *btd*. Developmental defects associated with such mutants occur in tissues and stages in which these genes are expressed. In contrast, our temporally and spatially limited optogenetic perturbation was not aimed at the process of head involution and ventral nerve cord condensation. Thus, we maintain that the most parsimonious explanation is that our targeted inhibition of CF formation leads to the observed late-stage defects.

In sum, our experiments provide clear evidence that loss of CF compromises embryonic development. In the meantime, we acknowledge that we do not have mechanistic insights into how loss of the CF leads to these late-stage embryonic defects, and we agree with the reviewer that these data are insufficient to claim that CF evolved under adaptive selection.

Minor comments:

1. I appreciate that the authors have used the accepted Linnean form of abbreviated genus/full species, after the first mention of the species with its full binary species name. However, the authors do not need to put the genus abbreviation/full species

into parentheses after the first incidence of species mention. For example, in line 65, the authors can simply say "Drosophila melanogaster" and then proceed to use "D. melanogaster" in line 74, without having put that abbreviation into parentheses in line 65.

Thank you. The revised manuscript now uses the Linnean nomenclature and abbreviations as suggested.

Referee #3

The revision of the manuscript by Dey et al. has strengthen the argument that they put forward. On its own it makes a strong case for an evolutionary divergence in the way a tissue in the early embryo copes with mechanical stresses. The additional optogenetic experiments are particularly enlightening. The response to the reviewers has resulted in a number of additional experiments and observations that make the case clear. The summary figure is much appreciated as it helps conceptualize the issues at stake and the solution. Together with Vellutini et al. make an important point for the field of evolutionary and developmental biology.

Alfonso Martinez Arias

We would like to thank Alfonso Martinez Arias for his kind words and the appreciation of our work; it means a lot to us.